# Directional Convergence Near Small Initializations and Saddles in Two-Homogeneous Neural Networks

**Akshay Kumar**                                                                  *kumar511@umn.edu*
*Department of Electrical and Computer Engineering*
*University of Minnesota, Minneapolis, MN*

**Jarvis Haupt**                                                                  *jdhaupt@umn.edu*
*Department of Electrical and Computer Engineering*
*University of Minnesota, Minneapolis, MN*

**Reviewed on OpenReview:** *https://openreview.net/forum?id=hfrPag75Y0*

## Abstract

This paper examines gradient flow dynamics of *two-homogeneous neural networks* for small initializations, where all weights are initialized near the origin. For both square and logistic losses, it is shown that for sufficiently small initializations, the gradient flow dynamics spend sufficient time in the neighborhood of the origin to allow the weights of the neural network to approximately converge in direction to the Karush-Kuhn-Tucker (KKT) points of a *neural correlation function* that quantifies the correlation between the output of the neural network and corresponding labels in the training data set. For square loss, it has been observed that neural networks undergo saddle-to-saddle dynamics when initialized close to the origin. Motivated by this, this paper also shows a similar directional convergence among weights of small magnitude in the neighborhood of certain saddle points.

## 1 Introduction

Massively overparameterized deep neural networks trained with (stochastic) gradient descent are widely known to be immensely successful architectures for inference. Recent works have attributed this success to the *implicit regularization* of gradient descent – the mysterious ability of gradient descent to find a solution that generalizes well despite the non-convexity of the loss landscape, the presence of spurious optima, and no explicit regularization (Neyshabur et al., 2015; Soudry et al., 2018).

To resolve this mystery, several works have studied the dynamics of gradient descent during training of neural networks (Jacot et al., 2018; Chizat et al., 2019; Mei et al., 2019; Chizat & Bach, 2018). An important observation emerging from these studies has been the effect of initialization on the trajectory of gradient descent. For large initialization, the gradient descent dynamics are approximately linear and can be described by the so-called *Neural Tangent Kernel* (NTK) (Jacot et al., 2018; Arora et al., 2019b). This regime is also referred to as *lazy training* (Chizat et al., 2019), since the weights of the neural networks do not change much and remain near their initializations throughout training, preventing the neural networks from learning the underlying features from the data.

In contrast, for small initializations, gradient descent dynamics is highly non-linear and exhibits *feature learning* (Geiger et al., 2020; Yang & Hu, 2021; Mei et al., 2019). Additionally, the benefit of small initializations over large in terms of generalization performance has also been observed under various settings (Geiger et al., 2020). For example, Chizat et al. (2019) train deep convolutional neural networks with varying scales of initialization while keeping other aspects of the neural network fixed, and a significant drop in performance is observed upon increasing the scale of initialization. In other recent works such as Jacot et al. (2022); Boursier et al. (2022); Pesme & Flammarion (2023), a phenomenon termed *saddle-to-saddle dynamics* has been observed during training. These works reveal that the trajectory of gradient descent passes through

a sequence of saddle points during training, in stark contrast to the linear dynamics observed in the NTK regime.

The investigation into the gradient descent dynamics of neural networks with small initializations has spurred numerous inquiries, yet a comprehensive theoretical framework remains elusive. The study of linear neural networks has led to valuable insights into the sparsity-inducing tendencies of gradient descent (Woodworth et al., 2020; Arora et al., 2019a). These tendencies also appear to be present in non-linear neural networks (Chizat et al., 2019; Chizat & Bach, 2018), however, rigorous results are limited to two-layer Rectified Linear Unit (ReLU) and Leaky-ReLU networks under various simple data-specific assumptions such as orthogonal inputs (Boursier et al., 2022), linearly separable data (Min et al., 2024; Wang & Ma, 2022; Lyu et al., 2021; Wang & Ma, 2023), the XOR mapping (Brutzkus & Globerson, 2019) or univariate data (Williams et al., 2019; Safran et al., 2022). Another important line of work has uncovered an intriguing phenomenon of directional convergence among neural network weights during the initial training stages (Maennel et al., 2018; Luo et al., 2021; Brutzkus & Globerson, 2019; Atanasov et al., 2022; Chen et al., 2023). In Maennel et al. (2018), it is shown that the weights of two-layer ReLU neural networks, trained using gradient flow with small initialization, converge in direction early in the training process while maintaining small norm. Although this result primarily describes dynamics near initialization, it constitutes a crucial step towards a comprehensive understanding of neural network training dynamics and has contributed significantly towards understanding the training dynamics in some of the aforementioned works (Boursier et al., 2022; Min et al., 2024; Wang & Ma, 2023; Lyu et al., 2021). However, the work of Maennel et al. (2018) is limited to two-layer ReLU networks and raises the question of whether this phenomenon holds for other neural networks.

## 2 Our Contributions

This work establishes the phenomenon of directional convergence in a more general setting. Specifically, we study the gradient flow dynamics resulting from training of two-homogeneous neural networks near small initializations and *also at certain saddle points*.

A neural network $\mathcal{H}$, where $\mathcal{H}(\mathbf{x}; \mathbf{w})$ is the real-valued output of the neural network, $\mathbf{x}$ is the input, and $\mathbf{w}$ is a vector containing all the weights, is defined here to be two-(positively) homogeneous if

$$\mathcal{H}(\mathbf{x}; c\mathbf{w}) = c^2 \mathcal{H}(\mathbf{x}; \mathbf{w}), \text{ for all } c \geq 0.$$

While this class does not encompass deep neural networks, it is broad enough to include several interesting types of neural networks. Let $\sigma(x)$ denote the ReLU (or Leaky-ReLU) function, then, some examples of two-homogeneous neural networks include

- Two-layer ReLU networks: $\mathcal{H}(\mathbf{x}; \{v_k, \mathbf{u}_k\}_{k=1}^H) = \sum_{k=1}^H v_k \sigma(\mathbf{x}^\top \mathbf{u}_k)$.

- Single-layer squared ReLU networks: $\mathcal{H}(\mathbf{x}; \{\mathbf{u}_k\}_{k=1}^H) = \sum_{k=1}^H p_k \sigma(\mathbf{x}^\top \mathbf{u}_k)^2$, where $p_k \in \{-1, 1\}$.

- Deep ReLU networks with only two trainable layers, for example $\mathcal{H}(\mathbf{x}; \mathbf{W}_1, \mathbf{W}_2) = \mathbf{v}^\top \sigma(\mathbf{W}_2 \sigma(\mathbf{W}_1 \mathbf{x}))$, where $\mathbf{v}$ is a fixed vector. We emphasize that this class includes any $L-$layer deep ReLU network with exactly two trainable layers (not necessarily two consecutive layers).

We consider a supervised learning setup for training and assume that $\{\mathbf{x}_i, y_i\}_{i=1}^n$ is the training dataset, $\mathbf{X} = [\mathbf{x}_1, \ldots, \mathbf{x}_n] \in \mathbb{R}^{d \times n}$, $\mathbf{y} = [y_1, \ldots, y_n]^\top \in \mathbb{R}^n$, and $\mathcal{H}(\mathbf{X}; \mathbf{w}) = [\mathcal{H}(\mathbf{x}_1; \mathbf{w}), \ldots, \mathcal{H}(\mathbf{x}_n; \mathbf{w})]^\top \in \mathbb{R}^n$ is the vector containing the output of neural network for all inputs, where $\mathbf{w} \in \mathbb{R}^k$. We do not make any structural assumptions on the training dataset.

In describing our results, a vital role is played by a quantity we refer to as the *neural correlation function* (NCF), which for a fixed vector $\mathbf{z} \in \mathbb{R}^n$ and neural network $\mathcal{H}$ as above is defined as

$$\mathcal{N}_{\mathbf{z}, \mathcal{H}}(\mathbf{w}) = \mathbf{z}^\top \mathcal{H}(\mathbf{X}; \mathbf{w}).$$

The NCF is a measure of the correlation between the vector $\mathbf{z}$ and the output of the neural network. For a given NCF, we refer to the following constrained optimization problem as a *constrained NCF*:

$$\max_{\|\mathbf{w}\|_2^2 = 1} \mathcal{N}_{\mathbf{z}, \mathcal{H}}(\mathbf{w}).$$

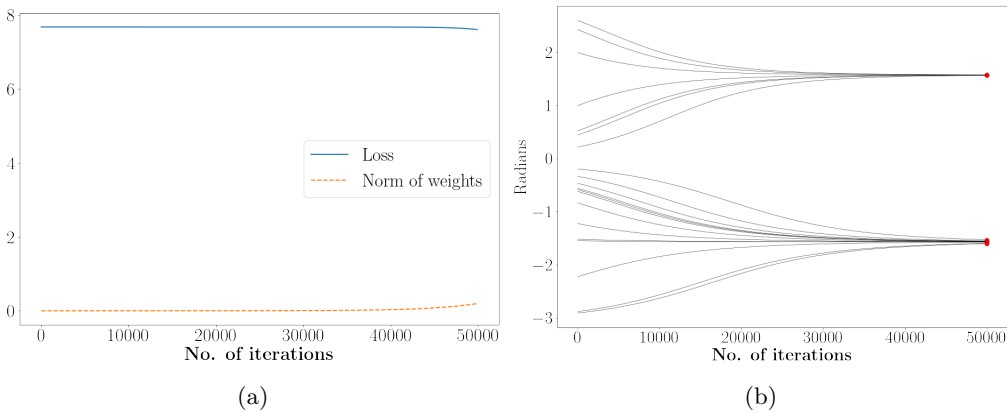

Figure 1: A two-dimensional scenario where a single-layer squared ReLU neural network with 20 hidden neurons is trained by gradient descent. The network architecture is defined as $\mathcal{H}(x_1, x_2; \{\mathbf{u}_i\}_{i=1}^{20}) = \sum_{i=1}^{20} \max(0, \mathbf{u}_{1i}x_1 + \mathbf{u}_{2i}x_2)^2$, where $\mathbf{u}_i$ represents the weights for the $i$th neuron. For training, we use 50 unit norm inputs and corresponding labels are generated using the function $\mathcal{H}^*(x_1, x_2) = 5\max(0, x_1)^2 + 4\max(0, -x_1)^2$. We use square loss and optimize using gradient descent for 50000 iterations with step-size $5 \cdot 10^{-5}$ . At initialization, the weights of each hidden neuron are drawn from Gaussian distribution with standard deviation $10^{-5}$. Panel ($a$): the evolution of training loss and the $\ell_2$-norm of all the weights with iterations. Panel ($b$): the evolution of $\arctan(\mathbf{u}_{2i}(t)/\mathbf{u}_{1i}(t))$ (the angle $\mathbf{u}_i(t)$ makes with the positive $x-$axis) for all hidden neurons. We see that the norm of the weights remain small and loss barely changes, though the weight vectors converge in direction to their final location (denoted with red dots).

Our first main result (Theorem 5.1) shows that, for square and logistic loss [1], if the initialization is sufficiently small, then, the gradient dynamics spends sufficient time near the origin such that the weights $\mathbf{w}$ are either approximately $\mathbf{0}$, or approximately converge in direction to non-negative Karush-Kuhn-Tucker (KKT) points of the constrained NCF

$$\max_{\|\mathbf{w}\|_2^2=1} \mathcal{N}_{\mathbf{y}, \mathcal{H}}(\mathbf{w}).$$

Our next main result (Theorem 5.5) shows a similar directional convergence near certain saddle points for square loss. Specifically, we show that if initialized in a sufficiently small neighborhood of that saddle point, then the gradient dynamics spends sufficient time near the saddle point such that the weights with small magnitude either approximately converge in direction to non-negative KKT points of the constrained NCF defined with respect to the residual error at that saddle point, or are approximately $\mathbf{0}$.

For illustration, we provide a brief "toy" example showing the phenomenon of directional convergence near small initialization. We train a single-layer squared ReLU neural network using gradient descent and small initialization, and provide in Figure 1 a visual depiction of ($a$) the overall loss and the $\ell_2$ norm of the network weights, and ($b$) the angle the weight vectors make with the positive horizontal axis, all as a function of the number of training iterations. (See the figure caption for more specific experimental details.) It is evident that the training loss barely changes, and the norm of all the weights remains small, indicating that gradient dynamics is still near the origin. This is not surprising since the origin is a saddle point. However, the directions of the individual weight vectors for the neurons undergo significant changes. This experiment suggests that while the gradient dynamics may not significantly change the weight vector magnitudes, it does change their directions. Further, and perhaps more interestingly, these directions not only change but also appear to converge. The objective of this paper is to explain how such a phenomenon could occur by just minimizing the loss using gradient descent, and also characterize the directions along which the weights converge.

Probably the most related existing work to our effort here is Maennel et al. (2018), which exclusively focuses on two-layer ReLU neural networks. Compared to that work, ours establishes directional convergence near small initializations for two-homogeneous neural networks, a much wider class of neural networks, highlighting

---

[1]The results in Theorem 5.1 apply, in fact, to a more general class of loss functions.

the inherent importance of homogeneity for these types of phenomena. As alluded above, this class also includes deep ReLU networks with only two trainable layers (not necessarily two consecutive layers), for which the results of Maennel et al. (2018) are inapplicable. Further, while Maennel et al. (2018) only focuses on initialization, we also establish directional convergence near certain saddle points. This extension is particularly pertinent because it has been observed in previous works that neural networks exhibit saddle-to-saddle dynamics under small initialization. Consequently, our result describing dynamics near small initialization and saddle points could be important for a better understanding of the training dynamics in the future.

Finally, while the result of Maennel et al. (2018) has certainly advanced our understanding, their analysis near small initializations relies on heuristic arguments and are not completely rigorous; see Min et al. (2024, Section 2.2) for specific details. Our proof technique is rigorous and fundamentally different from Maennel et al. (2018) to handle a wider class of neural networks.

## 3 Preliminaries

In this section, we briefly describe some preliminary concepts that will be useful in rigorously describing the problem.

Throughout the paper, $\| \cdot \|_2$ denotes the $\ell_2$ norm for a vector and the spectral norm for a matrix. For any $N \in \mathbb{N}$, we let $[N] = \{1, 2, \ldots, N\}$ denote the set of positive integers less than or equal to $N$. We denote derivatives by $\dot{x}(t) = \frac{dx(t)}{dt}$, and for the sake of brevity we may remove the independent variable $t$ if it is clear from the context. For a vector $\mathbf{x}$, $x_i$ denotes its $i$-th entry. The $k$-dimensional sphere is denoted by $\mathcal{S}^{k-1}$. A KKT point of an optimization problem is defined to be a non-negative KKT point if the objective value at that KKT point is non-negative.

A function $f : X \to \mathbb{R}$ is called *locally Lipschitz continuous* if for every $\mathbf{x} \in X$ there exists a neighborhood $U$ of $\mathbf{x}$ such that $f$ restricted to $U$ is Lipschitz continuous. A locally Lipschitz continuous function is differentiable almost everywhere (Borwein & Lewis, 2000, Theorem 9.1.2).

For any locally Lipschitz continuous function, $f : X \to \mathbb{R}$ , its *Clarke subdifferential* at a point $\mathbf{x} \in X$ is the set

$$\partial f(\mathbf{x}) = \text{conv} \left\{ \lim_{i \to \infty} \nabla f(\mathbf{x}_i) : \lim_{i \to \infty} \mathbf{x}_i = \mathbf{x}, \mathbf{x}_i \in \Omega \right\},$$

where $\Omega$ is any full-measure subset of $X$ such that $f$ is differentiable for all $\mathbf{x} \in \Omega$. The set $\partial f(\mathbf{x})$ is nonempty, convex, and compact for all $\mathbf{x} \in X$, and the mapping $\mathbf{x} \to \partial f(\mathbf{x})$ is upper-semicontinuous (Clarke et al., 1998, Proposition 1.5). We denote by $\overline{\partial} f(\mathbf{x})$ the unique minimum norm subgradient.

Since the neural networks considered in this paper may be non-smooth (as a function of $\mathbf{w}$), to rigorously define the gradient flow for such functions we use the notion of *o-minimal structures* (Coste, 2000). In particular, we consider neural networks that are definable under some *o-minimal structure*, a mild technical assumption that is satisfied by almost all modern neural networks (Ji & Telgarsky, 2020), including the examples presented in Section 2. Formally, an o-minimal structure is a collection $\mathcal{S} = \{\mathcal{S}_n\}_{n=1}^{\infty}$ where each $\mathcal{S}_n$ is a set of subsets of $\mathbb{R}^n$ containing all algebraic subsets of $\mathbb{R}^n$ and is closed under finite union and intersection, complement, projection, and Cartesian product. The elements of $\mathcal{S}_1$ are the finite unions of points and intervals. For a given o-minimal structure $\mathcal{S}$, a set $\mathbf{A} \subset \mathbb{R}^n$ is definable if $\mathbf{A} \in \mathcal{S}_n$. A function $f : \mathbf{D} \to \mathbb{R}^m$ with $\mathbf{D} \subset \mathbb{R}^n$ is definable if the graph of $f$ is in $\mathcal{S}_{n+m}$. Since a set remains definable under projection, the domain $\mathbf{D}$ is also definable; see Coste (2000) for a detailed introduction of o-minimal structures.

Using the notion of definability under o-minimal structures, we define *gradient flow* for non-smooth functions following Davis et al. (2018); Ji & Telgarsky (2020); Lyu & Li (2020). A function $\mathbf{z} : I \to \mathbb{R}^d$ on the interval $I$ is an *arc* if it is absolutely continuous for any compact sub-interval of $I$. An arc is differentiable almost everywhere, and the composition of an arc with a locally Lipschitz function is also an arc. For any locally Lipschitz and definable function $f(\mathbf{x})$, $\mathbf{x}(t)$ evolves under gradient flow of $f(\mathbf{x})$ if it is an arc, and

$$\dot{\mathbf{x}}(t) \in -\partial f(\mathbf{x}(t)), \text{ for a.e. } t \geq 0. \tag{1}$$

If $\mathbf{x}(t)$ evolves under positive gradient flow of $f(\mathbf{x})$, i.e., $\dot{\mathbf{x}}(t) \in \partial f(\mathbf{x}(t))$, for a.e. $t \geq 0$, we still call $\mathbf{x}(t)$ a gradient flow of $f(\mathbf{x})$. In what follows, it will be clear from the context whether it is positive or negative gradient flow.

## 4 Problem Setup

Within the framework introduced above, we consider the minimization of

$$L(\mathbf{w}) = \sum_{i=1}^{n} \ell(\mathcal{H}(\mathbf{x}_i; \mathbf{w}), y_i), \tag{2}$$

where $\ell(\hat{y}, y)$ is a loss function; in this work, we assume the loss function is definable and have locally Lipschitz gradient.

**Assumption 1.** The loss function $\ell(\hat{y}, y)$ is definable under some o-minimal structure that includes polynomials and exponentials, and $\nabla_{\hat{y}} \ell(\hat{y}, y)$ is locally Lipschitz in $\hat{y}$.

The above assumption is satisfied by commonly used loss functions such as square loss and logistic loss. Next, as alluded above, we also assume that the neural networks under consideration are *two-homogeneous*, a property we formalize via the following assumption.

**Assumption 2.** *For any fixed $\mathbf{x}$, $\mathcal{H}(\mathbf{x}; \mathbf{w})$ is locally Lipschitz and definable under some o-minimal structure that includes polynomials and exponentials, and for all $c \geq 0$, $\mathcal{H}(\mathbf{x}; c\mathbf{w}) = c^2 \mathcal{H}(\mathbf{x}; \mathbf{w})$.*

In Wilkie (1996), it was shown that there exists an o-minimal structure in which polynomials and exponential functions are definable. Also, the definability of a function is stable under algebraic operations, composition, inverse, maximum, and minimum. Since ReLU/Leaky-ReLU is a maximum of two polynomials, typical neural networks involving ReLU activation function are definable (Ji & Telgarsky, 2020). Also, under the above assumptions, $L(\mathbf{w})$ is definable. Finally, we also require $\mathcal{H}$ to be two-homogeneous for our results to hold, which rules out deep neural networks such as deep ReLU networks with more than 2 trainable layers.

Next, since $L(\mathbf{w})$ is definable, the gradient flow $\mathbf{w}(t)$ is an arc that satisfies for a.e. $t \geq 0$

$$\dot{\mathbf{w}}(t) \in -\partial L(\mathbf{w}(t)), \mathbf{w}(0) = \delta \mathbf{w}_0, \tag{3}$$

where $\mathbf{w}_0$ is a vector and $\delta$ is a positive scalar that controls the scale of initialization.

For differential inclusions, it is possible to have multiple solutions for the same initialization. This leads to technical difficulties in proving our results. We will address this difficulty by making use of the following definition which is inspired by Lyu et al. (2021) and will be discussed in more detail in the later sections.

**Definition 4.1.** Suppose $g(\mathbf{w}) : \mathbb{R}^k \to \mathbb{R}$ is locally Lipschitz and definable under some o-minimal structure, and consider the following differential inclusion with initialization $\tilde{\mathbf{w}}$

$$\frac{d\mathbf{w}}{dt} \in \partial g(\mathbf{w}), \mathbf{w}(0) = \tilde{\mathbf{w}}, \text{ for a.e. } t \geq 0.$$

We say $\tilde{\mathbf{w}}$ is a non-branching initialization if the differential inclusion has a unique solution for all $t \geq 0$.

Before proceeding to the main results, we briefly state additional definitions which are used in the paper. Let

$$\beta := \sup\{\|\mathcal{H}(\mathbf{X}; \mathbf{w})\|_2 : \mathbf{w} \in \mathcal{S}^{k-1}\},$$

where $\mathbf{X}$ and $\mathbf{y}$ denote the training examples and labels. For $\mathbf{z} \in \mathbb{R}^n$, we define $\ell'(\mathbf{z}, \mathbf{y}) = [\nabla_{\hat{y}} \ell(z_1, y_1), \ldots, \nabla_{\hat{y}} \ell(z_n, y_n)]^\top \in \mathbb{R}^n$. Now, since $\nabla_{\hat{y}} \ell(\hat{y}, y)$ is locally Lipschitz in $\hat{y}$, we define $\hat{\beta}$ such that if $\|\mathbf{z}\|_2 \leq \beta$, then

$$\|\ell'(\mathbf{z}, \mathbf{y}) - \ell'(\mathbf{0}, \mathbf{y})\|_2 \leq \hat{\beta}\|\mathbf{z}\|_2, \tag{4}$$

and $\tilde{\beta} = \hat{\beta}\beta + \|\ell'(\mathbf{0}, \mathbf{y})\|_2$.

## 5 Main Results

### 5.1 Directional Convergence Near Initialization

We are now in position to state our first main result establishing approximate directional convergence of the weights near small initialization.

**Theorem 5.1.** *Let* $\mathbf{w}_0$ *be a unit norm vector and a non-branching initialization of the differential inclusion*

$$\dot{\mathbf{u}} \in \partial \mathcal{N}_{-\ell'(\mathbf{0},\mathbf{y}),\mathcal{H}}(\mathbf{u}), \mathbf{u}(0) = \mathbf{w}_0. \tag{5}$$

*For any* $\epsilon \in (0,\eta)$, *where* $\eta$ *is a positive constant[2], there exist* $C > 1$ *and* $\overline{\delta} > 0$ *such that the following holds: for any* $\delta \in (0,\overline{\delta})$ *and solution* $\mathbf{w}(t)$ *of eq. (3) with initialization* $\mathbf{w}(0) = \delta \mathbf{w}_0$, *we have*

$$\|\mathbf{w}(t)\|_2 \le \sqrt{C}\delta, \ \text{for all } t \in \left[0,\overline{T}\right],$$

*where* $\overline{T} = \frac{\ln(C)}{4\beta\tilde{\beta}}$. *Further, either*

$$\|\mathbf{w}(\overline{T})\|_2 \ge \delta\eta, \ \text{and} \ \frac{\mathbf{w}(\overline{T})^\top \hat{\mathbf{u}}}{\|\mathbf{w}(\overline{T})\|_2} \ge 1 - \left(1 + \frac{3}{2\eta}\right)\epsilon,$$

*where* $\hat{\mathbf{u}}$ *is a non-negative KKT point of*

$$\max_{\|\mathbf{u}\|_2^2=1} \mathcal{N}_{-\ell'(\mathbf{0},\mathbf{y}),\mathcal{H}}(\mathbf{u}) = -\ell'(\mathbf{0},\mathbf{y})^\top \mathcal{H}(\mathbf{X};\mathbf{u}),$$

*or*

$$\|\mathbf{w}(\overline{T})\|_2 \le 2\delta\epsilon.$$

Here, $\epsilon$ represents the level of directional convergence of the weight, and $C$ represents how long gradient flow needs to stay near the origin to ensure the desired level of directional convergence.

In words, the first part of the result establishes that for a given choice of $\epsilon > 0$, we can choose $\delta$ sufficiently small such that the norm of the weights remains small for all $t \in [0,\overline{T}]$, indicating that gradient flow remains near the origin. The second part quantifies what happens at the time $\overline{T}$; there are two possible outcomes. In one scenario, the weights approximately converge in direction towards a non-negative KKT point of the constrained NCF defined with respect to $-\ell'(\mathbf{0},\mathbf{y})$ and neural network $\mathcal{H}$ (additionally, $\|\mathbf{w}(\overline{T})\|_2 \ge \delta\eta$, where $\eta$ is a constant that does not depend on $\delta$.) In contrast, in the second scenario $\|\mathbf{w}(\overline{T})\|_2 \le 2\delta\epsilon$, where we can choose $\epsilon$ and $\delta$ both to be arbitrarily small. Thus, compared to the first scenario, in the second scenario, the weights get much closer to the origin. In fact, as it will become more clear from the proof sketch later, this happens because the gradient dynamics of the NCF can converge to $\mathbf{0}$.

It is easy to verify that for square loss, $\ell(\hat{y},y) = \frac{1}{2}(\hat{y}-y)^2$ and $-\ell'(\mathbf{0},\mathbf{y}) = \mathbf{y}$. Similarly, for logistic loss, $\ell(\hat{y},y) = \ln(1+e^{-\hat{y}y})$ and $-\ell'(\mathbf{0},\mathbf{y}) = \mathbf{y}/2$. Hence, for both of these loss functions, the weights approximately converge in direction towards a non-negative KKT point of

$$\max_{\|\mathbf{u}\|_2^2=1} \mathcal{N}_{\mathbf{y},\mathcal{H}}(\mathbf{u}) = \mathbf{y}^\top \mathcal{H}(\mathbf{X};\mathbf{u}).$$

Now, for general training data, it may not be possible to get a closed form expression for the KKT points of the NCF. In Appendix G we provide some simple examples where closed form expressions can be computed, which may be helpful to the readers.

Finally, note that we require $\mathbf{w}_0$ to be a non-branching initialization of eq. (5). The necessity for such a requirement essentially arises because there could exist multiple solutions for differential inclusions. We discuss it in more detail after providing the proof sketch of the above theorem. However, we note that if the neural network $\mathcal{H}(\mathbf{x};\mathbf{w})$ has locally Lipschitz gradients then this requirement is always satisfied, since in that case eq. (5) always has a unique solution. This would include, for example, the squared ReLU neural network.

---

[2]Here, $\eta$ depends on the solution of eq. (5), which solely relies on $\mathbf{X}, \mathbf{y}, \mathcal{H}, \mathbf{w}_0$, and is independent of $\delta$. See Lemma C.7 and the proof for more details.

### 5.1.1  Proof Sketch of Theorem 5.1

We provide a brief proof sketch for Theorem 5.1 here; the complete proof can be found in Appendix C. The proof ultimately relies upon two lemmas. The first one describes the approximate dynamics of $\mathbf{w}(t)$ in the initial stages of training for small initialization.

**Lemma 5.2.** *Let $C > 1$ be an arbitrarily large constant and $\mathbf{w}(t)$ be any solution of eq. (3) with initialization $\mathbf{w}(0) = \delta\mathbf{w}_0$, where $\delta \leq \sqrt{\frac{1}{C}}$ and $\|\mathbf{w}_0\|_2 = 1$. Then, for all $t \in \left[0, \frac{\ln(C)}{4\beta\tilde{\beta}}\right]$,*

$$\|\mathbf{w}(t)\|_2 \leq \sqrt{C}\delta. \tag{6}$$

*Further, for the differential inclusion*

$$\dot{\mathbf{u}} \in \partial\mathcal{N}_{-\ell'(\mathbf{0},\mathbf{y}),\mathcal{H}}(\mathbf{u}), \mathbf{u}(0) = \mathbf{w}_0, \tag{7}$$

*and any $\epsilon > 0$ there exists a small enough $\overline{\delta} > 0$ such that for any $\delta \in (0, \overline{\delta})$,*

$$\left\|\frac{\mathbf{w}(t)}{\delta} - \mathbf{u}(t)\right\|_2 \leq \epsilon, \text{ for all } t \in \left[0, \frac{\ln(C)}{4\beta\tilde{\beta}}\right], \tag{8}$$

*where $\mathbf{u}(t)$ is a certain solution of eq. (7).*

The first part of the lemma shows that for sufficiently small $\delta$ the gradient dynamics can spend an arbitrarily large time near the origin. To understand the implications of the second part, let us first focus on the differential inclusion in eq. (7), which is the positive gradient flow of the NCF defined with respect to $-\ell'(\mathbf{0}, \mathbf{y})$ and neural network $\mathcal{H}$. From eq. (8), we observe that for small initialization, in the initial stages of the dynamics, $\mathbf{w}(t)/\delta$ is approximately equal to $\mathbf{u}(t)$. To get an intuitive idea about the proof of the above lemma, consider the evolution of $\mathbf{w}(t)$ near small initializations, which approximately can be expressed as

$$\dot{\mathbf{w}} \in -\sum_{i=1}^{n} \nabla_{\hat{y}}\ell(\mathcal{H}(\mathbf{x}_i; \mathbf{w}), y_i)\partial\mathcal{H}(\mathbf{x}_i; \mathbf{w}) \approx -\sum_{i=1}^{n} \nabla_{\hat{y}}\ell(0, y_i)\partial\mathcal{H}(\mathbf{x}_i; \mathbf{w}).$$

Since $\mathcal{H}(\mathbf{x}; \mathbf{w})$ is two-homogeneous, we have that $\partial\mathcal{H}(\mathbf{x}; \mathbf{w})$ is one-homogeneous. Hence, dividing the above equation by $\delta$ we get

$$\dot{\mathbf{w}}/\delta \approx -\frac{1}{\delta}\sum_{i=1}^{n} \nabla_{\hat{y}}\ell(0, y_i)\partial\mathcal{H}(\mathbf{x}_i; \mathbf{w}) = -\sum_{i=1}^{n} \nabla_{\hat{y}}\ell(0, y_i)\partial\mathcal{H}(\mathbf{x}_i; \mathbf{w}/\delta) = \partial\mathcal{N}_{-\ell'(\mathbf{0},\mathbf{y}),\mathcal{H}}(\mathbf{w}/\delta).$$

Since $\mathbf{w}(0)/\delta = \mathbf{w}_0$, from the above equation we observe that the evolution of $\mathbf{w}(t)/\delta$ is approximately governed by the differential inclusion in eq. (7). Thus, one expects $\mathbf{w}(t)/\delta$ to be close to a certain solution of eq. (7), provided $\|\mathbf{w}(t)\|_2$ remains small.

Now, since $\delta$ is a positive scalar, dividing $\mathbf{w}(t)$ by it does not change the direction of $\mathbf{w}(t)$. Further, note that the dynamics of $\mathbf{u}(t)$ do not depend on $\delta$ and the approximation in eq. (8) can be made to hold for an arbitrarily long time by choosing sufficiently small $\delta$. Thus, if we choose $C$ large enough such that the approximation in eq. (8) is valid for a sufficiently long time in which $\mathbf{u}(t)$ approximately converges in direction, then by virtue of eq. (8), $\mathbf{w}(t)$ would also approximately converge in direction.

Thus, our aim is to establish approximate directional convergence of $\mathbf{u}(t)$ within some finite time. For this, we turn towards analyzing the gradient flow dynamics of the NCF. Recall that, for a given vector $\mathbf{z}$, the NCF is defined as

$$\mathcal{N}_{\mathbf{z},\mathcal{H}}(\mathbf{u}) = \mathbf{z}^{\top}\mathcal{H}(\mathbf{X}; \mathbf{u}), \tag{9}$$

and gradient flow will satisfy for a.e. $t \geq 0$

$$\frac{d\mathbf{u}}{dt} \in \partial\mathcal{N}_{\mathbf{z},\mathcal{H}}(\mathbf{u}), \mathbf{u}(0) = \mathbf{u}_0, \tag{10}$$

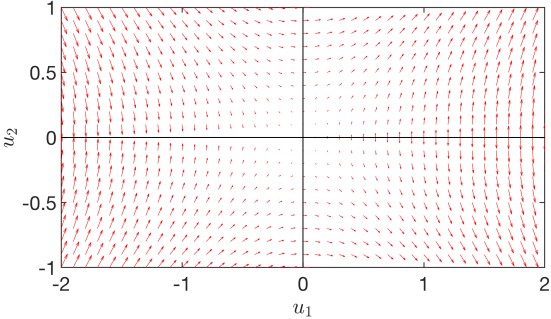

Figure 2: The gradient field of $f(u_1, u_2) = u_1|u_2|$.

where $\mathbf{u}_0$ is the initialization.

Since the function value increases along the gradient flow trajectory and $\mathcal{N}_{\mathbf{z},\mathcal{H}}(\mathbf{u})$ may not be bounded from above, the gradient flow trajectory can potentially diverge to infinity and take $\mathcal{N}_{\mathbf{y},\mathcal{H}}(\mathbf{u})$ to infinity along with it. However, in the following lemma we show that the gradient flow will always converge in direction, and also characterize those directions.

**Lemma 5.3.** *For any solution* $\mathbf{u}(t)$ *of eq. (10), either* $\lim_{t\to\infty} \mathcal{N}_{\mathbf{z},\mathcal{H}}(\mathbf{u}(t)) = \infty$ *or* $\lim_{t\to\infty} \mathcal{N}_{\mathbf{z},\mathcal{H}}(\mathbf{u}(t)) = 0$. *Also, either* $\lim_{t\to\infty} \frac{\mathbf{u}(t)}{\|\mathbf{u}(t)\|_2}$ *exists or* $\lim_{t\to\infty} \mathbf{u}(t) = 0$. *If* $\lim_{t\to\infty} \frac{\mathbf{u}(t)}{\|\mathbf{u}(t)\|_2}$ *exists then its value, say* $\mathbf{u}^*$, *must be a non-negative KKT point of the optimization problem*

$$\max_{\|\mathbf{u}\|_2^2=1} \mathcal{N}_{\mathbf{z},\mathcal{H}}(\mathbf{u}) = \mathbf{z}^\top \mathcal{H}(\mathbf{X}; \mathbf{u}). \tag{11}$$

The above lemma states that any solution of eq. (10) will either converge to $\mathbf{0}$ or converge in direction to a KKT point of the constrained NCF. To establish directional convergence in the above lemma, we follow a similar technique as in Ji & Telgarsky (2020, Theorem 3.1).

To prove Theorem 5.1 from here, we combine Lemma 5.2 and Lemma 5.3. From a given initialization $\delta\mathbf{w}_0$, we get $\mathbf{w}_0$. Then, for the solution $\mathbf{u}(t)$ of the differential inclusion in eq. (7), using Lemma 5.3, we choose $\overline{T}$ large enough such that $\mathbf{u}(\overline{T})$ either approximately converges in direction to the KKT point of the NCF or gets close to $\mathbf{0}$. Then, based on that $\overline{T}$, using Lemma 5.2, we choose $\delta$ sufficiently small such that $\mathbf{w}(t)/\delta$ is close to $\mathbf{u}(t)$, for all $t \in [0, \overline{T}]$. The result follows.

There is, however, one issue with the above argument. The differential inclusions could have multiple solutions, and in eq. (8), the approximation holds for *some* solution of eq. (7); it is not known beforehand which solution it would be. Therefore, we would need to choose $\overline{T}$ large enough such that *all* solutions of eq. (7) have approximately converged in direction. However, this may not be possible for all initializations $\mathbf{w}_0$. To illustrate this we consider a simple example.

Consider the function $f(u_1, u_2) = u_1|u_2|$ that satisfies Assumption 2, and is differentiable everywhere except along the line $u_2 = 0$. In Figure 2 we plot its gradient field. Note that $\tilde{\mathbf{u}} = [1, 0]^\top$ is a critical point for $f(u_1, u_2)$, i.e., $\mathbf{0} \in \partial f(\tilde{\mathbf{u}})$. Thus, if initialized at $\tilde{\mathbf{u}}$, one possible gradient flow solution is to stay at $\tilde{\mathbf{u}}$ for all $t \geq 0$. However, $\partial f(\tilde{\mathbf{u}})$ contains other vectors which could lead to gradient flow escaping from $\tilde{\mathbf{u}}$. Moreover, one could construct a gradient flow solution that can spend arbitrary amount of time at $\tilde{\mathbf{u}}$ before escaping it. Specifically, for any finite $T$,

$$\mathbf{u}_T(t) = \begin{cases} \begin{bmatrix} 1 \\ 0 \end{bmatrix}, & \text{for all } t \in [0, T] \\ \begin{bmatrix} \cosh{(t-T)} \\ \sinh{(t-T)} \end{bmatrix}, & \text{for all } t \geq T, \end{cases}$$

is a possible gradient flow solution. We note that $\lim_{t\to\infty} \mathbf{u}_T(t)/\|\mathbf{u}_T(t)\|_2 = [1/\sqrt{2}, 1/\sqrt{2}]^\top$, however, clearly for any finite time $\overline{T}$ we can choose $T$ large such that $\mathbf{u}_T(\overline{T})/\|\mathbf{u}_T(\overline{T})\|_2$ stays away from $[1/\sqrt{2}, 1/\sqrt{2}]^\top$.

Thus, we can not establish finite time approximate directional convergence for all possible gradient flow solutions. Complete details for this example can be found in Appendix E.

To address this issue, we only consider initialization which leads to a unique solution. In particular, we assume $\mathbf{w}_0$ to be a non-branching initialization of eq. (5). As noted earlier, if the neural network has locally Lipschitz gradients, then this requirement is always satisfied. However, for more general networks such as two-layer ReLU neural networks the above assumption may appear somewhat restrictive. That said, it is worth noting that a similar assumption was also made in Lyu et al. (2021), where the full training dynamics of two-layer Leaky-ReLU neural networks were investigated in a simple setting involving linearly separable data. Furthermore, Maennel et al. (2018) addresses this challenge of non-uniqueness by asserting that the differential inclusion resulting from the gradient flow of the loss function will have a unique solution in "almost all" cases, but do not provide a formal proof. We leave it as an important future research direction to handle more general initializations.

Another potential research direction is to characterize the set of non-branching initializations. For example, in the function depicted in Figure 2, it can be shown that except for the set $\{u_2 = 0, u_1 > 0\}$, all the other points are non-branching, which implies that almost all points are non-branching; for details see Lemma E.2. Whether a similar behavior happen for a broader class of datasets and neural networks is an interesting future direction, and a positive answer would justify the non-branching assumption in those cases. However, establishing such a result, even for a two-layer (Leaky) ReLU neural networks, seems to be difficult. In the following lemma, we show that for two-layer Leaky-ReLU neural networks there exists sets, near certain KKT points, within which all points are non-branching. We hope it can help future works in better characterization of the set of non-branching initializations.

**Lemma 5.4.** *Let $\mathcal{H}(\mathbf{x}; v, \mathbf{u}) = v\sigma(\mathbf{x}^\top \mathbf{u})$, where $\sigma(x) = \max(x, \alpha x)$, for some $\alpha \in \mathbb{R}$. Suppose $\{v_*, \mathbf{u}_*\}$ is a KKT point of*

$$\max_{v^2 + \|\mathbf{u}\|_2^2 = 1} \mathcal{N}_{\mathbf{z}, \mathcal{H}}(v, \mathbf{u}) = v\mathbf{z}^\top \sigma(\mathbf{X}^\top \mathbf{u}),$$

*where $v_* \mathbf{z}^\top \sigma(\mathbf{X}^\top \mathbf{u}_*) > 0$ and $\min_{i \in [n]} |\mathbf{x}_i^\top \mathbf{u}_*| > 0$. Then there exists a $\gamma > 0$ such that every element of the set $\mathcal{S} = \{v, \mathbf{u} : sign(v_*)v > \|\mathbf{u}\|_2, \|\mathbf{u} - \mathbf{u}_*\|_2 \leq \gamma\}$ is a non-branching initialization of the differential inclusion*

$$\begin{bmatrix} \dot{v} \\ \dot{\mathbf{u}} \end{bmatrix} \in \begin{bmatrix} \partial_v \mathcal{N}_{\mathbf{z}, \mathcal{H}}(v, \mathbf{u}) \\ \partial_{\mathbf{u}} \mathcal{N}_{\mathbf{z}, \mathcal{H}}(v, \mathbf{u}) \end{bmatrix}.$$

The proof can be found in Appendix F. Note that in the above lemma $\min_{i \in [n]} |\mathbf{x}_i^\top \mathbf{u}_*| > 0$ implies that the hyperplane defined by $\mathbf{u}_*$ does not pass through any vector in the training set. Also, although we have only considered a single neuron, the above lemma can be extended to multi-neuron setting by ensuring that the initialization of each neuron is in a set similar to $\mathcal{S}$.

### 5.1.2 A Corollary for Separable Neural Networks

We next consider the case when $\mathcal{H}(\mathbf{x}; \mathbf{w})$ is separable and can be divided into smaller neural networks. In the following lemma, we describe the directional convergence for such neural networks near small initialization.

**Corollary 5.4.1.** *Suppose we can write $\mathbf{w} = [\mathbf{w}_1, \ldots, \mathbf{w}_H]^\top$ such that $\mathcal{H}(\mathbf{x}; \mathbf{w}) = \sum_{i=1}^{H} \mathcal{H}_i(\mathbf{x}; \mathbf{w}_i)$, for all $\mathbf{x}$. Consider the same setting as in Theorem 5.1, then, for any $\epsilon \in (0, \eta)$, where $\eta$ is a positive constant, there exist $C > 1$ and $\overline{\delta} > 0$ such that for any $\delta \in (0, \overline{\delta})$, and for all $i \in [H]$, either*

$$\sqrt{C}\delta \geq \|\mathbf{w}_i(\overline{T})\|_2 \geq \delta\eta, \text{ and } \frac{\mathbf{w}_i(\overline{T})^\top \hat{\mathbf{u}}_i}{\|\mathbf{w}_i(\overline{T})\|_2} \geq 1 - \left(1 + \frac{3}{2\eta}\right)\epsilon,$$

*where $\hat{\mathbf{u}}_i$ is a non-negative KKT point of*

$$\max_{\|\mathbf{u}\|_2^2 = 1} \mathcal{N}_{-\ell'(\mathbf{0}, \mathbf{y}), \mathcal{H}_i}(\mathbf{u}) = -\ell'(\mathbf{0}, \mathbf{y})^\top \mathcal{H}_i(\mathbf{X}; \mathbf{u}), \tag{12}$$

*or*

$$\|\mathbf{w}_i(\overline{T})\|_2 \leq 2\delta\epsilon, \text{ where } \overline{T} = \frac{\ln(C)}{4\beta\tilde{\beta}}.$$

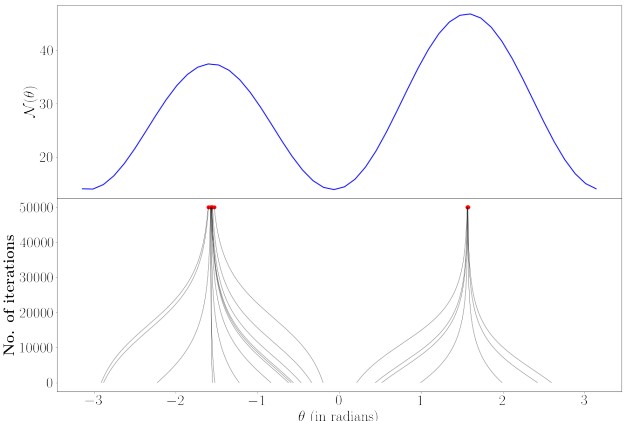

Figure 3: The lower part shows the content of Figure 1b with the horizontal and vertical axes interchanged. The top plot shows the constrained NCF $\mathcal{N}_{\mathbf{y},\mathcal{H}}(\theta) = \sum_{i=1}^{n} y_i \max(0, [\cos(\theta), \sin(\theta)]^{\top} \mathbf{x}_i)^2$. As predicted by Corollary 5.4.1, the neuron weights converge in direction to the KKT points of the NCF.

The above result establishes that for separable neural networks, the weights of smaller neural networks approximately converge in direction to the KKT points of the optimization problem in eq. (12), the constrained NCF defined with respect to the output of smaller neural networks.

Indeed, this is precisely what we observed for the "toy" experiments depicted in Figure 1. Recall, in that case, the neural network was the sum of squared ReLU functions and hence satisfies the setting of Corollary 5.4.1. In the bottom of Figure 3, we again plot the evolution of the direction of the weights for each hidden neuron. On the top, we plot the constrained NCF with respect to the output of each neuron, which will be identical for each neuron. We clearly observe that the weights of each neuron converge in direction towards KKT points of the constrained NCF.

Finally, we would like to emphasize that while in Theorem 5.1 the directional convergence is established for all the weights, Corollary 5.4.1 further establishes directional convergence for weights of smaller neural networks as well. In fact, the results in previous works on 2-layer ReLU neural network (Maennel et al., 2018), which is separable, are along the lines of Corollary 5.4.1 and it is possible to get those results using Corollary 5.4.1. We provide these details in Appendix H.

## 5.2 Directional Convergence Near Saddle Points

Several theoretical and empirical works have observed a saddle-to-saddle dynamics during training of neural networks with small initialization and square loss (Jacot et al., 2022; Pesme & Flammarion, 2023; Boursier et al., 2022; Jin et al., 2023). The evolution of loss alternates between being stagnant and decreasing sharply, almost like a piecewise constant function. This indicates that weights move from one saddle of the loss function to another during training. Some theoretical works further show that at each saddle point only certain number of weights are non-zero. For example, Pesme & Flammarion (2023) proves saddle-to-saddle dynamics in two-homogeneous diagonal linear networks, where at each saddle points only few weights are non-zero. The authors of Boursier et al. (2022) study two-layer ReLU network with orthogonal inputs, and show that gradient flow enters neighborhood of a saddle point where one set of neurons have high norm while others have zero norm.

In this section, we show that the directional convergence near initialization described in the previous section also occurs near certain saddle points for square loss. However, there could be different kinds of saddle points throughout the loss landscape. The choice of saddle points considered here is motivated by the above observations, such as only a certain number of weights being non-zero at the saddle points.

We assume that the weights of neural network $\mathbf{w}$ can be divided into two sets, $\mathbf{w} = [\mathbf{w}_n, \mathbf{w}_z]$, such that $\mathcal{H}(\mathbf{x}; \mathbf{w}) = \mathcal{H}_n(\mathbf{x}; \mathbf{w}_n) + \mathcal{H}_z(\mathbf{x}; \mathbf{w}_z)$, where $\mathcal{H}_n(\mathbf{x}; \mathbf{w}_n)$ and $\mathcal{H}_z(\mathbf{x}; \mathbf{w}_z)$ each satisfy Assumption 2. For square

loss, we minimize

$$L\left(\mathbf{w}_n, \mathbf{w}_z\right) = \frac{1}{2} \sum_{i=1}^n \|\mathcal{H}_n(\mathbf{x}_i; \mathbf{w}_n) + \mathcal{H}_z(\mathbf{x}_i; \mathbf{w}_z) - y_i\|^2 = \frac{1}{2} \|\mathcal{H}_n(\mathbf{X}; \mathbf{w}_n) + \mathcal{H}_z(\mathbf{X}; \mathbf{w}_z) - \mathbf{y}\|^2. \quad (13)$$

The saddle point of eq. (13) that we consider here satisfies the following.

**Assumption 3.** *We assume* $\{\overline{\mathbf{w}}_n, \overline{\mathbf{w}}_z\}$ *is a saddle point of eq. (13) such that*

$$\|\overline{\mathbf{w}}_n\|_2 \in [m, M], \ and \ \|\overline{\mathbf{w}}_z\|_2 = 0, \quad (14)$$

*where* $m, M$ *are positive constants. Further, if* $\mathcal{H}_n(\mathbf{x}; \mathbf{w}_n)$ *does not have a locally Lipschitz gradient, then we assume that there exists* $\gamma, \kappa > 0$ *such that for all* $\mathbf{w}_n$ *satisfying* $\|\mathbf{w}_n - \overline{\mathbf{w}}_n\|_2 \le \gamma$ *it holds that*

$$\langle \overline{\mathbf{w}}_n - \mathbf{w}_n, \mathbf{s} \rangle \ge -\kappa \|\overline{\mathbf{w}}_n - \mathbf{w}_n\|_2^2, where \ \mathbf{s} \in -\partial_{\mathbf{w}_n} L(\mathbf{w}_n, \mathbf{0}). \quad (15)$$

In the above assumption, $\overline{\mathbf{w}}_n$ is the set of weights with high norm while $\overline{\mathbf{w}}_z$ contains sets of weights with zero norm. Due to homogeneity, $\mathcal{H}_z(\mathbf{x}; \overline{\mathbf{w}}_z) = \mathbf{0}$, and thus $\mathcal{H}_n(\mathbf{x}; \overline{\mathbf{w}}_n)$ is effectively the output of the neural network at $\{\overline{\mathbf{w}}_n, \overline{\mathbf{w}}_z\}$.

When $\mathcal{H}_n(\mathbf{x}; \mathbf{w}_n)$ does not have locally Lipschitz gradient, we require eq. (15) to ensure that if $\mathbf{w}_n$ initialized near $\overline{\mathbf{w}}_n$, then it stays near it for a sufficiently long time. We discuss the motivation for this inequality after discussing our main theorem of this section, stated below.

**Theorem 5.5.** *Let* $\{\overline{\mathbf{w}}_n, \overline{\mathbf{w}}_z\}$ *satisfy Assumption 3, and define* $\overline{\mathbf{y}} = \mathbf{y} - \mathcal{H}_n(\mathbf{X}; \overline{\mathbf{w}}_n)$*. Suppose* $\zeta_z$ *is a unit norm vector and a non-branching initialization of the differential inclusion*

$$\dot{\mathbf{u}} \in \partial \mathcal{N}_{\overline{\mathbf{y}}, \mathcal{H}_z}(\mathbf{u}), \mathbf{u}(0) = \zeta_z. \quad (16)$$

*For any* $\epsilon \in (0, \eta)$*, where* $\eta$ *is a positive constant[3], there exist* $C > 1$ *and* $\overline{\delta} > 0$ *such that the following holds: for any* $\delta \in (0, \overline{\delta})$ *and gradient flow solution* $\{\mathbf{w}_n(t), \mathbf{w}_z(t)\}$ *of eq. (13) that satisfies for a.e.* $t \ge 0$

$$\begin{bmatrix} \dot{\mathbf{w}}_n \\ \dot{\mathbf{w}}_z \end{bmatrix} \in - \begin{bmatrix} \partial_{\mathbf{w}_n} L\left(\mathbf{w}_n, \mathbf{w}_z\right) \\ \partial_{\mathbf{w}_z} L\left(\mathbf{w}_n, \mathbf{w}_z\right) \end{bmatrix}, \begin{bmatrix} \mathbf{w}_n(0) \\ \mathbf{w}_z(0) \end{bmatrix} = \begin{bmatrix} \overline{\mathbf{w}}_n + \delta \zeta_n \\ \overline{\mathbf{w}}_z + \delta \zeta_z \end{bmatrix}, \quad (17)$$

*where* $\zeta_n$ *is a unit norm vector, we have*

$$\|\mathbf{w}_n(t) - \overline{\mathbf{w}}_n\|_2^2 + \|\mathbf{w}_z(t) - \overline{\mathbf{w}}_z\|_2^2 \le C\delta^2, \ for \ all \ t \in [0, \overline{T}], \quad (18)$$

*where* $\overline{T} = \frac{1}{M_2} \ln(C)$*, and* $M_2$ *is a constan[4]. Further, either*

$$\|\mathbf{w}_z(\overline{T})\|_2 \ge \delta \eta, \ and \ \frac{\mathbf{w}_z(\overline{T})^\top \hat{\mathbf{u}}}{\|\mathbf{w}_z(\overline{T})\|_2} \ge 1 - \left(1 + \frac{3}{2\eta}\right) \epsilon,$$

*where* $\hat{\mathbf{u}}$ *is a non-negative KKT point of*

$$\max_{\|\mathbf{u}\|_2^2 = 1} \mathcal{N}_{\overline{\mathbf{y}}, \mathcal{H}_z}(\mathbf{u}) = \overline{\mathbf{y}}^\top \mathcal{H}_z(\mathbf{X}; \mathbf{u}),$$

*or*

$$\|\mathbf{w}_z(\overline{T})\|_2 \le 2\delta \epsilon.$$

In the above theorem, the initialization is near a saddle point which satisfies Assumption 3, and $\delta$ controls how far the initialization is from the saddle point. The vector $\overline{\mathbf{y}}$ represents the residual error at the saddle point and plays the same role as $-\ell'(\mathbf{0}, \mathbf{y})$ did in Theorem 5.1. Provided $\zeta_z$ is a non-branching initialization

---

[3]Here, $\eta$ depends on the solution of eq. (16), which solely relies on $\mathbf{X}, \overline{\mathbf{y}}, \mathcal{H}_z, \zeta_z$, and is independent of $\delta$. See the proof for more details.

[4]$M_2$ depends on $\beta, \|\overline{\mathbf{y}}\|_2$ and various parameters associated with $\mathcal{H}_z$ and $\mathcal{H}_n$ near $\{\overline{\mathbf{w}}_n, \overline{\mathbf{w}}_z\}$ such as their Lipschitz constant, maximum value etc. Importantly, it does not depend on $\epsilon$ and $\delta$.

of eq. (16), we show that for sufficiently small $\delta$, the gradient flow spends enough time near the saddle point such that weights of small magnitude $\mathbf{w}_z$ either approximately converge in direction to the KKT point of the NCF defined with respect to $\overline{\mathbf{y}}$ and $\mathcal{H}_z$, or gets close to $\mathbf{0}$. The above theorem, similar to Theorem 5.1, shows directional convergence among weights of small magnitude. The proof technique is similar to the proof of Theorem 5.1; for details see Appendix D. We also demonstrate the phenomenon of directional convergence near saddle points using a numerical experiment in Appendix D.

We next explain the motivation for eq. (15) in Assumption 3 when $\mathcal{H}_n(\mathbf{x}; \mathbf{w}_n)$ does not have locally Lipschitz gradients. Our proof of the above theorem crucially relies on showing that $\mathbf{w}_n(t)$ remains close to $\overline{\mathbf{w}}_n$ and $\mathbf{w}_z(t)$ remains small for a sufficiently long time; i.e., eq. (18) holds. Suppose $\mathbf{w}_z(t)$ remains small, then the evolution of $\mathbf{w}_n(t)$ is approximately governed by the gradient flow of $L(\mathbf{w}_n, 0)$. To understand the gradient flow dynamics of $L(\mathbf{w}_n, 0)$ near $\overline{\mathbf{w}}_n$, we use the following lemma.

**Lemma 5.6.** *Suppose $\{\overline{\mathbf{w}}_n, \overline{\mathbf{w}}_z\}$ is a saddle point of eq. (13) such that $\|\overline{\mathbf{w}}_n\|_2 \in [m, M]$, and $\|\overline{\mathbf{w}}_z\|_2 = 0$. Then,*

$$\mathbf{0} \in \partial_{\mathbf{w}_n} L(\overline{\mathbf{w}}_n, 0).$$

If $\mathcal{H}_n(\mathbf{x}; \mathbf{w}_n)$ has locally Lipschitz gradients, then $\nabla_{\mathbf{w}_n} L(\mathbf{w}_n, 0)$ would be small and vary smoothly in the neighborhood of $\overline{\mathbf{w}}_n$. This suffices to ensure that if $\mathbf{w}_n(0)$ is close to $\overline{\mathbf{w}}_n$, then $\mathbf{w}_n(t)$ remains close to $\overline{\mathbf{w}}_n$ for a sufficiently long time.

However, if $\mathcal{H}_n(\mathbf{x}; \mathbf{w}_n)$ does not have locally Lipschitz gradients, then $\mathbf{0} \in \partial_{\mathbf{w}_n} L(\overline{\mathbf{w}}_n, 0)$ but $\partial_{\mathbf{w}_n} L(\mathbf{w}_n, 0)$ may be large near $\overline{\mathbf{w}}_n$. This prevents us from ensuring $\mathbf{w}_n(t)$ remains near $\overline{\mathbf{w}}_n$. For example consider $g(\mathbf{u}) = (u_1|u_2| - 1)^2$ and let $\tilde{\mathbf{u}} = [1, 0]^T$. Then, $\mathbf{0} \in \partial g(\tilde{\mathbf{u}})$. Let $\mathbf{u}^\delta = [1 + \delta, \delta]^T$. Then, for any $\delta \in (0, 0.1)$ and $\mathbf{s} \in -\partial g(\mathbf{u}^\delta)$, $\|\mathbf{s}\|_2 \geq 1$, and no matter how close $\mathbf{u}^\delta$ is to $\tilde{\mathbf{u}}$, if initialized at $\mathbf{u}^\delta$, the gradient flow will quickly get away from $\tilde{\mathbf{u}}$ (see the Appendix I for details).

Hence, we require eq. (15) which implies that $\partial_{\mathbf{w}_n} L(\mathbf{w}_n, \mathbf{0})$ varies smoothly along $\overline{\mathbf{w}}_n - \mathbf{w}_n$. This is sufficient for us to ensure $\mathbf{w}_n(t)$ remains near $\overline{\mathbf{w}}_n$. In fact, if $\mathcal{H}_n(\mathbf{x}; \mathbf{w}_n)$ have a locally Lipschitz gradient, then eq. (15) is automatically satisfied, since

$$\langle \overline{\mathbf{w}}_n - \mathbf{w}_n, \mathbf{s} \rangle \geq -\|\overline{\mathbf{w}}_n - \mathbf{w}_n\|_2 \|\nabla_{\mathbf{w}_n} L(\mathbf{w}_n, 0)\|_2$$
$$= -\|\overline{\mathbf{w}}_n - \mathbf{w}_n\|_2 \|\nabla_{\mathbf{w}_n} L(\mathbf{w}_n, 0) - \nabla_{\mathbf{w}_n} L(\overline{\mathbf{w}}_n, 0)\|_2 \geq -\kappa \|\overline{\mathbf{w}}_n - \mathbf{w}_n\|_2^2,$$

where the equality holds since $\nabla_{\mathbf{w}_n} L(\overline{\mathbf{w}}_n, 0) = \mathbf{0}$ from Lemma 5.6. The last inequality follows since $\mathcal{H}_n$ and the loss function have locally Lipschitz gradient. We also note that if the gradient of $\mathcal{H}_n(\mathbf{x}; \mathbf{w}_n)$ is not locally Lipschitz globally, but is locally Lipschitz in the neighborhood of $\overline{\mathbf{w}}_n$, then eq. (15) is also satisfied. For example again consider $g(\mathbf{u}) = (u_1|u_2| - 1)^2$ and let $\tilde{\mathbf{u}} = [1, 1]^T$. Then, it is easy to show that $\mathbf{0} \in \partial g(\tilde{\mathbf{u}})$, and $g(\mathbf{u})$ has locally Lipschitz gradient around $\tilde{\mathbf{u}}$.

### 5.3 Higher Orders of Homogeneity

Given the results of Theorem 5.1 and Theorem 5.5, it is natural to ask whether neural networks with higher orders of homogeneity also exhibit similar behavior. Presently, we are unsure if such a behavior is possible. The main difficulty is due to the relative scaling between the weights and the gradient at small initialization. From our discussion in Section 5.1.1, we know that near small initialization, the evolution of $\mathbf{w}(t)$ can approximately be expressed as

$$\dot{\mathbf{w}} \approx -\sum_{i=1}^{n} \nabla_{\hat{y}} \ell(0, y_i) \partial \mathcal{H}(\mathbf{x}_i; \mathbf{w}).$$

Now, from Lemma B.1, we know that if $\mathcal{H}(\mathbf{x}; \mathbf{w})$ is $L$-homogeneous, then $\partial \mathcal{H}(\mathbf{x}; \mathbf{w})$ is $(L-1)$-homogeneous. Thus, if $\|\mathbf{w}(0)\|_2$ scales as $\delta$, then $\|\partial \mathcal{H}(\mathbf{x}; \mathbf{w}(0))\|_2$ is expected to scale as $\delta^{L-1}$. For $L = 2$, $\|\mathbf{w}(0)\|_2$ and $\|\partial \mathcal{H}(\mathbf{x}; \mathbf{w}(0))\|_2$ both scale as $\delta$. Hence, for small $\delta$, in the initial stages of training, the gradient flow will not change the norm of the weights significantly, but can have significant impact on its direction. However, for $L > 2$ and small $\delta$, the gradient is smaller than the weights, where the relative scaling between the two

further worsens upon increasing $L$. Thus, it seems that, in the initials stages of training, the gradient will not have much impact on the direction and magnitude of the weights. Nonetheless, it is possible that in the slightly later stages of training the gradient can be large enough to change the direction of weights. However, it is unclear to us how to rigorously analyze such scenarios and is therefore a subject of future research.

## 6    Conclusions and Future Directions

In this work, we studied the gradient flow dynamics of two-homogeneous neural networks near small initializations and saddle points, and showed the approximate directional convergence of their weights in the initial stages of training.

An important future direction is, of course, to study the entire gradient flow dynamics of neural networks, and our work could be an important step towards a comprehensive understanding of training dynamics of neural networks. Particularly, for successful training under small initialization, the gradient dynamics will have to eventually escape the origin. The escape direction may be determined by the directions to which the weights converge while the dynamics is near the origin. This also holds true while escaping other saddle points encountered by gradient dynamics during the training process, which notably is known to undergo saddle-to-saddle dynamics.

Another possible future direction is to investigate similar directional convergence in deeper neural networks. For non-smooth neural networks, we required an additional assumption on the initialization to ensure uniqueness. It would be interesting to analyze scenarios when such assumptions do not hold. We defer that to a future investigation.

### Acknowledgments

The authors graciously acknowledge gift funding from InterDigital, which partially supported this work.

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

# Appendices

## A  Key Properties of o-minimal Structures and Clarke Subdifferentials

In this section we give a brief overview of o-minimal structures, and the relevant properties of Clarke subdifferentials that are used in the proofs of our results. We borrow much of the discussion below from Ji & Telgarsky (2020); Davis et al. (2018)

An o-minimal structure is a collection $\mathcal{S} = \{\mathcal{S}_n\}_{n=1}^{\infty}$, where each $\mathcal{S}_n$ is set of subsets of $\mathbb{R}^n$, that satisfies following axioms:

- The elements of $\mathcal{S}_1$ are finite unions of points and intervals.

- All algebraic subsets of $\mathbb{R}^n$ are in $\mathcal{S}_n$.

- For all $n$, $\mathcal{S}_n$ is a Boolean subalgebra of the power set of $\mathbb{R}^n$.

- If $\mathbf{A} \in \mathcal{S}_n$, $\mathbf{B} \in \mathcal{S}_m$, then $\mathbf{A} \times \mathbf{B} \in \mathcal{S}_{n+m}$

- If $\mathcal{P} : \mathbb{R}^{n+1} \to \mathbb{R}^n$ is the projection on first $n$ coordinates and $\mathbf{A} \in \mathcal{S}_{n+1}$, then $\mathcal{P}(\mathbf{A}) \in \mathcal{S}_n$.

For a given o-minimal structure $\mathcal{S}$, a set $\mathbf{A} \subset \mathbb{R}^n$ is definable if $\mathbf{A} \in \mathcal{S}_n$. A function $f : \mathbf{D} \to \mathbb{R}^m$ with $\mathbf{D} \subset \mathbb{R}^n$ is definable if the graph of $f$ is in $\mathcal{S}_{n+m}$. Since a set remains definable under projection, the domain $\mathbf{D}$ is also definable.

In Wilkie (1996), it was shown that there exists an o-minimal structure in which polynomials and exponential functions are definable. Also, the definability of a function is stable under algebraic operations, composition,

inverse, maximum, and minimum. Since ReLU and Leaky-ReLU can each be expressed as maximums of two polynomials, it can be shown that the functions we will consider in this paper are definable.

It is also true that deep neural networks with ReLU or Leaky-ReLU activation, and different kinds of layers, are definable. For completeness, we state that result here as a lemma.

**Lemma A.1.** *(Ji & Telgarsky, 2020, Lemma B.2) Suppose there exist $k, d_0, d_1, ..., d_L > 0$ and $L$ definable functions $(g_1, ..., g_L)$ where $g_j : \mathbb{R}^{d_0} \times \ldots \times \mathbb{R}^{d_{j-1}} \times \mathbb{R}^k \to \mathbb{R}^{d_j}$. Let $h_1(\mathbf{x}, \mathbf{w}) := g_1(\mathbf{x}, \mathbf{w})$, and for $2 \leq j \leq L$,*

$$h_j(\mathbf{x}, \mathbf{w}) := g_j(\mathbf{x}, h_1(\mathbf{x}, \mathbf{w}), ..., h_{j-1}(\mathbf{x}, \mathbf{w}), \mathbf{w})$$

*then all $h_j$ are definable. (It suffices if each output coordinate of $g_j$ is the minimum or maximum over some finite set of polynomials, which allows for linear, convolutional, ReLU, max-pooling layers and skip connections.)*

We also note that, since definability is stable under composition, the objective functions arising for definable losses are also definable.

## A.1 Chain Rules for Non-differentiable Functions

Recall that for any locally Lipschitz continuous function $f : X \to \mathbb{R}$, its Clarke subdifferential at a point $\mathbf{x} \in X$ is the set

$$\partial f(\mathbf{x}) = \text{conv} \left\{ \lim_{i \to \infty} \nabla f(\mathbf{x}_i) : \lim_{i \to \infty} \mathbf{x}_i = \mathbf{x}, \mathbf{x}_i \in \Omega \right\},$$

where $\Omega$ is any full-measure subset of $X$ such that $f$ is differentiable at each of its points, and $\overline{\partial} f(\mathbf{x})$ denotes the unique minimum norm subgradient.

The functions considered in this paper can be compositions of non-differentiable functions. We use Clarke's chain rule of differentiation, described in the following lemma, to compute the Clarke subdifferentials in such cases.

**Lemma A.2.** *(Clarke, 1983, Theorem 2.3.9 ) Let $h_1, \ldots, h_n : \mathbb{R}^d \to \mathbb{R}$ and $g : \mathbb{R}^n \to \mathbb{R}$ be locally Lipschitz functions, and $f(\mathbf{x}) = g(h_1(\mathbf{x}), \ldots, h_n(\mathbf{x}))$, then,*

$$\partial f(\mathbf{x}) \subseteq conv \left\{ \sum_{i=1}^n \alpha_i \zeta_i : \zeta_i \in \partial h_i(\mathbf{x}), \alpha \in \partial g(h_1(\mathbf{x}), \ldots, h_n(\mathbf{x})) \right\}.$$

The chain rule for gradient flow described in the next lemma, which is crucial for our analysis, essentially implies that for differential inclusions $\overline{\partial} f(\mathbf{x})$ plays the same role as $\nabla f(\mathbf{x})$ does for differential equations.

**Lemma A.3.** *(Davis et al., 2018, Lemma 5.2)(Ji & Telgarsky, 2020, Lemma B.9) Given a locally Lipschitz definable function $f : \mathbf{D} \to \mathbb{R}$ with an open domain $\mathbf{D}$, for any interval $I$ and any arc $\mathbf{x} : I \to \mathbf{D}$, it holds for a.e. $t \in I$ that*

$$\frac{d(f(\mathbf{x}(t)))}{dt} = \langle \mathbf{x}^*(t), \dot{\mathbf{x}}(t) \rangle, \text{ for all } \mathbf{x}^*(t) \in \partial f(\mathbf{x}(t)).$$

*Further, if $\mathbf{x} : I \to \mathbf{D}$ satisfies*

$$\dot{\mathbf{x}} \in \partial f(\mathbf{x}), \text{ for a.e. } t \geq 0,$$

*then, it holds for a.e. $t \geq 0$ that*

$$\dot{\mathbf{x}}(t) = \overline{\partial} f(\mathbf{x}(t)), \text{ and } \frac{d(f(\mathbf{x}(t)))}{dt} = \|\overline{\partial} f(\mathbf{x}(t))\|_2^2,$$

*and therefore,*

$$f(\mathbf{x}(t)) - f(\mathbf{x}(0)) = \int_0^t \|\overline{\partial} f(\mathbf{x}(s))\|_2^2 ds, \forall t \geq 0.$$

### A.2 The Kurdyka-Lojasiewicz Inequality

For gradient flow trajectories that are bounded, the *Kurdyka-Lojasiewicz Inequality* is useful for showing convergence, essentially by establishing the existence of a *desingularizing* function, which is formally defined as follows.

**Definition A.1.** A function $\Psi : [0, \nu) \to \mathbb{R}$ is called a *desingularizing function* when $\Psi$ is continuous on $[0, \nu)$ with $\Psi(0) = 0$, and it is continuously differentiable on $(0, \nu)$ with $\Psi' > 0$.

The following lemma, which can be seen as an unbounded version of the Kurdyka-Lojasiewicz Inequality, plays an important role in establishing the directional convergence of gradient flow trajectories of the neural correlation function where the trajectories can be unbounded.

**Lemma A.4.** *(Ji & Telgarsky, 2020, Lemma 3.6) Given a locally Lipschitz definable function $f$ with an open domain $\mathbf{D} \subset \{\mathbf{x} | \|\mathbf{x}\|_2 > 1\}$, for any $c, \eta > 0$, there exists a $\nu > 0$ and a definable desingularizing function $\Psi$ on $[0, \nu)$ such that*

$$\Psi'(f(\mathbf{x}))\|\mathbf{x}\|_2\|\overline{\partial}f(\mathbf{x})\|_2 \geq 1, \ \ if \ f(\mathbf{x}) \in (0, \nu) \ and \ \|\overline{\partial}_\perp f(\mathbf{x})\|_2 \geq c\|\mathbf{x}\|_2^\eta\|\overline{\partial}_r f(\mathbf{x})\|_2,$$

*where $\overline{\partial}_r f(\mathbf{x}) = \left\langle \overline{\partial} f(\mathbf{x}), \frac{\mathbf{x}}{\|\mathbf{x}\|_2} \right\rangle \frac{\mathbf{x}}{\|\mathbf{x}\|_2}$ and $\overline{\partial}_\perp f(\mathbf{x}) = \overline{\partial} f(\mathbf{x}) - \overline{\partial}_r f(\mathbf{x})$.*

## B Additional Notation and Some Preliminary Lemmata

For notational convenience when dealing with Clarke subdifferentials, we introduce the following notation for sets containing vectors.

- $\forall A, B \subseteq \mathbb{R}^d, A \pm B := \{\mathbf{x} \pm \mathbf{y} : \mathbf{x} \in A, \mathbf{y} \in B\}$

- $\forall B \subseteq \mathbb{R}^d$, and $c \in \mathbb{R}, cB := \{c\mathbf{y} : \mathbf{y} \in B\}$

- $\forall B \subseteq \mathbb{R}^d$, and $\mathbf{p} \in \mathbb{R}^d, \langle \mathbf{p}, B \rangle := \{\mathbf{p}^\top \mathbf{y} : \mathbf{y} \in B\} \subseteq \mathbb{R}$

- For any norm $\|\cdot\|$ on $\mathbb{R}^d, \|\mathbf{B}\| := \{\|\mathbf{y}\|, \mathbf{y} \in B\} \subseteq \mathbb{R}$

The following lemma states two important properties of homogeneous functions.

**Lemma B.1.** *((Lyu & Li, 2020, Theorem B.2), (Ji & Telgarsky, 2020, Lemma C.1)) Let $F : \mathbb{R}^k \to \mathbb{R}$ be a locally Lipschitz and $L-$positively homogeneous for some $L > 0$, then:*

*1. For any $\mathbf{w} \in \mathbb{R}^k$ and $c \geq 0$,*
$$\partial F(c\mathbf{w}) = c^{L-1}\partial F(\mathbf{w}).$$

*2. For any $\mathbf{w} \in \mathbb{R}^k$,*
$$\mathbf{w}^\top \mathbf{s} = LF(\mathbf{w}), \ for \ all \ \mathbf{s} \in \partial F(\mathbf{w}).$$

This result gives rise to the following corollary, which we use frequently in our analysis.

**Corollary B.1.1.** *For any $\mathbf{w} \in \mathbb{R}^k$ and $c \geq 0$,*

$$\mathbf{w}^\top \mathbf{s} = 2\mathcal{H}(\mathbf{x}; \mathbf{w}), \ for \ all \ \mathbf{s} \in \partial\mathcal{H}(\mathbf{x}; \mathbf{w}),$$

*and*

$$\partial\mathcal{H}(\mathbf{x}; c\mathbf{w}) = c\partial\mathcal{H}(\mathbf{x}; \mathbf{w}).$$

## C Proofs Omitted from Section 5.1

In this section we first prove Lemma 5.2 and Lemma 5.3, and then use them to ultimately prove Theorem 5.1.

### C.1 Proof of Lemma 5.2

To prove Lemma 5.2, we make use of the following lemma. It shows that if two differential inclusions are initialized at the same point, and the difference between them is small in a bounded interval, then the difference between their solutions is also small. This is a well-known stability result (Filippov, 1988, Theorem 1, Section 8); we provide a proof in Appendix J for completeness.

**Lemma C.1.** *Consider the following differential inclusions for $t \in [0, T]$:*

$$\frac{d\tilde{\mathbf{u}}}{dt} \in \sum_{i=1}^{n} z_i \partial \mathcal{H}(\mathbf{x}_i; \tilde{\mathbf{u}}), \tilde{\mathbf{u}}(0) = \mathbf{u}_0, \tag{19}$$

*and*

$$\frac{d\mathbf{u}}{dt} \in \sum_{i=1}^{n} (z_i + f_i(t)) \partial \mathcal{H}(\mathbf{x}_i; \mathbf{u}), \mathbf{u}(0) = \mathbf{u}_0, \tag{20}$$

*where $T$ is finite, and $|f_i(t)|_2 \leq \delta$ for all $i \in [n]$ and $t \in [0, T]$. Then, for any $\epsilon > 0$ there exists a $\delta > 0$, such that for each solution $\mathbf{u}(t)$ of eq. (20) there exists a solution $\tilde{\mathbf{u}}(t)$ of eq. (19) satisfying*

$$\max_{t \in [0, T]} \|\mathbf{u}(t) - \tilde{\mathbf{u}}(t)\|_2 \leq \epsilon. \tag{21}$$

*Proof of Lemma 5.2.* Using the chain rule from Lemma A.2, the gradient flow dynamics are

$$\dot{\mathbf{w}} \in -\sum_{i=1}^{n} \nabla_{\hat{y}} \ell \left( \mathcal{H}(\mathbf{x}_i; \mathbf{w}), y_i \right) \partial \mathcal{H}(\mathbf{x}_i; \mathbf{w}), \mathbf{w}(0) = \delta \mathbf{w}_0. \tag{22}$$

Now, we define $z(t) = \|\mathbf{w}(t)\|_2^2$ and note that $z(0) = \delta^2 \leq 1/C < 1$. Since $\mathbf{w}(t)$ is a continuous function, so is $z(t)$. Hence, there exists some $\gamma > 0$, such that for all $t \in (0, \gamma)$, $z(t) < 1$. We define $\hat{T}$ to be the smallest $t > 0$ such that $z(\hat{T}) = 1$. It follows that for all $t \in [0, \hat{T}]$, $z(t) \leq 1$.

Now, if $z(t) \leq 1$ and since $\beta := \sup\{\|\mathcal{H}(\mathbf{X}; \mathbf{w})\|_2 : \mathbf{w} \in \mathcal{S}^{k-1}\}$, then $\|\mathcal{H}(\mathbf{X}; \mathbf{w}(t))\|_2 \leq \beta \|\mathbf{w}(t)\|_2^2 \leq \beta$, for all $t \in [0, \hat{T}]$. From the definition of $\hat{\beta}$ in eq. (4), we get

$$\|\ell'(\mathcal{H}(\mathbf{X}; \mathbf{w}(t)), \mathbf{y})\|_2 \leq \hat{\beta} \|\mathcal{H}(\mathbf{X}; \mathbf{w}(t))\|_2 + \|\ell'(\mathbf{0}, \mathbf{y})\|_2 \leq \hat{\beta}\beta + \|\ell'(\mathbf{0}, \mathbf{y})\|_2 = \tilde{\beta}.$$

From the above equation and using Corollary B.1.1, we have

$$\dot{z} = 2\mathbf{w}^\top \dot{\mathbf{w}} = -4 \sum_{i=1}^{n} \nabla_{\hat{y}} \ell \left( \mathcal{H}(\mathbf{x}_i; \mathbf{w}), y_i \right) \mathcal{H}(\mathbf{x}_i; \mathbf{w}) = -4\mathcal{H}(\mathbf{X}; \mathbf{w})^\top \ell'(\mathcal{H}(\mathbf{X}; \mathbf{w}(t)) \leq 4\beta \|\mathbf{w}\|_2^2 \tilde{\beta} = 4\beta \tilde{\beta} z, \tag{23}$$

and so $z(t) \leq \delta^2 e^{4\beta \tilde{\beta} t}$, which implies

$$\hat{T} \geq \frac{1}{4\beta \tilde{\beta}} \ln \left( \frac{1}{\delta^2} \right).$$

Further, since $\delta \leq \frac{1}{\sqrt{C}}$, we have that $\frac{1}{4\beta \tilde{\beta}} \ln (C) \leq \hat{T}$ implies

$$z(t) \leq C\delta^2, \forall t \in \left[ 0, \frac{\ln (C)}{4\beta \tilde{\beta}} \right]. \tag{24}$$

Now, we consider $t \in \left[ 0, \frac{\ln(C)}{4\beta \tilde{\beta}} \right]$. Note that

$$\|\mathcal{H}(\mathbf{X}; \mathbf{w}(t))\|_2 \leq \beta \|\mathbf{w}(t)\|_2^2 \leq \beta C\delta^2. \tag{25}$$

Define $\xi(t) := \ell'(\mathcal{H}(\mathbf{X}; \mathbf{w}(t)), \mathbf{y}) - \ell'(\mathbf{0}, \mathbf{y})$; then,

$$\|\xi(t)\|_2 = \|\ell'(\mathcal{H}(\mathbf{X}; \mathbf{w}(t)), \mathbf{y}) - \ell'(\mathbf{0}, \mathbf{y})\|_2 \leq \hat{\beta}\|\mathcal{H}(\mathbf{X}; \mathbf{w}(t)\|_2 \leq \hat{\beta}\beta C\delta^2.$$

Next, the dynamics of $\mathbf{w}(t)$ can be written as

$$\dot{\mathbf{w}} \in -\sum_{i=1}^{n} \nabla_{\hat{y}} \ell\left(\mathcal{H}(\mathbf{x}_i; \mathbf{w}), y_i\right) \partial \mathcal{H}(\mathbf{x}_i; \mathbf{w}) = \sum_{i=1}^{n} (-\nabla_{\hat{y}} \ell\left(0, y_i\right) - \xi_i(t)) \partial \mathcal{H}(\mathbf{x}_i; \mathbf{w}). \tag{26}$$

Dividing eq. (26) by $\delta$, from 1-homogeneity of $\partial \mathcal{H}(\mathbf{x}; \mathbf{w})$ (corollary B.1.1) we have

$$\frac{\dot{\mathbf{w}}}{\delta} \in \frac{1}{\delta} \sum_{i=1}^{n} (-\nabla_{\hat{y}} \ell\left(0, y_i\right) - \xi_i(t)) \partial \mathcal{H}(\mathbf{x}_i; \mathbf{w}) = \sum_{i=1}^{n} (-\nabla_{\hat{y}} \ell\left(0, y_i\right) - \xi_i(t)) \partial \mathcal{H}(\mathbf{x}_i; \mathbf{w}/\delta). \tag{27}$$

Now, consider the differential inclusion

$$\frac{d\tilde{\mathbf{w}}}{dt} \in -\sum_{i=1}^{n} \nabla_{\hat{y}} \ell\left(0, y_i\right) \partial \mathcal{H}(\mathbf{x}_i; \tilde{\mathbf{w}}), \tilde{\mathbf{w}}(0) = \mathbf{w}_0. \tag{28}$$

Using the fact that for all $t \in \left[0, \frac{\ln(C)}{4\beta\hat{\beta}}\right]$, $\|\xi(t)\|_2 \leq \delta^2 \hat{\beta}\beta C$, and using Lemma C.1, we have that there exists a small enough $\overline{\delta}$ such that for all $\delta \leq \overline{\delta}$,

$$\left\|\tilde{\mathbf{w}}(t) - \frac{\mathbf{w}(t)}{\delta}\right\|_2 \leq \epsilon,$$

where $\tilde{\mathbf{w}}(t)$ is a solution of eq. (28). $\qquad\square$

## C.2   Proof of Lemma 5.3

Recall that $\mathcal{N}_{\mathbf{z},\mathcal{H}}(\mathbf{u}) = \mathbf{z}^\top \mathcal{H}\left(\mathbf{X}; \mathbf{u}\right)$. Throughout this section, for the sake of brevity, we will use $\mathcal{N}(\mathbf{u})$ instead of $\mathcal{N}_{\mathbf{z},\mathcal{H}}(\mathbf{u})$. The gradient flow $\mathbf{u}(t)$ satisfies, for a.e. $t \geq 0$,

$$\frac{d\mathbf{u}}{dt} \in \partial\mathcal{N}(\mathbf{u}) \subseteq \sum_{i=1}^{n} z_i \partial \mathcal{H}(\mathbf{x}_i; \mathbf{u}), \mathbf{u}(0) = \mathbf{u}_0. \tag{29}$$

We will first prove some auxiliary lemmata. The first follows simply from two-homogeneity of $\mathcal{N}(\mathbf{u})$ and Lemma B.1.

**Lemma C.2.** *For any $s \in \partial\mathcal{N}(\mathbf{u})$, $\mathbf{s}^\top \mathbf{u} = 2\mathcal{N}(\mathbf{u})$. For any $c \geq 0$, $\partial\mathcal{N}(c\mathbf{u}) = c\partial\mathcal{N}(\mathbf{u})$*

The next lemma states that if the initialization is non-zero, then gradient flow stays away from the origin for all finite time.

**Lemma C.3.** *Suppose $\mathbf{u}(t)$ is a solution of eq. (29), where $\mathbf{u}_0$ is a non-zero vector. Then, for all finite $t > 0$, $\|\mathbf{u}(t)\|_2 > 0$.*

*Proof.* Since $\|\mathbf{u}(0)\|_2 > 0$, from continuity of $\mathbf{u}(t)$, there exists some $\gamma > 0$ such that $\|\mathbf{u}(t)\|_2 > 0$, for all $t \in (0, \gamma)$. For the sake of contradiction, suppose there exists some finite $T > 0$ such that $\|\mathbf{u}(T)\|_2 = 0$ for the first time. Then, for all $t \in [0, T)$, $\|\mathbf{u}(t)\|_2 > 0$. Since for a.e. $t \in [0, T)$

$$\frac{d\log(\|\mathbf{u}\|_2^2)}{dt} = \frac{1}{\|\mathbf{u}\|_2^2} \frac{d\|\mathbf{u}\|_2^2}{dt} = \frac{4\mathbf{z}^\top \mathcal{H}(\mathbf{X}; \mathbf{u})}{\|\mathbf{u}\|_2^2} \geq -4\beta\|\mathbf{z}\|_2,$$

it follows that for all $t \in (0, T)$,

$$\|\mathbf{u}(t)\|_2^2 \geq \|\mathbf{u}_0\|_2^2 e^{-4t\beta\|\mathbf{z}\|_2}.$$

Taking $t \to T$, we have $\|\mathbf{u}(T)\|_2^2 \geq \|\mathbf{u}_0\|_2^2 e^{-4T\beta\|\mathbf{z}\|_2} > 0$ which leads to a contradiction. $\qquad\square$

**Lemma C.4.** *If $\mathcal{N}(\mathbf{u}(t_0)) \geq 0$, for any $t_0 \geq 0$, then, $\mathcal{N}(\mathbf{u}(t)) \geq 0$ and $\|\mathbf{u}(t)\|_2 \geq \|\mathbf{u}(t_0)\|_2$, for all $t \geq t_0$.*

*Proof.* Since, using Lemma A.3,

$$\mathcal{N}(\mathbf{u}(t)) - \mathcal{N}(\mathbf{u}(t_0)) = \int_{t_0}^t \|\dot{\mathbf{u}}(s)\|_2^2 ds,$$

we have that for $t \geq t_0$, $\mathcal{N}(\mathbf{u}(t)) \geq \mathcal{N}(\mathbf{u}(t_0)) \geq 0$. The second claim is true since for a.e. $t \geq 0$, $\frac{d\|\mathbf{u}\|_2^2}{dt} = 4\mathcal{N}(\mathbf{u})$ implies

$$\|\mathbf{u}(t)\|_2^2 - \|\mathbf{u}(t_0)\|_2^2 = 4 \int_{t_0}^t \mathcal{N}(\mathbf{u}(s)) ds \geq 0.$$

$\square$

The following lemma states the conditions for first-order KKT point of the constrained NCF.

**Lemma C.5.** *If a vector $\mathbf{u}^* \in \mathbb{R}^{k \times 1}$ is a first-order KKT point of*

$$\max_{\|\mathbf{u}\|_2^2 = 1} \mathcal{N}(\mathbf{u}) = \mathbf{z}^\top \mathcal{H}(\mathbf{X}; \mathbf{u}), \tag{30}$$

*then*

$$\sum_{i=1}^n z_i \partial \mathcal{H}(\mathbf{x}_i; \mathbf{u}^*) = \lambda^* \mathbf{u}^*, \|\mathbf{u}^*\|_2^2 = 1, \tag{31}$$

*where $\lambda^* \in \mathbb{R}$ is the Lagrange multiplier. Also, $2\mathcal{N}(\mathbf{u}^*) = \lambda^*$ and hence, for a non-negative KKT point $\lambda^* \geq 0$.*

*Proof.* The Lagrangian is equal to

$$L(\mathbf{u}, \lambda) = \mathcal{N}(\mathbf{u}) + \lambda(\|\mathbf{u}\|_2^2 - 1).$$

If $\mathbf{u}^*$ is a first-order KKT point then it must satisfy the constraint set and, for some $\overline{\lambda}$,

$$\mathbf{0} \in \partial \mathcal{N}(\mathbf{u}^*) + 2\overline{\lambda}\mathbf{u}^*,$$

implying

$$\mathbf{0} \in \sum_{i=1}^n z_i \partial \mathcal{H}(\mathbf{x}_i; \mathbf{u}^*) + 2\overline{\lambda}\mathbf{u}^*.$$

Choosing $\lambda^* = -2\overline{\lambda}$ we get eq. (31). By Lemma C.2,

$$\lambda^* = \lambda^* \|\mathbf{u}^*\|_2^2 = {\mathbf{u}^*}^\top \partial \mathcal{N}(\mathbf{u}^*) = 2\mathcal{N}(\mathbf{u}^*)$$

$\square$

In the following lemma we define $\tilde{\mathcal{N}}(\mathbf{u})$, which is central to our proof, and its minimum norm Clarke subdifferential.

**Lemma C.6.** *For any nonzero $\mathbf{u} \in \mathbb{R}^k$ we define $\tilde{\mathcal{N}}(\mathbf{u}) = \mathcal{N}(\mathbf{u})/\|\mathbf{u}\|_2^2$, then,*

$$\partial \tilde{\mathcal{N}}(\mathbf{u}) = \left\{ \frac{\mathbf{s}}{\|\mathbf{u}\|_2^2} - \frac{2\mathcal{N}(\mathbf{u})\mathbf{u}}{\|\mathbf{u}\|_2^4} \middle| \mathbf{s} \in \partial \mathcal{N}(\mathbf{u}) \right\} = \left\{ \left( \mathbf{I} - \frac{\mathbf{u}\mathbf{u}^\top}{\|\mathbf{u}\|_2^2} \right) \frac{\mathbf{s}}{\|\mathbf{u}\|_2^2} \middle| \mathbf{s} \in \partial \mathcal{N}(\mathbf{u}) \right\},$$

*and*

$$\overline{\partial} \tilde{\mathcal{N}}(\mathbf{u}) = \left( \mathbf{I} - \frac{\mathbf{u}\mathbf{u}^\top}{\|\mathbf{u}\|_2^2} \right) \frac{\overline{\partial} \mathcal{N}(\mathbf{u})}{\|\mathbf{u}\|_2^2}.$$

*Proof.* First, note that $\tilde{\mathcal{N}}(\mathbf{u})$ is differentiable if and only if $\mathcal{N}(\mathbf{u})$ is differentiable. Therefore, for any non-zero $\mathbf{u}$ such that $\mathcal{N}(\mathbf{u})$ is differentiable,

$$\nabla\tilde{\mathcal{N}}(\mathbf{u}) = \frac{\nabla\mathcal{N}(\mathbf{u})}{\|\mathbf{u}\|_2^2} - \frac{2\mathcal{N}(\mathbf{u})\mathbf{u}}{\|\mathbf{u}\|_2^4}.$$

The first claim follows from the definition of Clarke subdifferential and Lemma C.2. For the second claim, note that

$$\left\|\frac{\mathbf{s}}{\|\mathbf{u}\|_2^2} - \frac{2\mathcal{N}(\mathbf{u})\mathbf{u}}{\|\mathbf{u}\|_2^4}\right\|_2^2 = \frac{\|\mathbf{s}\|_2^2}{\|\mathbf{u}\|_2^4} - \frac{4\mathcal{N}(\mathbf{u})^2}{\|\mathbf{u}\|_2^6} + \frac{4\mathcal{N}(\mathbf{u})^2}{\|\mathbf{u}\|_2^8},$$

where we use $\mathbf{s}^\top\mathbf{u} = 2\mathcal{N}(\mathbf{u})$, for all $\mathbf{s} \in \partial\mathcal{N}(\mathbf{u})$. Hence, for the minimum norm subdifferential of $\tilde{\mathcal{N}}(\mathbf{u})$, we must choose the minimum norm subdifferential of $\mathcal{N}(\mathbf{u})$. $\qquad\square$

We now proceed to proving Lemma 5.3. We begin by showing that either $\mathbf{u}(t)$ converges to $\mathbf{0}$ or $\lim_{t\to\infty}\mathbf{u}(t)/\|\mathbf{u}(t)\|_2$ exists. We consider two cases.

**Case 1:** $\mathcal{N}(\mathbf{u}(0)) > 0$.
In this case, we show that $\lim_{t\to\infty}\mathbf{u}(t)/\|\mathbf{u}(t)\|_2$ exists using a similar technique as in Ji & Telgarsky (2020). Specifically, we show that the length of the curve swept by $\mathbf{u}(t)/\|\mathbf{u}(t)\|_2$, which is defined as

$$\int_0^\infty \left\|\frac{d}{dt}\left(\frac{\mathbf{u}}{\|\mathbf{u}\|_2}\right)\right\|_2 dt,$$

has finite length, and thus $\lim_{t\to\infty}\mathbf{u}(t)/\|\mathbf{u}(t)\|_2$ exists.

We assume $\mathcal{N}(\mathbf{u}(0)) = \gamma > 0$, and thus $\|\mathbf{u}(0)\|_2 > 0$. From Lemma C.4, for all $t \geq 0$,

$$\mathcal{N}(\mathbf{u}(t)) \geq \mathcal{N}(\mathbf{u}(0)) = \gamma,$$

implying

$$\frac{1}{2}\frac{d\|\mathbf{u}\|_2^2}{dt} = \mathbf{u}^\top\dot{\mathbf{u}} = 2\mathcal{N}(\mathbf{u}(t)) \geq 2\gamma, \text{ for a.e. } t \geq 0,$$

which in turn implies

$$\|\mathbf{u}(t)\|_2^2 \geq \|\mathbf{u}(0)\|_2^2 + 4\gamma t, \text{ for all } t \geq 0. \tag{32}$$

Recall that $\tilde{\mathcal{N}}(\mathbf{u}) = \mathcal{N}(\mathbf{u})/\|\mathbf{u}\|_2^2$. Since $\|\mathbf{u}(t)\|_2 > 0$, $\tilde{\mathcal{N}}(\mathbf{u}(t))$ is defined for all $t \geq 0$. Also, by Lemma A.3, for a.e. $t \geq 0$,

$$\frac{d\mathcal{N}(\mathbf{u})}{dt} = \|\dot{\mathbf{u}}\|_2^2, \text{ and } \dot{\mathbf{u}} = \overline{\partial}\mathcal{N}(\mathbf{u}).$$

Therefore, using the chain rule from Lemma A.3, for a.e. $t \geq 0$ we have

$$\frac{d\tilde{\mathcal{N}}(\mathbf{u})}{dt} = \dot{\mathbf{u}}^\top\left(\mathbf{I} - \frac{\mathbf{u}\mathbf{u}^T}{\|\mathbf{u}\|_2^2}\right)\frac{\overline{\partial}\mathcal{N}(\mathbf{u})}{\|\mathbf{u}\|_2^2} = \frac{\overline{\partial}\mathcal{N}(\mathbf{u})^\top}{\|\mathbf{u}\|_2^2}\left(\mathbf{I} - \frac{\mathbf{u}\mathbf{u}^T}{\|\mathbf{u}\|_2^2}\right)\overline{\partial}\mathcal{N}(\mathbf{u}) \geq 0, \tag{33}$$

where in the second equality we used that $\dot{\mathbf{u}} = \overline{\partial}\mathcal{N}(\mathbf{u})$, for a.e. $t \geq 0$. Hence, for all $t_2 \geq t_1 \geq 0$,

$$\tilde{\mathcal{N}}(\mathbf{u}(t_2)) - \tilde{\mathcal{N}}(\mathbf{u}(t_1)) = \int_{t_1}^{t_2}\frac{\overline{\partial}\mathcal{N}(\mathbf{u})^\top}{\|\mathbf{u}\|_2^2}\left(\mathbf{I} - \frac{\mathbf{u}\mathbf{u}^T}{\|\mathbf{u}\|_2^2}\right)^\top\overline{\partial}\mathcal{N}(\mathbf{u})dt \geq 0.$$

Therefore, $\tilde{\mathcal{N}}(\mathbf{u}(t)))$ is an increasing function, and hence, for any $t \geq 0$, that $\tilde{\mathcal{N}}(\mathbf{u}(t)) \geq \tilde{\mathcal{N}}(\mathbf{u}(0))$ implies

$$\mathcal{N}(\mathbf{u}(t)) \geq \tilde{\mathcal{N}}(\mathbf{u}(0))\|\mathbf{u}(t)\|_2^2.$$

From the above inequality and eq. (32), we have $\lim_{t\to\infty} \mathcal{N}(\mathbf{u}(t)) = \infty$.

Now, since $\mathcal{N}(\mathbf{u}) \leq \|\mathbf{z}\|_2 \|\mathcal{H}(\mathbf{X};\mathbf{u})\|_2 \leq \beta \|\mathbf{z}\|_2 \|\mathbf{u}\|_2^2$, we have that $\tilde{\mathcal{N}}(\mathbf{u})$ is bounded. Hence, by monotone convergence theorem, $\lim_{t\to\infty} \tilde{\mathcal{N}}(\mathbf{u}(t))$ exists; here, we suppose it is equal to $\overline{f}$.

Note that, by the chain rule, for a.e. $t \geq 0$,

$$\frac{d}{dt}\left(\frac{\mathbf{u}}{\|\mathbf{u}\|_2}\right) = \left(\mathbf{I} - \frac{\mathbf{u}\mathbf{u}^T}{\|\mathbf{u}\|_2^2}\right)\frac{\dot{\mathbf{u}}}{\|\mathbf{u}\|_2} = \left(\mathbf{I} - \frac{\mathbf{u}\mathbf{u}^T}{\|\mathbf{u}\|_2^2}\right)\frac{\overline{\partial}\mathcal{N}(\mathbf{u})}{\|\mathbf{u}\|_2}$$

implies

$$\left\|\frac{d}{dt}\left(\frac{\mathbf{u}}{\|\mathbf{u}\|_2}\right)\right\|_2 = \left\|\left(\mathbf{I} - \frac{\mathbf{u}\mathbf{u}^T}{\|\mathbf{u}\|_2^2}\right)\overline{\partial}\mathcal{N}(\mathbf{u})\right\|_2 \frac{1}{\|\mathbf{u}\|_2}. \tag{34}$$

Suppose $\tilde{\mathcal{N}}(\mathbf{u}(t))$ converges to $\overline{f}$ in finite time, i.e., $\tilde{\mathcal{N}}(\mathbf{u}(T)) = \overline{f}$ for some finite $T$. Then, for a.e. $t \geq T$, $\frac{d\tilde{\mathcal{N}}(\mathbf{u})}{dt} = 0$ implies

$$\left\|\left(\mathbf{I} - \frac{\mathbf{u}(t)\mathbf{u}(t)^T}{\|\mathbf{u}(t)\|_2^2}\right)\overline{\partial}\mathcal{N}(\mathbf{u}(t))\right\|_2 = 0.$$

Therefore, from eq. (34), we have

$$\frac{d}{dt}\left(\frac{\mathbf{u}}{\|\mathbf{u}\|_2}\right) = \mathbf{0}, \text{ for a.e. } t \geq T,$$

and hence, $\lim_{t\to\infty}\frac{\mathbf{u}(t)}{\|\mathbf{u}(t)\|_2}$ exists and is equal to $\frac{\mathbf{u}(T)}{\|\mathbf{u}(T)\|_2}$.

Thus, we may assume $\overline{f} - \tilde{\mathcal{N}}(\mathbf{u}(t)) > 0$, for all finite $t$. Define $g(\mathbf{u}) = \overline{f} - \tilde{\mathcal{N}}(\mathbf{u})$. Then, since

$$\|\overline{\partial}_r \tilde{\mathcal{N}}(\mathbf{u})\|_2 = 0,$$

we have that

$$\|\overline{\partial}_\perp g(\mathbf{u})\|_2 \geq \|\mathbf{u}\|_2 \|\overline{\partial}_r g(\mathbf{u})\|_2 = 0.$$

Hence, from Theorem A.4, there exists a $\nu > 0$ and a desingularizing function $\Psi(\cdot)$ defined on $[0,\nu)$ such that if $\|\mathbf{u}\|_2 > 1$ and $g(\mathbf{u}) < \nu$, then

$$1 \leq \Psi'(g(\mathbf{u}))\|\mathbf{u}\|_2\|\overline{\partial}g(\mathbf{u})\|_2 = \Psi'(\overline{f} - \tilde{\mathcal{N}}(\mathbf{u}))\|\mathbf{u}\|_2\|\overline{\partial}\tilde{\mathcal{N}}(\mathbf{u})\|_2. \tag{35}$$

Since $\lim_{t\to\infty}\tilde{\mathcal{N}}(\mathbf{u}(t)) = \overline{f}$, and eq. (32) holds, we may choose $T$ large enough such that $\|\mathbf{u}(t)\|_2 > 1$, and $g(\mathbf{u}(t)) < \nu$, for all $t \geq T$. Hence, for a.e. $t \geq T$,

$$\begin{aligned}
\frac{d\tilde{\mathcal{N}}(\mathbf{u})}{dt} &= \frac{\overline{\partial}\mathcal{N}(\mathbf{u})^\top}{\|\mathbf{u}\|_2^2}\left(\mathbf{I} - \frac{\mathbf{u}\mathbf{u}^T}{\|\mathbf{u}\|_2^2}\right)^\top \overline{\partial}\mathcal{N}(\mathbf{u}) = \left\|\left(\mathbf{I} - \frac{\mathbf{u}\mathbf{u}^T}{\|\mathbf{u}\|_2^2}\right)\frac{\overline{\partial}\mathcal{N}(\mathbf{u})}{\|\mathbf{u}\|_2}\right\|_2^2 \\
&= \left\|\left(\mathbf{I} - \frac{\mathbf{u}\mathbf{u}^T}{\|\mathbf{u}\|_2^2}\right)\frac{\overline{\partial}\mathcal{N}(\mathbf{u})}{\|\mathbf{u}\|_2}\right\|_2 \left\|\frac{d}{dt}\left(\frac{\mathbf{u}}{\|\mathbf{u}\|_2}\right)\right\|_2 \\
&= \|\mathbf{u}\|_2 \|\overline{\partial}\tilde{\mathcal{N}}(\mathbf{u})\|_2 \left\|\frac{d}{dt}\left(\frac{\mathbf{u}}{\|\mathbf{u}\|_2}\right)\right\|_2 \\
&\geq \frac{1}{\Psi'(\overline{f} - \tilde{\mathcal{N}}(\mathbf{u}))}\left\|\frac{d}{dt}\left(\frac{\mathbf{u}}{\|\mathbf{u}\|_2}\right)\right\|_2
\end{aligned}$$

implying

$$\left\|\frac{d}{dt}\left(\frac{\mathbf{u}}{\|\mathbf{u}\|_2}\right)\right\|_2 \leq -\frac{d\Psi(\overline{f} - \tilde{\mathcal{N}}(\mathbf{u}))}{dt}.$$

In the chain of equalities and inequalities above, we used Lemma C.6 for the fourth equality, and for the first inequality we used eq. (35). Now, integrating both sides of the above from $T$ to any $t_1 \geq T$, we have

$$\int_T^{t_1} \left\| \frac{d}{dt}\left(\frac{\mathbf{u}}{\|\mathbf{u}\|_2}\right)\right\|_2 dt \leq \Psi(\overline{f} - \tilde{\mathcal{N}}(\mathbf{u}(T))) - \Psi(\overline{f} - \tilde{\mathcal{N}}(\mathbf{u}(t_1))) \leq \Psi(\overline{f} - \tilde{\mathcal{N}}(\mathbf{u}(T))) < \infty,$$

which implies

$$\int_0^\infty \left\| \frac{d}{dt}\left(\frac{\mathbf{u}}{\|\mathbf{u}\|_2}\right)\right\|_2 dt = \int_0^T \left\| \frac{d}{dt}\left(\frac{\mathbf{u}}{\|\mathbf{u}\|_2}\right)\right\|_2 dt + \int_T^\infty \left\| \frac{d}{dt}\left(\frac{\mathbf{u}}{\|\mathbf{u}\|_2}\right)\right\|_2 dt$$
$$\leq \int_0^T \left\| \frac{d}{dt}\left(\frac{\mathbf{u}}{\|\mathbf{u}\|_2}\right)\right\|_2 dt + \Psi(\overline{f} - \tilde{\mathcal{N}}(\mathbf{u}(T))) < \infty,$$

completing the proof.

**Case 2:** $\mathcal{N}(\mathbf{u}(0)) \leq 0$.
In this case, we may further assume that $\mathcal{N}(\mathbf{u}(t)) \leq 0$, for all $t \geq 0$, since if for some finite $\overline{t}$, $\mathcal{N}(\mathbf{u}(\overline{t})) > 0$, we can use the proof for Case 1 by choosing $\overline{t}$ as the starting time to prove the claim. Thus, we assume $\mathcal{N}(\mathbf{u}(t)) \leq 0$, for all $t \geq 0$. Now, since

$$\frac{1}{2}\frac{d\|\mathbf{u}\|_2^2}{dt} = \mathbf{u}^\top \dot{\mathbf{u}} = 2\mathcal{N}(\mathbf{u}(t)) \leq 0, \text{ for a.e. } t \geq 0,$$

it follows that $\|\mathbf{u}(t)\|_2$ decreases with time. Hence, $\lim_{t\to\infty} \|\mathbf{u}(t)\|_2$ exists. If $\lim_{t\to\infty} \|\mathbf{u}(t)\|_2 = 0$, then $\lim_{t\to\infty} \mathbf{u}(t) = \mathbf{0}$ and $\lim_{t\to\infty} \mathcal{N}(\mathbf{u}(t)) = 0$ and we are done.

Otherwise, assume that $\lim_{t\to\infty} \|\mathbf{u}(t)\|_2 = \eta > 0$. In this case, since $\|\mathbf{u}(t)\|_2$ is a decreasing function, $\|\mathbf{u}(t)\|_2 \geq \eta$, for all $t \geq 0$. Since by Lemma A.3, for a.e. $t \geq 0$,

$$\frac{d\mathcal{N}(\mathbf{u})}{dt} = \|\dot{\mathbf{u}}\|_2^2,$$

we have that $\mathcal{N}(\mathbf{u}(t))$ increases with time. But, we also assume $\mathcal{N}(\mathbf{u}(t)) \leq 0$, and so by the monotone convergence theorem, $\mathcal{N}(\mathbf{u}(t))$ converges. We further claim that $\lim_{t\to\infty} \mathcal{N}(\mathbf{u}(t)) = 0$. Suppose for the sake of contradiction $\lim_{t\to\infty} \mathcal{N}(\mathbf{u}(t)) = -\gamma < 0$. Since $\mathcal{N}(\mathbf{u}(t))$ increases with time, we have $\mathcal{N}(\mathbf{u}(t)) \leq -\gamma$, for all $t \geq 0$. Hence,

$$\frac{1}{2}\frac{d\|\mathbf{u}\|_2^2}{dt} = \mathbf{u}^\top \dot{\mathbf{u}} = 2\mathcal{N}(\mathbf{u}(t)) \leq -2\gamma, \text{ for a.e. } t \geq 0.$$

The above equation implies that $\|\mathbf{u}(t)\|_2$ will become less than $\eta$ within a finite time, which leads to a contradiction, and therefore, $\lim_{t\to\infty} \mathcal{N}(\mathbf{u}(t)) = 0$.

We next show that $\lim_{t\to\infty} \mathbf{u}(t)/\|\mathbf{u}(t)\|_2$ exists. We can do this in same way in the proof of Case 1. Define $\hat{\mathbf{u}}(t) = 2\mathbf{u}(t)/\eta$, and note that if $\hat{\mathbf{u}}(t)$ converges in direction, then $\mathbf{u}(t)$ also converges in direction. We make this transformation because to use Lemma A.4 we require $\|\mathbf{u}(t)\|_2 > 1$ after some time $T$. While $\mathbf{u}(t)$ may never exceed 1, we do have $\|\hat{\mathbf{u}}(t)\|_2 \geq 2 > 1$, for all $t \geq 0$.

Next, $\tilde{\mathcal{N}}(\mathbf{u}(t))$ is defined for all $t \geq 0$, since $\|\mathbf{u}(t)\|_2 > 0$ for all $t \geq 0$. Also, since $\mathcal{N}(\mathbf{u}(t))$ converges to 0, $\tilde{\mathcal{N}}(\mathbf{u}(t))$ also converges to 0. Also, $\tilde{\mathcal{N}}(\mathbf{u}(t)) = \tilde{\mathcal{N}}(\hat{\mathbf{u}}(t))$, thus $\tilde{\mathcal{N}}(\hat{\mathbf{u}}(t))$ converges to 0 as well. From here to prove directional convergence of $\hat{\mathbf{u}}(t)$ we can use the the same approach as in Case 1, specifically from eq. (34) onward.

We next turn towards showing that if $\lim_{t\to\infty} \frac{\mathbf{u}(t)}{\|\mathbf{u}(t)\|_2}$ exists, then the limit must be a non-negative KKT point of the constrained NCF. Suppose $\mathbf{u}^*$ is the limit. We have already shown that $\mathcal{N}(\mathbf{u}^*) = \tilde{\mathcal{N}}(\mathbf{u}^*) \geq 0$. Thus, we only need to prove that $\mathbf{u}^*$ is a KKT point, i.e., from eq. (31), it must satisfy

$$2\mathcal{N}(\mathbf{u}^*)\mathbf{u}^* \in \partial\mathcal{N}(\mathbf{u}^*), \tag{36}$$

Assume for the sake of contradiction that there exists some $\gamma > 0$ such that for all $\mathbf{s} \in \partial \mathcal{N}(\mathbf{u}^*)$, we have

$$\|\mathbf{s} - 2\mathcal{N}(\mathbf{u}^*)\mathbf{u}^*\|_2 \geq \gamma. \tag{37}$$

Define $\mathbf{u}_\epsilon = \{\mathbf{u} : \|\mathbf{u} - \mathbf{u}^*\| \leq \epsilon\}$. Given $\gamma$, by upper semi-continuity of the Clarke subdifferential, we may choose $\epsilon \in (0,1)$ sufficiently small such that for all $\mathbf{u} \in \mathbf{u}_\epsilon$, we have

$$\partial \mathcal{N}(\mathbf{u}) \subseteq \{\mathbf{p} : \mathbf{p} = \mathbf{q} + \mathbf{r}, \mathbf{q} \in \partial \mathcal{N}(\mathbf{u}^*), \|\mathbf{r}\|_2 \leq \gamma/4\}. \tag{38}$$

Since $\frac{\mathbf{u}(t)}{\|\mathbf{u}(t)\|_2}$ converges to $\mathbf{u}^*$, and $\mathcal{N}(\mathbf{u})$ is continuous, we can choose $T$ large enough such that for all $t \geq T$

$$\left\| \frac{\mathbf{u}(t)}{\|\mathbf{u}(t)\|_2} - \mathbf{u}^* \right\|_2 \leq \epsilon, \text{ and } \left\| \frac{2\mathbf{u}(t)}{\|\mathbf{u}(t)\|_2} \mathcal{N}\left(\frac{\mathbf{u}(t)}{\|\mathbf{u}(t)\|_2}\right) - 2\mathbf{u}^* \mathcal{N}(\mathbf{u}^*) \right\|_2 \leq \gamma/4. \tag{39}$$

Suppose $\mathbf{s} \in \partial \mathcal{N}(\mathbf{u}^*)$, then, for all $\mathbf{u} \in \mathbb{R}^k \backslash \{\mathbf{0}\}$ we have

$$
\begin{aligned}
\left\| \left( \mathbf{I} - \frac{\mathbf{u}\mathbf{u}^\top}{\|\mathbf{u}\|_2^2} \right) \frac{\overline{\partial}\mathcal{N}(\mathbf{u})}{\|\mathbf{u}\|_2} \right\|_2 &= \left\| \left( \frac{\overline{\partial}\mathcal{N}(\mathbf{u})}{\|\mathbf{u}\|_2} - \frac{2\mathbf{u}\mathcal{N}(\mathbf{u})}{\|\mathbf{u}\|_2^3} \right) \right\|_2 \\
&= \left\| \overline{\partial}\mathcal{N}\left( \frac{\mathbf{u}}{\|\mathbf{u}\|_2} \right) - \frac{2\mathbf{u}}{\|\mathbf{u}\|_2} \mathcal{N}\left( \frac{\mathbf{u}}{\|\mathbf{u}\|_2} \right) \right\|_2 \\
&\geq \left\| \mathbf{s} - \frac{2\mathbf{u}}{\|\mathbf{u}\|_2} \mathcal{N}\left( \frac{\mathbf{u}}{\|\mathbf{u}\|_2} \right) \right\|_2 - \left\| \overline{\partial}\mathcal{N}\left( \frac{\mathbf{u}}{\|\mathbf{u}\|_2} \right) - \mathbf{s} \right\|_2 \\
&\geq \|\mathbf{s} - 2\mathbf{u}^* \mathcal{N}(\mathbf{u}^*)\|_2 - \left\| \overline{\partial}\mathcal{N}\left( \frac{\mathbf{u}}{\|\mathbf{u}\|_2} \right) - \mathbf{s} \right\|_2 - \left\| 2\mathbf{u}^* \mathcal{N}(\mathbf{u}^*) - \frac{2\mathbf{u}}{\|\mathbf{u}\|_2} \mathcal{N}\left( \frac{\mathbf{u}}{\|\mathbf{u}\|_2} \right) \right\|_2,
\end{aligned}
$$

where in the second equality we used $1-$homogeneity of $\overline{\partial}\mathcal{N}(\mathbf{u})$ and $2-$homogeneity of $\mathcal{N}(\mathbf{u})$, and the inequalities follow from triangle inequality of norms. Hence, for a.e. $t \geq T$, using eq. (37), eq. (38) and eq. (39) we have

$$\left\| \left( \mathbf{I} - \frac{\mathbf{u}(t)\mathbf{u}(t)^\top}{\|\mathbf{u}(t)\|_2^2} \right) \frac{\overline{\partial}\mathcal{N}(\mathbf{u}(t))}{\|\mathbf{u}(t)\|_2} \right\|_2 \geq \gamma - \gamma/4 - \gamma/4 = \gamma/2,$$

implying

$$\frac{d}{dt} \left( \mathcal{N}\left( \frac{\mathbf{u}(t)}{\|\mathbf{u}(t)\|_2} \right) \right) = \left\| \left( \mathbf{I} - \frac{\mathbf{u}(t)\mathbf{u}(t)^\top}{\|\mathbf{u}(t)\|_2^2} \right) \frac{\overline{\partial}\mathcal{N}(\mathbf{u}(t))}{\|\mathbf{u}(t)\|_2} \right\|_2^2 \geq \gamma^2/4,$$

which contradicts the fact that $\lim_{t \to \infty} \mathcal{N}\left( \frac{\mathbf{u}(t)}{\|\mathbf{u}(t)\|_2} \right)$ converges, thus proving our claim.

Now, before we turn to prove Theorem 5.1, we state another useful lemma.

**Lemma C.7.** *If $\lim_{t \to \infty} \mathbf{u}(t) \neq \mathbf{0}$, then there exists $\eta > 0$ and $T \geq 0$ such that $\|\mathbf{u}(t)\|_2 \geq \eta$, for all $t \geq T$.*

*Proof.* The proof is built using the argument already presented in the proof of Lemma 5.3, and we consider two cases.

**Case 1:** $\mathcal{N}(\mathbf{u}(0)) > 0$.
We assume $\mathcal{N}(\mathbf{u}(0)) = \gamma > 0$, and therefore, $\|\mathbf{u}(0)\|_2 > 0$. From Lemma C.4, for all $t \geq 0$, we have $\mathcal{N}(\mathbf{u}(t)) \geq \mathcal{N}(\mathbf{u}(0)) = \gamma$, which implies

$$\frac{1}{2}\frac{d\|\mathbf{u}\|_2^2}{dt} = \mathbf{u}^\top \dot{\mathbf{u}} = 2\mathcal{N}(\mathbf{u}(t)) \geq 2\gamma, \text{ for a.e. } t \geq 0, \tag{40}$$

which in turn implies

$$\|\mathbf{u}(t)\|_2 \geq \|\mathbf{u}(0)\|_2 \text{ for all } t \geq 0. \tag{41}$$

Thus, we can choose $\eta = \|\mathbf{u}(0)\|_2$, and $T = 0$.

**Case 2:** $\mathcal{N}(\mathbf{u}(0)) \leq 0$.

For this case we may further assume that $\mathcal{N}(\mathbf{u}(t)) \leq 0$, for all $t \geq 0$, since if for some $\bar{t}$, $\mathcal{N}(\mathbf{u}(\bar{t})) > 0$, then using Lemma C.4, we have $\|\mathbf{u}(t)\|_2 \geq \|\mathbf{u}(\bar{t})\|_2$, for all $t \geq \bar{t}$. Then, since $\mathcal{N}(\mathbf{u}(\bar{t})) > 0$ implies $\|\mathbf{u}(\bar{t})\|_2 > 0$, we may choose $\eta = \|\mathbf{u}(\bar{t})\|_2$ and $T = \bar{t}$.

So, let us assume that $\mathcal{N}(\mathbf{u}(t)) \leq 0$, for all $t \geq 0$. Then,

$$\frac{1}{2}\frac{d\|\mathbf{u}\|_2^2}{dt} = \mathbf{u}^\top \dot{\mathbf{u}} = 2\mathcal{N}(\mathbf{u}(t)) \leq 0, \text{ for a.e. } t \geq 0.$$

Therefore, $\|\mathbf{u}(t)\|_2$ decreases with time, and hence, $\lim_{t\to\infty}\|\mathbf{u}(t)\|_2$ exists and $\|\mathbf{u}(t)\|_2 \geq \lim_{t\to\infty}\|\mathbf{u}(t)\|_2$, for all $t \geq 0$. Since we have assumed $\lim_{t\to\infty}\mathbf{u}(t) \neq \mathbf{0}$, we have $\lim_{t\to\infty}\|\mathbf{u}(t)\|_2 > 0$. Thus, we may choose $\eta = \lim_{t\to\infty}\|\mathbf{u}(t)\|_2$, and $T = 0$.

$\square$

### C.3 Proof of Theorem 5.1

Consider the differential inclusion

$$\dot{\mathbf{u}} \in \partial\mathcal{N}_{-\ell'(\mathbf{0},\mathbf{y}),\mathcal{H}}(\mathbf{u}) \subseteq -\sum_{i=1}^{n}\nabla_{\hat{y}}\ell(0,y_i)\partial\mathcal{H}(\mathbf{x}_i;\mathbf{u}), \mathbf{u}(0) = \mathbf{w}_0, \tag{42}$$

and let $\mathbf{u}(t)$ be its unique solution. By Lemma 5.3, either $\lim_{t\to\infty}\mathbf{u}(t) = \mathbf{0}$ or $\lim_{t\to\infty}\frac{\mathbf{u}(t)}{\|\mathbf{u}(t)\|_2}$ exists.

We first consider the case when $\lim_{t\to\infty}\mathbf{u}(t) = \mathbf{0}$. Here, we define $\eta = 1$ and fix an $\epsilon \in (0, \eta)$. Then, we choose $\overline{T}$ large enough such that

$$\|\mathbf{u}(t)\|_2 \leq \epsilon, \text{ for all } t \geq \overline{T}. \tag{43}$$

Next, if $\lim_{t\to\infty}\mathbf{u}(t) \neq 0$, then from Lemma C.7, there exists $\eta > 0$ and $T \geq 0$ such that $\|\mathbf{u}(t)\|_2 \geq 2\eta$, for all $t \geq T$. Also, from Lemma 5.3,

$$\lim_{t\to\infty}\mathbf{u}(t)/\|\mathbf{u}(t)\|_2 = \hat{\mathbf{u}},$$

where $\hat{\mathbf{u}}$ is a non-negative KKT point of

$$\max_{\|\mathbf{u}\|_2^2=1}\mathcal{N}_{-\ell'(\mathbf{0},\mathbf{y}),\mathcal{H}}(\mathbf{u}) = -\ell'(\mathbf{0},\mathbf{y})^\top\mathcal{H}(\mathbf{X};\mathbf{u}). \tag{44}$$

For a fixed $\epsilon \in (0, \eta)$, we choose $\overline{T} > T$ such that

$$\frac{\mathbf{u}(t)^\top\hat{\mathbf{u}}}{\|\mathbf{u}(t)\|_2} \geq 1 - \epsilon, \text{ for all } t \geq \overline{T}. \tag{45}$$

Having chosen $\overline{T}$ for a fixed $\epsilon \in (0, \eta)$ in both cases, we next choose $C$ such that $\frac{\ln(C)}{4\beta\tilde{\beta}} = \overline{T}$. From Lemma 5.2, there exists $\overline{\delta}$ such that for any $\delta \leq \overline{\delta}$

$$\|\mathbf{w}(t)\|_2 \leq \sqrt{C}\delta, \text{ for all } t \in \left[0, \frac{\ln(C)}{4\beta\tilde{\beta}}\right],$$

and

$$\left\|\frac{\mathbf{w}(\overline{T})}{\delta} - \mathbf{u}(\overline{T})\right\|_2 \leq \epsilon. \tag{46}$$

Thus, we may write $\frac{\mathbf{w}(\overline{T})}{\delta} = \mathbf{u}(\overline{T}) + \zeta$, where $\|\zeta\|_2 \leq \epsilon$. If $\lim_{t\to\infty}\mathbf{u}(t) = \mathbf{0}$, then, using eq. (43),

$$\|\mathbf{w}(\overline{T})\|_2 \leq 2\delta\epsilon.$$

Else, since $\epsilon \in (0, \eta)$, and $\|\mathbf{u}(\overline{T})\|_2 \geq 2\eta$, we have $\|\mathbf{u}(\overline{T}) + \zeta\|_2 \geq \eta$. Hence,

$$\frac{\mathbf{w}(\overline{T})}{\|\mathbf{w}(\overline{T})\|_2} = \frac{\mathbf{u}(\overline{T}) + \zeta}{\|\mathbf{u}(\overline{T}) + \zeta\|_2},$$

which implies

$$\frac{\mathbf{w}(\overline{T})^\top \hat{\mathbf{u}}}{\|\mathbf{w}(\overline{T})\|_2} = \frac{\mathbf{u}(\overline{T})^\top \hat{\mathbf{u}} + \zeta^\top \hat{\mathbf{u}}}{\|\mathbf{u}(\overline{T}) + \zeta\|_2} = \left( \frac{\mathbf{u}(\overline{T})^\top \hat{\mathbf{u}}}{\|\mathbf{u}(\overline{T})\|_2} \right) \frac{\|\mathbf{u}(\overline{T})\|_2}{\|\mathbf{u}(\overline{T}) + \zeta\|_2} + \frac{\zeta^\top \hat{\mathbf{u}}}{\|\mathbf{u}(\overline{T}) + \zeta\|_2}.$$

Now, since

$$\frac{\|\mathbf{u}(\overline{T})\|_2}{\|\mathbf{u}(\overline{T}) + \zeta\|_2} \geq \frac{\|\mathbf{u}(\overline{T})\|_2}{\|\mathbf{u}(\overline{T})\|_2 + \|\zeta\|_2} = \frac{1}{1 + \frac{\|\zeta\|_2}{\|\mathbf{u}(\overline{T})\|_2}} \geq \frac{1}{1 + \frac{\epsilon}{2\eta}} \geq 1 - \frac{\epsilon}{2\eta},$$

and

$$\frac{\zeta^\top \hat{\mathbf{u}}}{\|\mathbf{u}(\overline{T}) + \zeta\|_2} \geq \frac{-\epsilon}{\eta},$$

we have

$$\frac{\mathbf{w}(\overline{T})^\top \hat{\mathbf{u}}}{\|\mathbf{w}(\overline{T})\|_2} \geq (1 - \epsilon) \left( 1 - \frac{\epsilon}{2\eta} \right) - \frac{\epsilon}{\eta} \geq 1 - \left( 1 + \frac{3}{2\eta} \right) \epsilon.$$

### C.4    Proof of Corollary 5.4.1

Consider the differential inclusion

$$\dot{\mathbf{u}} \in \partial \mathcal{N}_{-\ell'(\mathbf{0}, \mathbf{y}), \mathcal{H}}(\mathbf{u}) \subseteq -\sum_{i=1}^{n} \nabla_{\hat{y}} \ell(0, y_i) \partial \mathcal{H}(\mathbf{x}_i; \mathbf{u}), \mathbf{u}(0) = \mathbf{w}_0, \tag{47}$$

and let $\mathbf{u}(t)$ be its unique solution. From separability, we can write $\mathbf{u}(t) = [\mathbf{u}_1(t), \dots, \mathbf{u}_H(t)]$ such that for all $j \in [H]$ we have

$$\dot{\mathbf{u}}_j \in \partial \mathcal{N}_{-\ell'(\mathbf{0}, \mathbf{y}), \mathcal{H}_j}(\mathbf{u}_j) \subseteq -\sum_{i=1}^{n} \nabla_{\hat{y}} \ell(0, y_i) \partial \mathcal{H}_j(\mathbf{x}_i; \mathbf{u}_j), \mathbf{u}_j(0) = \mathbf{w}_{0j}, \tag{48}$$

where $\mathbf{w}_0 = [\mathbf{w}_{01}, \dots, \mathbf{w}_{0H}]^\top$. By Lemma 5.3, for all $j \in [H]$, either $\lim_{t \to \infty} \mathbf{u}_j(t) = \mathbf{0}$ or $\lim_{t \to \infty} \frac{\mathbf{u}_j(t)}{\|\mathbf{u}_j(t)\|_2}$ exists.

Let $\mathcal{Z}$ be the collection of all indices such that $\lim_{t \to \infty} \mathbf{u}_j(t) = \mathbf{0}$, for all $j \in \mathcal{Z}$, and $\mathcal{Z}^c$ be the complement of $\mathcal{Z}$ in $[H]$. For all $j \in \mathcal{Z}^c$, from Lemma C.7, there exists $\eta_j > 0$ and $T_j \geq 0$ such that $\|\mathbf{u}_j(t)\|_2 \geq 2\eta_j$, for all $t \geq T_j$. Define $\eta = \min(1, \min_{j \in \mathcal{Z}^c} \eta_j)$ and $T = \max_{j \in \mathcal{Z}^c} T_j$, and fix an $\epsilon \in (0, \eta)$.

From Lemma 5.3, for all $j \in \mathcal{Z}^c$ we have

$$\lim_{t \to \infty} \mathbf{u}_j(t) / \|\mathbf{u}_j(t)\|_2 = \hat{\mathbf{u}}_j,$$

where $\hat{\mathbf{u}}_j$ is a non-negative KKT point of

$$\max_{\|\mathbf{u}_j\|_2^2 = 1} \mathcal{N}_{-\ell'(\mathbf{0}, \mathbf{y}), \mathcal{H}_j}(\mathbf{u}_j) = -\ell'(\mathbf{0}, \mathbf{y})^\top \mathcal{H}_j(\mathbf{X}; \mathbf{u}_j). \tag{49}$$

Then, for a given $\epsilon$, we choose $\overline{T}_1 > T$ such that

$$\frac{\mathbf{u}_j(t)^\top \hat{\mathbf{u}}_j}{\|\mathbf{u}_j(t)\|_2} \geq 1 - \epsilon, \text{ for all } t \geq \overline{T}_1, \text{ and all } j \in \mathcal{Z}^c. \tag{50}$$

For all $j \in \mathcal{Z}$, we choose $\overline{T}_2$ large enough such that

$$\|\mathbf{u}_j(t)\|_2 \leq \epsilon, \text{ for all } t \geq \overline{T}_2 \text{ and all } j \in \mathcal{Z}. \tag{51}$$

Define $\overline{T} = \max(\overline{T}_1, \overline{T}_2)$ and choose $C$ such that $\frac{\ln(C)}{4\beta\tilde{\beta}} = \overline{T}$. From Lemma 5.2, there exists $\overline{\delta}$ such that for any $\delta \leq \overline{\delta}$

$$\|\mathbf{w}(t)\|_2 \leq \sqrt{C}\delta, \text{ for all } t \in \left[0, \frac{\ln(C)}{4\beta\tilde{\beta}}\right],$$

and

$$\left\|\frac{\mathbf{w}(\overline{T})}{\delta} - \mathbf{u}(\overline{T})\right\|_2 \leq \epsilon. \tag{52}$$

By separability, we may write $\frac{\mathbf{w}_j(\overline{T})}{\delta} = \mathbf{u}_j(\overline{T}) + \zeta_j$, where $\|\zeta_j\|_2 \leq \epsilon$. If $j \in \mathcal{Z}^c$, then, using eq. (51) we have

$$\|\mathbf{w}_j(\overline{T})\|_2 \leq 2\delta\epsilon.$$

Else, since $\epsilon \in (0, \eta)$, and $\|\mathbf{u}_j(\overline{T})\|_2 \geq 2\eta$, we have $\|\mathbf{u}_j(\overline{T}) + \zeta_j\|_2 \geq \eta$. Hence, using similar reasoning as in the later part of the proof of Theorem 5.1 we get for all $j \in \mathcal{Z}^c$,

$$\frac{\mathbf{w}_j(\overline{T})^\top \hat{\mathbf{u}}_j}{\|\mathbf{w}_j(\overline{T})\|_2} \geq 1 - \left(1 + \frac{3}{2\eta}\right)\epsilon.$$

# D   Proofs Omitted from Section 5.2

We first prove Lemma 5.6

## D.1   Proof of Lemma 5.6

We note that since $\{\overline{\mathbf{w}}_n, \overline{\mathbf{w}}_z\}$ is a saddle point of

$$L(\mathbf{w}_n, \mathbf{w}_z) = \frac{1}{2}\|\mathcal{H}_n(\mathbf{X}; \mathbf{w}_n) + \mathcal{H}_z(\mathbf{X}; \mathbf{w}_z) - \mathbf{y}\|^2, \tag{53}$$

and $\overline{\mathbf{w}}_z = 0$, we have

$$\begin{bmatrix} \mathbf{0} \\ \mathbf{0} \end{bmatrix} \in \begin{bmatrix} \partial_{\mathbf{w}_n} L(\overline{\mathbf{w}}_n, \mathbf{0}) \\ \partial_{\mathbf{w}_z} L(\overline{\mathbf{w}}_n, \mathbf{0}) \end{bmatrix}, \tag{54}$$

which establishes $\mathbf{0} \in \partial_{\mathbf{w}_n} L(\overline{\mathbf{w}}_n, \mathbf{0})$.

To prove Theorem 5.5, we first describe the approximate dynamics of $\{\mathbf{w}_n(t), \mathbf{w}_z(t)\}$ near the saddle point in the following lemma.

**Lemma D.1.** *Let $\{\overline{\mathbf{w}}_n, \overline{\mathbf{w}}_z\}$ satisfy Assumption 3, and define $\overline{\mathbf{y}} = \mathbf{y} - \mathcal{H}_n(\mathbf{X}; \overline{\mathbf{w}}_n)$. Let $C > 1$ be an arbitrarily large constant and $\{\mathbf{w}_n(t), \mathbf{w}_z(t)\}$ satisfy for a.e. $t \geq 0$*

$$\begin{bmatrix} \dot{\mathbf{w}}_n \\ \dot{\mathbf{w}}_z \end{bmatrix} \in -\begin{bmatrix} \partial_{\mathbf{w}_n} L(\mathbf{w}_n, \mathbf{w}_z) \\ \partial_{\mathbf{w}_z} L(\mathbf{w}_n, \mathbf{w}_z) \end{bmatrix}, \begin{bmatrix} \mathbf{w}_n(0) \\ \mathbf{w}_z(0) \end{bmatrix} = \begin{bmatrix} \overline{\mathbf{w}}_n + \delta\zeta_n \\ \overline{\mathbf{w}}_z + \delta\zeta_z \end{bmatrix}, \tag{55}$$

*where $\delta^2 \leq \min(\frac{1}{2C}, \frac{\gamma^2}{4})$ and $\|\zeta_n\|_2 = \|\zeta_z\|_2 = 1$. Then*

$$\|\mathbf{w}_n(t) - \overline{\mathbf{w}}_n\|_2^2 + \|\mathbf{w}_z(t) - \overline{\mathbf{w}}_z\|_2^2 \leq 2C\delta^2, \text{ for all } t \in \left[0, \frac{1}{M_2}\ln(C)\right], \tag{56}$$

*where $M_2$ is a positive constant[5]. Further, for the differential inclusion*

$$\dot{\mathbf{u}} \in \partial\mathcal{N}_{\overline{\mathbf{y}}, \mathcal{H}_z}(\mathbf{u}), \mathbf{u}(0) = \zeta_z, \tag{57}$$

---

[5]$M_2$ here is same as in Theorem 5.5. See the statement of Theorem 5.5 and the the proof of Lemma D.1 for more details.

*and for any $\epsilon > 0$ there exists a small enough $\overline{\delta} > 0$ such that for any $\delta \leq \overline{\delta}$,*

$$\left\| \frac{\mathbf{w}_z(t)}{\delta} - \mathbf{u}(t) \right\|_2 \leq \epsilon, \ \text{for all } t \in \left[0, \frac{\ln(C)}{M_2}\right], \tag{58}$$

*where $\mathbf{u}(t)$ is a certain solution of eq. (57).*

*Proof.* We note that since $\{\overline{\mathbf{w}}_n, \overline{\mathbf{w}}_z\}$ is a saddle point of

$$L\left(\mathbf{w}_n, \mathbf{w}_z\right) = \frac{1}{2} \|\mathcal{H}_n(\mathbf{X}; \mathbf{w}_n) + \mathcal{H}_z(\mathbf{X}; \mathbf{w}_z) - \mathbf{y}\|^2, \tag{59}$$

we have

$$\begin{bmatrix} \mathbf{0} \\ \mathbf{0} \end{bmatrix} \in \begin{bmatrix} \sum_{i=1}^n \overline{y}_i \partial \mathcal{H}_n(\mathbf{x}_i; \overline{\mathbf{w}}_n) \\ \sum_{i=1}^n \overline{y}_i \partial \mathcal{H}_z(\mathbf{x}_i; \overline{\mathbf{w}}_z) \end{bmatrix}. \tag{60}$$

We now define

$$\Delta_n(t) = \mathcal{H}_n(\mathbf{X}; \mathbf{w}_n(t)) - \mathcal{H}_n(\mathbf{X}; \overline{\mathbf{w}}_n), \Delta_z(t) = \mathcal{H}_z(\mathbf{X}; \mathbf{w}_z(t)), \text{ and } \mathbf{Z}(t) = \|\mathbf{w}_n(t) - \overline{\mathbf{w}}_n\|_2^2 + \|\mathbf{w}_z(t)\|_2^2.$$

Since $(\mathbf{w}_n(t), \mathbf{w}_z(t))$ is a continuous curve, $\mathbf{Z}(t)$ is also a continuous curve. Note that $\mathbf{Z}(0) = 2\delta^2 \leq \min(1/C, \gamma^2/2) < \min(1, \gamma^2)$. Therefore, there exists some $\overline{t} > 0$, such that $\mathbf{Z}(t) \leq \min(1, \gamma^2)$, for all $t \in [0, \overline{t}]$. Let $T^*$ be the smallest $t > 0$ such that $\mathbf{Z}(T^*) = \min(1, \gamma^2)$. Hence, for all $t \in [0, T^*]$, $\mathbf{Z}(t) \leq \min(1, \gamma^2)$. Our next goal is to find a lower bound for $T^*$. We operate in $[0, T^*]$.

Recall that

$$\|\mathcal{H}_z(\mathbf{X}; \mathbf{w}_z)\|_2 \leq \beta \|\mathbf{w}_z\|_2^2. \tag{61}$$

Moreover, by the locally Lipschitz property of $\mathcal{H}_n(\mathbf{X}; \mathbf{w}_n)$, there exists $\mu_1 > 0$ such that

$$\|\Delta_N(t)\|_2 \leq \mu_1 \|\mathbf{w}_n(t) - \overline{\mathbf{w}}_n\|_2, \text{ for all } t \in [0, T^*]. \tag{62}$$

Define $\mathbf{e}(t) = \mathcal{H}_n(\mathbf{X}; \mathbf{w}_n(t)) + \mathcal{H}_z(\mathbf{X}; \mathbf{w}_z(t)) - \mathbf{y}$, then, using $\mathbf{Z}(t) \leq 1$, we have

$$\|\mathbf{e}(t)\|_2 = \|\mathcal{H}_n(\mathbf{X}; \mathbf{w}_n(t)) + \mathcal{H}_z(\mathbf{X}; \mathbf{w}_z(t)) - \mathbf{y}\|_2$$
$$\leq \|\overline{\mathbf{y}}\|_2 + \|\Delta_N(t)\|_2 + \|\mathcal{H}_z(\mathbf{X}; \mathbf{w}_z(t))\|_2 \leq \|\overline{\mathbf{y}}\|_2 + \mu_1 + \beta := M_1,$$

where in the last inequality we used eq. (62) and eq. (61). Using Corollary B.1.1, we have

$$\frac{1}{2} \frac{d \|\mathbf{w}_z(t)\|_2^2}{dt} = -2\mathcal{H}_z(\mathbf{X}; \mathbf{w}_z)^\top \mathbf{e} \leq 2\beta M_1 \|\mathbf{w}_z(t)\|_2^2. \tag{63}$$

We first consider the case when $\mathcal{H}_n(\mathbf{x}; \mathbf{w}_n)$ has a locally Lipschitz gradient. Let

$$\mathbf{J}(\mathbf{w}_n) =: [\nabla \mathcal{H}_n(\mathbf{x}_1; \mathbf{w}_n), \ldots, \nabla \mathcal{H}_n(\mathbf{x}_n; \mathbf{w}_n)] \in \mathbb{R}^{d \times n}.$$

From eq. (60), we have

$$\mathbf{0} = \sum_{i=1}^n \overline{y}_i \nabla \mathcal{H}_n(\mathbf{x}_i; \overline{\mathbf{w}}_n) = \mathbf{J}(\overline{\mathbf{w}}_n)\overline{\mathbf{y}}. \tag{64}$$

By the locally Lipschitz property of $\nabla \mathcal{H}_n(\mathbf{x}; \mathbf{w}_n)$, we may assume that there exists $\mu_2 > 0$ such that

$$\|\mathbf{J}(\mathbf{w}_n(t))\overline{\mathbf{y}} - \mathbf{J}(\overline{\mathbf{w}}_n)\overline{\mathbf{y}}\|_2 \leq \mu_2 \|\mathbf{w}_n(t) - \overline{\mathbf{w}}_n\|_2. \tag{65}$$

Further, since $\mathbf{w}_n(t)$ is bounded for all $t \in [0, T^*]$, we may assume there exists $\mu_3 > 0$ such that

$$\|\mathbf{J}(\mathbf{w}_n(t))\|_2 \leq \mu_3. \tag{66}$$

Thus,

$$
\begin{aligned}
\frac{1}{2}\frac{d\|\mathbf{w}_n - \overline{\mathbf{w}}_n\|_2^2}{dt} &= -\langle \mathbf{w}_n - \overline{\mathbf{w}}_n, \mathbf{J}(\mathbf{w}_n)\mathbf{e}\rangle \\
&= -\langle \mathbf{w}_n - \overline{\mathbf{w}}_n, \mathbf{J}(\mathbf{w}_n)\left(\mathcal{H}_n(\mathbf{X};\mathbf{w}_n) + \mathcal{H}_z(\mathbf{X};\mathbf{w}_z) - \mathbf{y}\right)\rangle \\
&= -\langle \mathbf{w}_n - \overline{\mathbf{w}}_n, \mathbf{J}(\mathbf{w}_n)\left(\Delta_n(t) + \Delta_z(t) - \overline{\mathbf{y}}\right)\rangle \\
&= \langle \mathbf{w}_n - \overline{\mathbf{w}}_n, \mathbf{J}(\mathbf{w}_n)\overline{\mathbf{y}}\rangle - \langle \mathbf{w}_n - \overline{\mathbf{w}}_n, \mathbf{J}(\mathbf{w}_n)\left(\Delta_n(t) + \Delta_z(t)\right)\rangle \\
&= \langle \mathbf{w}_n - \overline{\mathbf{w}}_n, \mathbf{J}(\mathbf{w}_n)\overline{\mathbf{y}} - \mathbf{J}(\overline{\mathbf{w}}_n)\overline{\mathbf{y}}\rangle - \langle \mathbf{w}_n - \overline{\mathbf{w}}_n, \mathbf{J}(\mathbf{w}_n)\left(\Delta_n(t) + \Delta_z(t)\right)\rangle \\
&\leq \mu_2\|\mathbf{w}_n - \overline{\mathbf{w}}_n\|_2^2 + \|\mathbf{w}_n - \overline{\mathbf{w}}_n\|_2\|\mathbf{J}(\mathbf{w}_n)\|_2\left(\|\Delta_n(t)\|_2 + \|\Delta_z(t)\|_2\right) \\
&\leq \mu_2\|\mathbf{w}_n - \overline{\mathbf{w}}_n\|_2^2 + \mu_1\mu_3\|\mathbf{w}_n - \overline{\mathbf{w}}_n\|_2^2 + \beta\mu_3\|\mathbf{w}_n - \overline{\mathbf{w}}_n\|_2\|\mathbf{w}_z\|_2^2 \\
&\leq (\mu_2 + \mu_1\mu_3)\|\mathbf{w}_n - \overline{\mathbf{w}}_n\|_2^2 + \beta\mu_3\|\mathbf{w}_z\|_2^2.
\end{aligned}
$$

The third equality follows from definition of $\Delta_n(t)$ and $\Delta_z(t)$. In last equality, we use eq. (64). The first inequality follows from Cauchy-Schwartz and eq. (65). We get second inequality from eq. (66), eq. (62) and eq. (61). In the final inequality, we use $\|\mathbf{w}_n(t) - \overline{\mathbf{w}}_n\|_2 \leq 1$, for all $t \in [0, T^*]$.

Combining the above inequality with eq. (63), we obtain

$$
\frac{1}{2}\frac{d\mathbf{Z}(t)}{dt} \leq (\mu_2 + \mu_1\mu_3)\|\mathbf{w}_n(t) - \overline{\mathbf{w}}_n\|_2^2 + \beta(2M_1 + \mu_3)\|\mathbf{w}_z(t)\|_2^2 \leq M_2\mathbf{Z}(t),
$$

where $M_2 := \max(\mu_2 + \mu_1\mu_3, \beta(2M_1 + \mu_3))$. Therefore, for all $t \in [0, T^*]$, $\mathbf{Z}(t) \leq \mathbf{Z}(0)e^{tM_2}$ implies

$$
T^* \geq \frac{1}{M_2}\ln\left(\frac{1}{\mathbf{Z}(0)}\right) = \frac{1}{M_2}\ln\left(\frac{1}{2\delta^2}\right).
$$

Since we assume $2\delta^2 \leq \frac{1}{C}$, we have that $T^* \geq \frac{1}{M_2}\ln(C)$. Thus, for all $t \in \left[0, \frac{1}{M_2}\ln(C)\right]$,

$$
\mathbf{Z}(t) \leq C\mathbf{Z}(0) \leq 2C\delta^2,
$$

proving eq. (56).

We next consider the case when $\mathcal{H}_n(\mathbf{X};\mathbf{w}_n)$ does not have a locally Lipschitz gradient. Recall that if $\|\mathbf{w}_n - \overline{\mathbf{w}}_n\|_2 \leq \gamma$, then

$$
\langle \overline{\mathbf{w}}_n - \mathbf{w}_n, \mathbf{s}\rangle \geq -\kappa\|\overline{\mathbf{w}}_n - \mathbf{w}_n\|_2^2, \text{ where } \mathbf{s} \in -\partial_{\mathbf{w}_n}L(\mathbf{w}_n, \mathbf{0}). \tag{67}
$$

Further, we may assume that there exists a constant $\mu_3 > 0$ such that if $\|\mathbf{w}_n - \overline{\mathbf{w}}_n\|_2 \leq \gamma$, then

$$
\max_{i \in [n]}\|\mathbf{p}_i\|_2 \leq \mu_3, \text{ where } \mathbf{p}_i \in \partial\mathcal{H}(\mathbf{x}_i;\mathbf{w}_n). \tag{68}
$$

Using the chain rule, we also have

$$
\partial_{\mathbf{w}_n}L(\mathbf{w}_n, \mathbf{w}_z) \subseteq \partial_{\mathbf{w}_n}\left(L(\mathbf{w}_n, \mathbf{w}_z) - L(\mathbf{w}_n, \mathbf{0})\right) + \partial_{\mathbf{w}_n}L(\mathbf{w}_n, \mathbf{0}). \tag{69}
$$

Note that $\partial_{\mathbf{w}_n}\left(L(\mathbf{w}_n, \mathbf{w}_z) - L(\mathbf{w}_n, \mathbf{0})\right) \subseteq \sum_{i=1}^{n}\mathcal{H}_z(\mathbf{x}_i;\mathbf{w}_z)\partial\mathcal{H}_n(\mathbf{x}_i;\mathbf{w}_n)$. Therefore, if $\|\mathbf{w}_n - \overline{\mathbf{w}}_n\|_2 \leq \gamma$, then, using eq. (68), for any $\mathbf{w}_z$, and $\mathbf{p} \in \partial_{\mathbf{w}_n}\left(L(\mathbf{w}_n, \mathbf{w}_z) - L(\mathbf{w}_n, \mathbf{0})\right)$, we have

$$
\|\mathbf{p}\|_2 \leq \mu_3\|\mathcal{H}_n(\mathbf{X};\mathbf{w}_z)\|_1 \leq \mu_3\sqrt{n}\|\mathcal{H}_n(\mathbf{X};\mathbf{w}_z)\|_2 \leq \mu_3\sqrt{n}\beta\|\mathbf{w}_z\|_2^2. \tag{70}
$$

Next, using eq. (69) we have

$$
\begin{aligned}
\frac{1}{2}\frac{d\|\mathbf{w}_n - \overline{\mathbf{w}}_n\|_2^2}{dt} &= \langle \mathbf{w}_n - \overline{\mathbf{w}}_n, \dot{\mathbf{w}}_n\rangle \in -\langle \mathbf{w}_n - \overline{\mathbf{w}}_n, \partial_{\mathbf{w}_n}L(\mathbf{w}_n, \mathbf{w}_z)\rangle \\
&\in -\langle \mathbf{w}_n - \overline{\mathbf{w}}_n, \partial_{\mathbf{w}_n}\left(L(\mathbf{w}_n, \mathbf{w}_z) - L(\mathbf{w}_n, \mathbf{0})\right) + \partial_{\mathbf{w}_n}L(\mathbf{w}_n, \mathbf{0})\rangle.
\end{aligned}
$$

Since $\|\mathbf{w}_n(t) - \overline{\mathbf{w}}_n\|_2 \leq \gamma$ for all $t \in [0, T^*]$, using eq. (67) and eq. (70), we have

$$\frac{1}{2} \frac{d\|\mathbf{w}_n - \overline{\mathbf{w}}_n\|_2^2}{dt} \leq \mu_3\sqrt{n}\beta\|\mathbf{w}_n - \overline{\mathbf{w}}_n\|_2\|\mathbf{w}_z\|_2^2 + \kappa\|\overline{\mathbf{w}}_n - \mathbf{w}_n\|_2^2 \leq \mu_3\sqrt{n}\beta\|\mathbf{w}_z\|_2^2 + \kappa\|\overline{\mathbf{w}}_n - \mathbf{w}_n\|_2^2,$$

where in the last inequality we use $\|\mathbf{w}_n(t) - \overline{\mathbf{w}}_n\|_2 \leq 1$, for all $t \in [0, T^*]$. Combining above inequality with eq. (63), we have

$$\frac{1}{2} \frac{d\mathbf{Z}(t)}{dt} \leq \beta(2M_1 + \mu_3\sqrt{n})\|\mathbf{w}_z(t)\|_2^2 + \kappa\|\overline{\mathbf{w}}_n - \mathbf{w}_n\|_2^2 \leq M_2\mathbf{Z}(t),$$

where $M_2 := \max(\kappa, \beta(2M_1 + \mu_3\sqrt{n}))$. Therefore, for all $t \in [0, T^*]$, we have that $\mathbf{Z}(t) \leq \mathbf{Z}(0)e^{tM_2}$ implies

$$T^* \geq \frac{1}{M_2}\ln\left(\frac{1}{\mathbf{Z}(0)}\right) = \frac{1}{M_2}\ln\left(\frac{1}{2\delta^2}\right).$$

Since we assume $2\delta^2 \leq \frac{1}{C}$, we have that $T^* \geq \frac{1}{M_2}\ln(C)$. Thus, for all $t \in \left[0, \frac{1}{M_2}\ln(C)\right]$,

$$\mathbf{Z}(t) \leq C\mathbf{Z}(0) \leq 2C\delta^2,$$

proving eq. (56).

We now move towards proving the second part. We use a similar technique as in the proof of Lemma 5.2. We define $\xi(t) = \mathbf{e}(t) + \overline{\mathbf{y}}$. Then,

$$\begin{aligned}\|\xi(t)\|_2 &= \|\overline{\mathbf{y}} + \mathcal{H}_n(\mathbf{X}; \mathbf{w}_n(t)) + \mathcal{H}_z(\mathbf{X}; \mathbf{w}_z(t)) - \mathbf{y}\|_2 \\ &= \|\mathcal{H}_n(\mathbf{X}; \mathbf{w}_n(t)) - \mathcal{H}_n(\mathbf{X}; \overline{\mathbf{w}}_n) + \mathcal{H}_z(\mathbf{X}; \mathbf{w}_z(t))\|_2 \\ &\leq \mu_1\|\mathbf{w}_n(t) - \overline{\mathbf{w}}_n\|_2 + \beta\|\mathbf{w}_z(t)\|_2^2 \\ &\leq \mu_1\sqrt{C}\delta + \beta C\delta^2.\end{aligned}$$

Thus, the dynamics of $\mathbf{w}_z(t)$ can be written as

$$\dot{\mathbf{w}}_z \in -\sum_{i=1}^{n} e_i \partial\mathcal{H}_z(\mathbf{x}_i; \mathbf{w}_z) = \sum_{i=1}^{n} (\overline{y}_i - \xi(t)) \partial\mathcal{H}_z(\mathbf{x}_i; \mathbf{w}_z). \tag{71}$$

Dividing eq. (71) by $\delta$, and using 1-homogeneity of $\partial\mathcal{H}(\mathbf{x}; \mathbf{w})$ (Corollary B.1.1), we have

$$\frac{\dot{\mathbf{w}}_z}{\delta} \in \frac{1}{\delta}\sum_{i=1}^{n} (\overline{y}_i - \xi(t)) \partial\mathcal{H}_z(\mathbf{x}_i; \mathbf{w}_z) = \sum_{i=1}^{n} (\overline{y}_i - \xi(t)) \partial\mathcal{H}_z(\mathbf{x}_i; \mathbf{w}_z/\delta). \tag{72}$$

Now, consider the differential inclusion

$$\frac{d\tilde{\mathbf{w}}_z}{dt} \in \sum_{i=1}^{n} \overline{y}_i \partial\mathcal{H}_z(\mathbf{x}_i; \tilde{\mathbf{w}}_z), \tilde{\mathbf{w}}_z(0) = \zeta_z. \tag{73}$$

Since for all $t \in \left[0, \frac{1}{M_2}\ln(C)\right]$, $\|\xi(t)\|_2 \leq \mu_1\sqrt{C}\delta + \beta C\delta^2$, using Lemma C.1, there exists a small enough $\overline{\delta}$ such that for all $\delta \leq \overline{\delta}$,

$$\left\|\tilde{\mathbf{w}}_z(t) - \frac{\mathbf{w}_z(t)}{\delta}\right\|_2 \leq \epsilon,$$

where $\tilde{\mathbf{w}}(t)$ is a solution of eq. (73).

$\square$

## D.2 Proof of Theorem 5.5

Consider the differential inclusion

$$\dot{\mathbf{u}} \in \sum_{i=1}^{n} \overline{y}_i \partial \mathcal{H}_z(\mathbf{x}_i; \mathbf{u}), \mathbf{u}(0) = \zeta_z, \tag{74}$$

and let $\mathbf{u}(t)$ be its unique solution. By Lemma 5.3, either $\lim_{t \to \infty} \mathbf{u}(t) = \mathbf{0}$ or $\lim_{t \to \infty} \frac{\mathbf{u}(t)}{\|\mathbf{u}(t)\|_2}$ exists.

We first consider the case when $\lim_{t \to \infty} \mathbf{u}(t) = \mathbf{0}$. Here, we define $\eta = 1$ and fix an $\epsilon \in (0, \eta)$. Then, we choose $\overline{T}$ large enough such that

$$\|\mathbf{u}(t)\|_2 \leq \epsilon, \forall t \geq \overline{T}. \tag{75}$$

We next consider the case when $\lim_{t \to \infty} \mathbf{u}(t) \neq 0$. Then, from Lemma C.7, there exists $\eta > 0$ and $T \geq 0$ such that $\|\mathbf{u}(t)\|_2 \geq 2\eta$, for all $t \geq T$. Also, from Lemma 5.3,

$$\lim_{t \to \infty} \mathbf{u}(t)/\|\mathbf{u}(t)\|_2 = \hat{\mathbf{u}},$$

where $\hat{\mathbf{u}}$ is a non-negative KKT point of

$$\max \mathcal{H}_z(\mathbf{X}; \mathbf{u})^{\top} \overline{\mathbf{y}}, \text{ such that } \|\mathbf{u}\|_2^2 = 1. \tag{76}$$

For a fixed $\epsilon \in (0, \eta)$, we choose $\overline{T} > T$ such that

$$\frac{\mathbf{u}(t)^{\top} \hat{\mathbf{u}}}{\|\mathbf{u}(t)\|_2} \geq 1 - \epsilon, \text{ for all } t \geq \overline{T}. \tag{77}$$

Having chosen $\overline{T}$ for a fixed $\epsilon \in (0, \eta)$ in both cases, we next choose $C$ such that $\frac{\ln(C)}{M_2} = \overline{T}$. From Lemma D.1, there exists $\overline{\delta}$ such that for any $\delta \leq \overline{\delta}$

$$\|\mathbf{w}_n(t) - \overline{\mathbf{w}}_n\|_2^2 + \|\mathbf{w}_z(t) - \overline{\mathbf{w}}_z\|_2^2 \leq 2C\delta^2, \text{ for all } t \in \left[0, \frac{\ln(C)}{M_2}\right],$$

and

$$\left\| \frac{\mathbf{w}_z(\overline{T})}{\delta} - \mathbf{u}(\overline{T}) \right\|_2 \leq \epsilon. \tag{78}$$

Thus, we may write $\frac{\mathbf{w}_z(\overline{T})}{\delta} = \mathbf{u}(\overline{T}) + \zeta$, where $\|\zeta\|_2 \leq \epsilon$. If $\lim_{t \to \infty} \mathbf{u}(t) = \mathbf{0}$, then, using eq. (75),

$$\|\mathbf{w}_z(\overline{T})\|_2 \leq 2\delta\epsilon.$$

Else, since $\epsilon \in (0, \eta)$, and $\|\mathbf{u}(\overline{T})\|_2 \geq 2\eta$, we have $\|\mathbf{u}(\overline{T}) + \zeta\|_2 \geq \eta$. Hence,

$$\frac{\mathbf{w}_z(\overline{T})}{\|\mathbf{w}_z(\overline{T})\|_2} = \frac{\mathbf{u}(\overline{T}) + \zeta}{\|\mathbf{u}(\overline{T}) + \zeta\|_2}$$

which implies

$$\frac{\mathbf{w}_z(\overline{T})^{\top} \hat{\mathbf{u}}}{\|\mathbf{w}_z(\overline{T})\|_2} = \frac{\mathbf{u}(\overline{T})^{\top} \hat{\mathbf{u}} + \zeta^{\top} \hat{\mathbf{u}}}{\|\mathbf{u}(\overline{T}) + \zeta\|_2} = \left(\frac{\mathbf{u}(\overline{T})^{\top} \hat{\mathbf{u}}}{\|\mathbf{u}(\overline{T})\|_2}\right) \frac{\|\mathbf{u}(\overline{T})\|_2}{\|\mathbf{u}(\overline{T}) + \zeta\|_2} + \frac{\zeta^{\top} \hat{\mathbf{u}}}{\|\mathbf{u}(\overline{T}) + \zeta\|_2}.$$

Since

$$\frac{\|\mathbf{u}(\overline{T})\|_2}{\|\mathbf{u}(\overline{T}) + \zeta\|_2} \geq \frac{\|\mathbf{u}(\overline{T})\|_2}{\|\mathbf{u}(\overline{T})\|_2 + \|\zeta\|_2} = \frac{1}{1 + \frac{\|\zeta\|_2}{\|\mathbf{u}(\overline{T})\|_2}} \geq \frac{1}{1 + \frac{\epsilon}{2\eta}} \geq 1 - \frac{\epsilon}{2\eta},$$

and

$$\frac{\zeta^{\top} \hat{\mathbf{u}}}{\|\mathbf{u}(\overline{T}) + \zeta\|_2} \geq \frac{-\epsilon}{\eta},$$

we have that

$$\frac{\mathbf{w}_z(\overline{T})^{\top} \hat{\mathbf{u}}}{\|\mathbf{w}_z(\overline{T})\|_2} \geq (1 - \epsilon)\left(1 - \frac{\epsilon}{2\eta}\right) - \frac{\epsilon}{\eta} \geq 1 - \left(1 + \frac{3}{2\eta}\right)\epsilon.$$

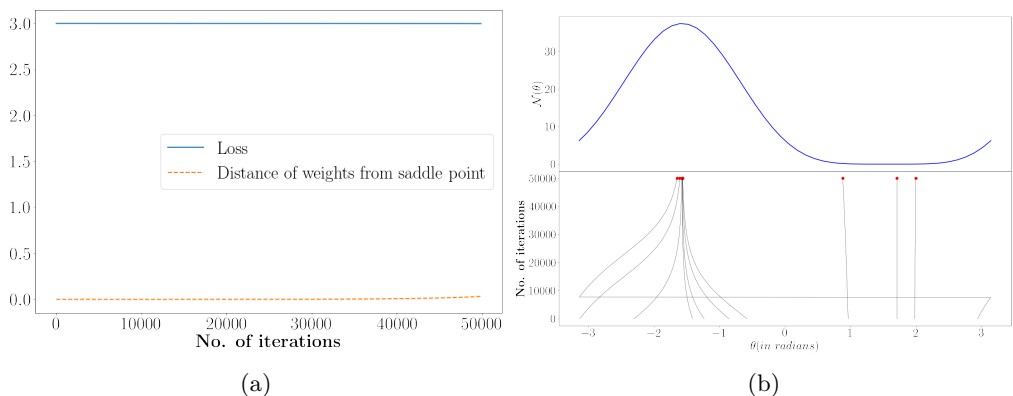

Figure 4: Panel ($a$): the evolution of training loss and the $\ell_2$-distance of the weights from the saddle point with iterations. Panel ($b$): The lower part shows the evolution of $\arctan(\mathbf{u}_{2i}(t)/\mathbf{u}_{1i}(t))$ for the last 10 hidden neurons. The top plot shows the constrained NCF $\mathcal{N}_{\bar{\mathbf{y}},\mathcal{H}}(\theta) = \sum_{i=1}^{n} \bar{y}_i \max(0, [\cos(\theta), \sin(\theta)]^\top \mathbf{x}_i)^2$, where $\bar{\mathbf{y}}$ is the residual error at the saddle point. We see that the weights remain near the saddle point and loss barely changes, though the weights of the last 10 neurons converge in direction to the KKT points of the constrained NCF.

### D.3 Numerical Experiments

We experimentally show the phenomenon of directional convergence among the weights of small magnitude near saddle points. For this, we again consider the example depicted in Figure 1. Recall, the network architecture is defined as $\mathcal{H}(x_1, x_2; \{\mathbf{u}_i\}_{i=1}^{20}) = \sum_{i=1}^{20} \max(0, \mathbf{u}_{1i}x_1 + \mathbf{u}_{2i}x_2)^2$, and there are 50 unit norm inputs with corresponding labels generated using the function $\mathcal{H}^*(x_1, x_2) = 5\max(0, x_1)^2 + 4\max(0, -x_1)^2$.

We consider the following saddle point $\{\bar{\mathbf{u}}_1, \cdots, \bar{\mathbf{u}}_{20}\}$, where $\bar{\mathbf{u}}_i = [\sqrt{1/2}, 0]$, if $i \leq 10$, and $\bar{\mathbf{u}}_i = [0, 0]$, if $i > 10$. Now, $\{\bar{\mathbf{u}}_1, \cdots, \bar{\mathbf{u}}_{20}\}$ is a saddle point since if $x_1 \geq 0$, then $\mathcal{H}(x_1, x_2; \{\bar{\mathbf{u}}_i\}_{i=1}^{20}) = 5\max(0, x_1)^2 = \mathcal{H}^*(x_1, x_2)$, and if $x_1 < 0$, then $\nabla_{\mathbf{u}_i}\mathcal{H}(x_1, x_2; \{\bar{\mathbf{u}}_i\}_{i=1}^{20}) = \mathbf{0}$, for all $i \leq 10$. Moreover, $\nabla_{\mathbf{u}_i}\mathcal{H}(x_1, x_2; \{\bar{\mathbf{u}}_i\}_{i=1}^{20}) = \mathbf{0}$, for all $i > 10$.

We initialize the weights by adding an i.i.d. Gaussian vector of standard deviation $10^{-5}$ to $\{\bar{\mathbf{u}}_1, \cdots, \bar{\mathbf{u}}_{20}\}$. We use square loss and optimize using gradient descent for 50000 iterations with step-size $5 \cdot 10^{-5}$ . We plot the evolution of the overall loss and the $\ell_2$ distance of the network weights from the saddle point with iterations in Figure 4a, and Figure 4b contains the evolution of the angle the weight vectors of the last 10 hidden neurons (they have small magnitude at initialization) make with the positive horizontal axis with iterations and the constrained NCF defined with respect to the residual error at the saddle point. It is evident that the training loss barely changes, and the weights remains close to the saddle point. Moreover, the individual weight vectors for the neurons with small magnitude converge to the the KKT point of the constrained NCF. We note that the constrained NCF seems to have a set of flat KKT points, and therefore, if the weights are initialized in that set, then they remain there as seen in Figure 4b.

## E Gradient Flow Dynamics of $f(u_1, u_2) = u_1|u_2|$

In the following lemma, we describe the gradient flow solutions of $f(u_1, u_2) = u_1|u_2|$ when initialized at $[1, 0]^\top$.

**Lemma E.1.** *For any $T > 0$, consider the following time-varying function*

$$\mathbf{u}_T(t) = \begin{bmatrix} u_{1T}(t) \\ u_{2T}(t) \end{bmatrix} = \begin{cases} \begin{bmatrix} 1 \\ 0 \end{bmatrix}, \text{ for all } t \in [0, T] \\ \begin{bmatrix} \cosh(t - T) \\ \sinh(t - T) \end{bmatrix}, \text{ for all } t \geq T, \end{cases} \tag{79}$$

*then, for a.e. $t \geq 0$, $\mathbf{u}_T(t)$ satisfies*

$$\begin{bmatrix} \dot{u}_1 \\ \dot{u}_2 \end{bmatrix} \in \begin{bmatrix} \partial_{u_1} f(u_1, u_2) \\ \partial_{u_2} f(u_1, u_2) \end{bmatrix} = \begin{bmatrix} |u_2| \\ u_1 \partial |u_2| \end{bmatrix}, \begin{bmatrix} u_1(0) \\ u_2(0) \end{bmatrix} = \begin{bmatrix} 1 \\ 0 \end{bmatrix}, \tag{80}$$

*where*

$$\partial |u_2| \in \begin{cases} [-1, 1], & \text{if } u_2 = 0 \\ 1, & \text{if } u_2 > 0, \\ -1, & \text{if } u_2 < 0 \end{cases}.$$

*Proof.* For $t \in [0, T)$, $\mathbf{u}_T(t)$ is a constant, hence,

$$\begin{bmatrix} \dot{u}_{1T} \\ \dot{u}_{2T} \end{bmatrix} = \begin{bmatrix} 0 \\ 0 \end{bmatrix}.$$

Since $u_{1T}(t) = 1$ and $u_{2T}(t) = 0$, for all $t \in [0, T)$, we have

$$\begin{bmatrix} |u_{2T}(t)| \\ u_{1T}(t) \partial |u_{2T}(t)| \end{bmatrix} = \begin{bmatrix} 0 \\ [-1, 1] \end{bmatrix} \ni \begin{bmatrix} 0 \\ 0 \end{bmatrix}.$$

For $t > T$, $u_{1T}(t)$ and $u_{2T}(t)$ are continuous and differentiable functions, thus,

$$\begin{bmatrix} \dot{u}_{1T} \\ \dot{u}_{2T} \end{bmatrix} = \begin{bmatrix} \sinh(t - T) \\ \cosh(t - T) \end{bmatrix}.$$

Since $u_{2T}(t) > 0$, hence, for all $t > T$, we have

$$\begin{bmatrix} |u_{2T}(t)| \\ u_{1T}(t) \partial |u_{2T}(t)| \end{bmatrix} = \begin{bmatrix} \sinh(t - T) \\ \cosh(t - T) \end{bmatrix},$$

completing the proof. $\qquad \square$

The above lemma shows that for any $T > 0$, $\mathbf{u}_T(t)$ defined in eq. (79) is a gradient flow solution of $f(u_1, u_2)$ when initialized at $[1, 0]^\top$. For any finite $T$, it is easy to see that $\lim_{t \to \infty} \mathbf{u}_T(t)/\|\mathbf{u}_T(t)\|_2 = [1/\sqrt{2}, 1/\sqrt{2}]^\top$. Hence, for a fixed $T$ and any $\epsilon > 0$, we can choose $\overline{T}$ such that

$$\left\| \frac{\mathbf{u}_T(\overline{T})}{\|\mathbf{u}_T(\overline{T})\|_2} - \begin{bmatrix} 1/\sqrt{2} \\ 1/\sqrt{2} \end{bmatrix} \right\| \geq \epsilon.$$

In the next lemma, we show that except for the set $\{u_2 = 0, u_1 > 0\}$, all points are non-branching for gradient flow dynamics of $f(u_1, u_2) = u_1 |u_2|$.

**Lemma E.2.** *Let $[z_1, z_2]^\top \in \mathbb{R}^2 \backslash \{u_2 = 0, u_1 > 0\}$, then the differential inclusion*

$$\begin{bmatrix} \dot{u}_1 \\ \dot{u}_2 \end{bmatrix} \in \begin{bmatrix} \partial_{u_1} f(u_1, u_2) \\ \partial_{u_2} f(u_1, u_2) \end{bmatrix} = \begin{bmatrix} |u_2| \\ u_1 \partial |u_2| \end{bmatrix}, \begin{bmatrix} u_1(0) \\ u_2(0) \end{bmatrix} = \begin{bmatrix} z_1 \\ z_2 \end{bmatrix} \tag{81}$$

*has a unique solution, where*

$$\partial |u_2| \in \begin{cases} [-1, 1], & \text{if } u_2 = 0 \\ 1, & \text{if } u_2 > 0, \\ -1, & \text{if } u_2 < 0 \end{cases}.$$

*Proof.* We first note that

$$\dot{u}_1 u_1 = \dot{u}_2 u_2,$$

which implies

$$\frac{d}{dt}(u_1^2(t) - u_2^2(t)) = 0.$$

Hence,

$$u_1^2(t) - u_2^2(t) = z_1^2 - z_2^2. \tag{82}$$

Now, we prove the lemma by considering different cases.

**Case 1:** $|z_2| > |z_1|$. From eq. (82), we have $u_1^2(t) - u_2^2(t) = z_1^2 - z_2^2$. Thus, $u_2^2(t) = u_1^2(t) + z_2^2 - z_1^2$. Now, since $|z_2| > |z_1|$, we get that $u_2(t) > 0, \forall t \geq 0$. Since $u_2(t)$ stays away from the set $\{u_2 = 0\}$, $\partial_{u_1} f(u_1(t), u_2(t))$ and $\partial_{u_2} f(u_1(t), u_2(t))$ are always unique and hence, $[u_1(t), u_2(t)]$ is unique.

**Case 2:** $|z_2| \leq |z_1|, z_2 \neq 0, z_1 > 0$. In this case, since $|z_2| > 0$, we have $z_1 = |z_1| > 0$. Now, since $\partial_{u_1} f(u_1(t), u_2(t)) = |u_2(t)| \geq 0$ and $z_1 > 0$, therefore, $u_1(t) \geq z_1 > 0, \forall t \geq 0$. Now,

$$\dot{(u_2^2)} = 2u_2\dot{u}_2 = 2|u_2|u_1.$$

Since $u_1(t) > 0, \forall t \geq 0$, we get $u_2^2(t) \geq u_2^2(0) = z_2^2 > 0, \forall t \geq 0$. Hence, $u_2(t)$ stays away from the set $\{u_2 = 0\}$, which implies $\partial_{u_1} f(u_1(t), u_2(t))$ and $\partial_{u_2} f(u_1(t), u_2(t))$ are always unique and thus, $[u_1(t), u_2(t)]$ is unique.

Before proceeding to the next cases, we first show that the set $\mathcal{S} = \{u_2 = 0, u_1 \leq 0\}$ is a stable critical point, that is, if $\{u_1(\bar{t}), u_2(\bar{t})\} \in \mathcal{S}$, for some time $\bar{t} \geq 0$, then, $[u_1(t), u_2(t)] = [u_1(\bar{t}), u_2(\bar{t})], \forall t \geq \bar{t}$.

Proving $[0, 0]$ is a stable critical point is trivial since $\partial_{u_1} f(0, 0) = 0 = \partial_{u_2} f(u_1, u_2)$.

Now, suppose $[u_1(\bar{t}), u_2(\bar{t})] = [\hat{u}_1, \hat{u}_2]$, where $\hat{u}_1 < 0$ and $\hat{u}_2 = 0$. Then, $f(\hat{u}_1, \hat{u}_2) = 0$, and since gradient flow can not decrease the objective value, we have $f(u_1(t), u_2(t)) \geq 0$, for all $t \geq \bar{t}$. Also, since for any $\tilde{u}_1 < 0$ and $\tilde{u}_2 \neq 0$, we have $f(\tilde{u}_1, \tilde{u}_2) < 0$, therefore, gradient flow can only move along the $u_2 = 0$ axis, i.e., only $u_1(t)$ can change. However, since $\partial_{u_1} f(\hat{u}_1, \hat{u}_2) = |\hat{u}_2| = 0$, $u_1(t)$ can not change for $t \geq \bar{t}$. Hence, $[u_1(t), u_2(t)] = [u_1(\bar{t}), u_2(\bar{t})]$, for all $t \geq \bar{t}$.

**Case 3:** $|z_2| < |z_1|, z_1 < 0$. From eq. (82), we have $u_1^2(t) - u_2^2(t) = z_1^2 - z_2^2$. Thus, $u_1^2(t) = u_2^2(t) + z_1^2 - z_2^2$. Now, since $|z_1| > |z_2|$, we get that $u_1^2(t) > 0, \forall t \geq 0$. Now, since $u_1(0) = z_1 < 0$, therefore, $u_1(t) < 0, \forall t \geq 0$. Hence, if $u_2(\bar{t})$ is 0 for the first time at some $\bar{t} \geq 0$, then from the stability of the set $\mathcal{S}$, we have $[u_1(t), u_2(t)] = [u_1(\bar{t}), u_2(\bar{t})], \forall t \geq \bar{t}$. For $0 \leq t < \bar{t}$, since $u_2(t) \neq 0$, $\partial_{u_1} f(u_1(t), u_2(t))$ and $\partial_{u_2} f(u_1(t), u_2(t))$ are always unique and hence, $[u_1(t), u_2(t)]$ is unique.

**Case 3:** $|z_2| = |z_1|, z_1 \leq 0$. From eq. (82), we have $|u_1(t)| = |u_2(t)|$. Now, since $u_1(0) = z_1 \leq 0$, therefore, if $u_1(\bar{t})$ is 0 for the first time at some $\bar{t} \geq 0$, then $u_2(\bar{t})$ is also 0 for the first time at $\bar{t}$. Then, from the stability of $[0, 0]$, we have $[u_1(t), u_2(t)] = [0, 0], \forall t \geq \bar{t}$. For $0 \leq t < \bar{t}$, since $u_2(t) \neq 0$, $\partial_{u_1} f(u_1(t), u_2(t))$ and $\partial_{u_2} f(u_1(t), u_2(t))$ are always unique and hence, $[u_1(t), u_2(t)]$ is unique. $\square$

## F   Proof of Lemma 5.4

Since $\{v_*, \mathbf{u}_*\}$ is a KKT point of

$$\max_{v^2 + \|\mathbf{u}\|_2^2 = 1} \mathcal{N}_{\mathbf{z}, \mathcal{H}}(v, \mathbf{u}) = v\mathbf{z}^\top \sigma(\mathbf{X}^\top \mathbf{u}),$$

therefore, for some $\lambda \in \mathbb{R}$, we have

$$0 = \lambda v_* + \mathbf{z}^\top \sigma(\mathbf{X}^\top \mathbf{u}_*), \tag{83}$$

$$\mathbf{0} \in \lambda \mathbf{u}_* + v_* \sum_{i=1}^n \sigma'(\mathbf{x}_i^\top \mathbf{u}_*) z_i \mathbf{x}_i, \tag{84}$$

where $\sigma'(\cdot)$ is the subdifferential of $\sigma(\cdot)$. Also, since $\mathbf{x}_i^\top \mathbf{u}_* \neq 0$, $\sigma'(\mathbf{x}_i^\top \mathbf{u}_*)$ is unique for all $i \in [n]$.

Now, we may further assume that $\min_{i \in [n]} |\mathbf{x}_i^\top \mathbf{u}_*| = \eta_1$ and $\max_{i \in [n]} \|\mathbf{x}_i\|_2 = \eta_2$, for some $\eta_1, \eta_2 > 0$. Choose $\gamma = \eta_1 / (2\eta_2)$, and recall $\mathcal{S} = \{v, \mathbf{u} : sign(v_*)v > \|\mathbf{u}\|_2, \|\mathbf{u} - \mathbf{u}_*\|_2 \leq \gamma\}$. Note that since $v_* \mathbf{z}^\top \sigma(\mathbf{X}^\top \mathbf{u}_*) > 0$, therefore, $|v_*| \neq 0$ and $sign(v_*) \in \{-1, 1\}$. Also, multiplying eq. (83) by $v_*$ and using $v_* \mathbf{z}^\top \sigma(\mathbf{X}^\top \mathbf{u}_*) > 0$, we get that $\lambda < 0$.

Now, the gradient flow equation is

$$\begin{bmatrix} \dot{v} \\ \dot{\mathbf{u}} \end{bmatrix} \in \begin{bmatrix} \mathbf{z}^\top \sigma(\mathbf{X}^\top \mathbf{u}) \\ v \sum_{i=1}^n \sigma'(\mathbf{x}_i^\top \mathbf{u}) z_i \mathbf{x}_i, \end{bmatrix}, \begin{bmatrix} v(0) \\ \mathbf{u}(0) \end{bmatrix} = \begin{bmatrix} p \\ \mathbf{q} \end{bmatrix},$$

where $\{p, \mathbf{q}\} \in \mathcal{S}$. We first note that

$$\frac{d}{dt}(v^2 - \|\mathbf{u}\|_2^2) = 2v\dot{v} - 2\mathbf{u}^\top \dot{\mathbf{u}} = 0,$$

which implies

$$v^2(t) - \|\mathbf{u}(t)\|_2^2 = p^2 - \|\mathbf{q}\|_2^2. \tag{85}$$

Since $sign(v_*)p > \|\mathbf{q}\|_2$, we get $sign(p) = sign(v_*)$ and thus $|p| > \|\mathbf{q}\|_2$. Therefore, from the above equation, we get $v^2(t) - \|\mathbf{u}(t)\|_2^2 > 0$, which implies $v^2(t) > 0$. Since $v^2(t)$ is continuous and never becomes 0, we have $sign(v(t)) = sign(v(0)) = sign(p) = sign(v_*)$.

To prove uniqueness, we will show that $\mathbf{x}_i^\top \mathbf{u}(t) \neq 0$, for all $i \in [n]$ and $t \geq 0$. Note that, since $\|\mathbf{u}_* - \mathbf{q}\|_2 \leq \gamma$, we may assume $\mathbf{q} = \mathbf{u}_* + \epsilon \mathbf{b}$, where $\|\mathbf{b}\|_2 = 1$ and $|\epsilon| \leq \gamma$. Thus,

$$|\mathbf{x}_i^\top \mathbf{u}(0)| = |\mathbf{x}_i^\top \mathbf{q}| \geq |\mathbf{x}_i^\top \mathbf{u}_*| - |\epsilon||\mathbf{x}_i^\top \mathbf{b}| \geq \eta_1 - \gamma\eta_2 = \eta_1/2,$$

which implies $\mathbf{x}_i^\top \mathbf{u}(0) \neq 0$, for all $i \in [n]$. For the sake of contradiction, suppose there exists some $\bar{t} > 0$ such that $\mathbf{x}_i^\top \mathbf{u}(\bar{t}) = 0$, for some $i \in [n]$, for the first time. Then, for all $t \in [0, \bar{t})$, we have $\mathbf{x}_i^\top \mathbf{u}(t) \neq 0$, for all $i \in [n]$. Furthermore, due to continuity of $\mathbf{u}(t)$, we have, for all $i \in [n]$,

$$sign(\mathbf{x}_i^\top \mathbf{u}(t)) = sign(\mathbf{x}_i^\top \mathbf{u}(0)), \forall t \in [0, \bar{t}).$$

Now, note that

$$\begin{aligned} \mathbf{x}_i^\top \mathbf{u}_* \mathbf{x}_i^\top \mathbf{u}(0) &= \mathbf{x}_i^\top \mathbf{u}_* \mathbf{x}_i^\top \mathbf{u}_* + \epsilon \mathbf{x}_i^\top \mathbf{u}_* \mathbf{x}_i^\top \mathbf{b} \\ &\geq |\mathbf{x}_i^\top \mathbf{u}_*|^2 - |\epsilon||\mathbf{x}_i^\top \mathbf{u}_*||\mathbf{x}_i^\top \mathbf{b}| \\ &= |\mathbf{x}_i^\top \mathbf{u}_*|(|\mathbf{x}_i^\top \mathbf{u}_*| - |\epsilon||\mathbf{x}_i^\top \mathbf{b}|) \geq \eta_1^2/2. \end{aligned}$$

Thus, for all $i \in [n]$,

$$sign(\mathbf{x}_i^\top \mathbf{u}(t)) = sign(\mathbf{x}_i^\top \mathbf{u}(0)) = sign(\mathbf{x}_i^\top \mathbf{u}_*), \forall t \in [0, \bar{t}),$$

which implies

$$\begin{aligned} \mathbf{u}(\bar{t}) &= \mathbf{u}(0) + \int_0^{\bar{t}} v(t) \sum_{i=1}^n \sigma'(\mathbf{x}_i^\top \mathbf{u}(t)) z_i \mathbf{x}_i dt \\ &= \mathbf{u}(0) + \left( \int_0^{\bar{t}} v(t) dt \right) \sum_{i=1}^n \sigma'(\mathbf{x}_i^\top \mathbf{u}_*) z_i \mathbf{x}_i \\ &= \mathbf{u}(0) + \left( \int_0^{\bar{t}} v(t)/v_* dt \right) v_* \sum_{i=1}^n \sigma'(\mathbf{x}_i^\top \mathbf{u}_*) z_i \mathbf{x}_i = \mathbf{u}(0) - \lambda \left( \int_0^{\bar{t}} v(t)/v_* dt \right) \mathbf{u}_*, \end{aligned}$$

where in the last equality we used eq. (84). Now, recall that $\lambda < 0$ and $sign(v(t)) = sign(v_*)$, for all $t \geq 0$. Thus, we can write

$$\mathbf{u}(\bar{t}) = \mathbf{u}(0) + \bar{\alpha}\mathbf{u}_*,$$

for some $\bar{\alpha} > 0$. This implies that, for all $i \in [n]$,

$$\mathbf{x}_i^\top \mathbf{u}_* \mathbf{x}_i^\top \mathbf{u}(\bar{t}) = \mathbf{x}_i^\top \mathbf{u}_* \mathbf{x}_i^\top \mathbf{u}(0) + \bar{\alpha}|\mathbf{x}_i^\top \mathbf{u}_*|^2 \geq \eta_1^2/2 + \bar{\alpha}\eta_1^2 > 0,$$

which contradicts $\mathbf{x}_i^\top \mathbf{u}(\bar{t}) = 0$, for some $i \in [n]$.

## G  KKT Points of NCF: Some Examples

In this section we provide some examples where KKT points of the NCF can be computed analytically.

### G.1  Symmetric Data and Squared ReLU

Let $n$ be even, and suppose the training set is the union of $\{\mathbf{x}_i, y_i\}_{i=1}^{n/2}$ and $\{-\mathbf{x}_i, -y_i\}_{i=1}^{n/2}$ such that $\|\sum_{i=1}^{n/2} y_i \mathbf{x}_i \mathbf{x}_i^\top\|_2 \neq 0$. Let the neural network $\mathcal{H}(\mathbf{x}; \mathbf{u}) = \sigma(\mathbf{x}^\top \mathbf{u})$, where $\sigma(x) = \max(x, \alpha x)^2$, for some $\alpha \in \mathbb{R}$. Then, for square or logistic loss, the constrained NCF is

$$\max_{\|\mathbf{u}\|_2^2 = 1} \sum_{i=1}^{n/2} y_i \left( \sigma(\mathbf{x}_i^\top \mathbf{u}) - \sigma(-\mathbf{x}_i^\top \mathbf{u}) \right).$$

Now, since $\sigma(x) - \sigma(-x) = (1 + \alpha^2) x^2$, the constrained NCF can be written as

$$\max_{\|\mathbf{u}\|_2^2 = 1} (1 + \alpha^2) \mathbf{u}^\top \left( \sum_{i=1}^{n/2} y_i \mathbf{x}_i \mathbf{x}_i^\top \right) \mathbf{u}.$$

It is well known that KKT points of the above problem will be the eigenvectors of the matrix $\sum_{i=1}^{n/2} y_i \mathbf{x}_i \mathbf{x}_i^\top$.

### G.2  Symmetric Data and 2-layer ReLU

This example is inspired from Lyu et al. (2021). Let $n$ be even, and suppose the training set is the union of $\{\mathbf{x}_i, y_i\}_{i=1}^{n/2}$ and $\{-\mathbf{x}_i, -y_i\}_{i=1}^{n/2}$ such that $\|\sum_{i=1}^{n/2} y_i \mathbf{x}_i\|_2 \neq 0$. Let the neural network $\mathcal{H}(\mathbf{x}; v, \mathbf{u}) = v\sigma(\mathbf{x}^\top \mathbf{u})$, where $\sigma(x) = \max(x, \alpha x)$, for some $\alpha \in \mathbb{R} \backslash \{-1\}$. Then, for square or logistic loss, the constrained NCF is

$$\max_{v^2 + \|\mathbf{u}\|_2^2 = 1} \sum_{i=1}^{n/2} v y_i \left( \sigma(\mathbf{x}_i^\top \mathbf{u}) - \sigma(-\mathbf{x}_i^\top \mathbf{u}) \right).$$

Now, since $\sigma(x) - \sigma(-x) = (1 + \alpha) x$, the constrained NCF can be written as

$$\max_{v^2 + \|\mathbf{u}\|_2^2 = 1} (1 + \alpha) v \left( \sum_{i=1}^{n/2} y_i \mathbf{x}_i^\top \right) \mathbf{u}.$$

Let $\mathbf{q} = \sum_{i=1}^{n/2} y_i \mathbf{x}_i$. Now, $\{v_*, \mathbf{u}_*\}$ is a KKT point of the above problem if there exists $\lambda_*$ such that

$$\lambda_* v_* + \mathbf{q}^\top \mathbf{u}_* = \mathbf{0}, \tag{86}$$
$$\lambda_* \mathbf{u}_* + v_* \mathbf{q} = \mathbf{0}, \tag{87}$$
$$v_*^2 + \|\mathbf{u}_*\|_2^2 = 1. \tag{88}$$

Now, if we choose $\lambda_* = 0$, then $v_* = 0$ and $\mathbf{u}_* = \hat{\mathbf{u}}$ satisfies the KKT equation, where $\hat{\mathbf{u}}$ is any vector such that $\hat{\mathbf{u}}^\top \mathbf{q} = 0$ and $\|\hat{\mathbf{u}}\|_2 = 1$.

Next, if we choose $\lambda_* \neq 0$, then $v_* \neq 0$. Because, if $v_* = 0$, then, from eq. (87), we get $\|\mathbf{u}_*\|_2 = 0$, which leads to violation of eq. (88). Since $v_* \neq 0$, we have $\mathbf{u}_* = -(v_*/\lambda_*)\mathbf{q}$. Putting this in eq. (86), we get $\lambda_*^2 = \|\mathbf{q}\|_2^2$. From eq. (88), we gave

$$1 = v_*^2 + \|\mathbf{u}_*\|_2^2 = v_*^2 + v_*^2/\lambda_*^2 \|\mathbf{q}\|_2^2,$$

which implies $v_*^2 = 1/\sqrt{2}$. Hence, $\{\pm 1/\sqrt{2}, \pm \mathbf{q}/(\sqrt{2}\|\mathbf{q}\|_2)\}$ is another set of KKT points.

# H  Directional Convergence for 2-layer (Leaky) ReLU Neural Network

Suppose $\sigma(x) = \max(x, \alpha x)$ is the Leaky ReLU activation for some $\alpha \in \mathbb{R}$, and $\mathcal{H}(\mathbf{x}; \{v_k, \mathbf{u}_k\}_{i=1}^H) = \sum_{k=1}^H v_k \sigma(\mathbf{x}^\top \mathbf{u}_k)$ is the 2-layer Leaky ReLU neural network with $H$ hidden neurons. Now, since $\mathcal{H}$ is separable, from Corollary 5.4.1, for all $k \in [H]$, in the initial stages of training $\{v_k, \mathbf{u}_k\}$ will either be approximately $\mathbf{0}$ or converge in direction to a non-negative KKT point of

$$\max_{v^2 + \|\mathbf{u}\|_2^2 = 1} v\mathbf{y}^\top \sigma(\mathbf{X}^\top \mathbf{u}), \tag{89}$$

where for the sake of simplicity we have assumed the loss function to be either square or logistic. The following lemma sheds more light into the KKT points of the above optimization problem.

**Lemma H.1.** *Suppose $\{v_*, \mathbf{u}_*\}$ is a non-zero KKT point of eq. (89), that is, $v_*\mathbf{y}^\top \sigma(\mathbf{X}^\top \mathbf{u}_*) \neq 0$. Then $|v_*| = \|\mathbf{u}_*\|_2 = 1/\sqrt{2}$, and $\sqrt{2}\mathbf{u}_*$ is a KKT point of*

$$\max_{\|\mathbf{u}\|_2^2 = 1} \mathbf{y}^\top \sigma(\mathbf{X}^\top \mathbf{u}). \tag{90}$$

Using the above lemma we get that if $\{v_k, \mathbf{u}_k\}$ converges in direction to a non-zero KKT point of eq. (89), then $\mathbf{u}_k$ will have converged in direction to a KKT point of eq. (90). This is precisely the result stated in Maennel et al. (2018). We also note that the result in Maennel et al. (2018) were derived under the *balanced initialization* assumption, where $\|\mathbf{u}_k\|_2 = |v_k|$ holds at initialization and remains such throughout training. However, our results hold for more general initializations as well.

## H.1  Proof of Lemma H.1

*Proof.* Since $\{v_*, \mathbf{u}_*\}$ is a KKT point of eq. (89), we have

$$0 = \lambda v_* + \mathbf{y}^\top \sigma(\mathbf{X}^\top \mathbf{u}_*)$$
$$\mathbf{0} \in \lambda \mathbf{u}_* + v_* \mathbf{X}\mathrm{diag}(\sigma'(\mathbf{X}^\top \mathbf{u}_*))\mathbf{y}, \tag{91}$$

where $\sigma'(\cdot)$ is the subdifferential of $\sigma(\cdot)$ and is applied elementwise. Also, for a vector $\mathbf{z}$, $\mathrm{diag}(\mathbf{z})$ denotes a diagonal matrix constructed using entries from $\mathbf{z}$. Upon multiplying the top and bottom equation by $v_*$ and $\mathbf{u}_*^\top$ respectively we get

$$0 = \lambda |v_*|^2 + v_* \mathbf{y}^\top \sigma(\mathbf{X}^\top \mathbf{u}_*)$$
$$\mathbf{0} = \lambda \|\mathbf{u}_*\|_2^2 + v_* \mathbf{u}_*^\top \mathbf{X}\mathrm{diag}(\sigma'(\mathbf{X}^\top \mathbf{u}_*))\mathbf{y} = \lambda \|\mathbf{u}_*\|_2^2 + v_* \mathbf{y}^\top \sigma(\mathbf{X}^\top \mathbf{u}_*),$$

where we used $\sigma'(x)x = \sigma(x)$. Now, since $v_*\mathbf{y}^\top \sigma(\mathbf{X}^\top \mathbf{u}_*) \neq 0$, by adding the two equations and using $|v_*|^2 + \|\mathbf{u}_*\|_2^2 = 1$, we get $\lambda \neq 0$. Since $\lambda \neq 0$, from the above equation we also get $|v_*|^2 = \|\mathbf{u}_*\|_2^2$. Combining this with the fact that $|v_*|^2 + \|\mathbf{u}_*\|_2^2 = 1$, we have $|v_*| = \|\mathbf{u}_*\|_2 = 1/\sqrt{2}$.

Furthermore, since $\lambda \neq 0$, from eq. (91) we get

$$\mathbf{0} \in \mathbf{u}_* + (v_*/\lambda)\mathbf{X}\mathrm{diag}(\sigma'(\mathbf{X}^\top \mathbf{u}_*))\mathbf{y},$$

which implies $\sqrt{2}\mathbf{u}^*$ is a KKT point of eq. (90). $\qquad\square$

# I  Gradient Flow Dynamics of $g(\mathbf{u}) = (u_1|u_2| - 1)^2$

We first describe the gradient field of $g(\mathbf{u}) = (u_1|u_2| - 1)^2$ near $[1, 0]^T$.

**Lemma I.1.** *Let $\tilde{\mathbf{u}} = [1, 0]^T$, then, $\mathbf{0} \in \partial g(\tilde{\mathbf{u}})$. Further, for any $\delta \in (0, 0.1)$, let $\mathbf{u}^\delta = [1 + \delta, \delta]^\top$. Then, for any $\mathbf{s} \in -\partial g(\mathbf{u}^\delta), \|\mathbf{s}\|_2 \geq 1$.*

*Proof.* Since

$$\begin{bmatrix} \partial_{u_1} g(u_1, u_2) \\ \partial_{u_2} g(u_1, u_2) \end{bmatrix} = \begin{bmatrix} 2|u_2|(u_1|u_2| - 1) \\ 2u_1 \partial |u_2|(u_1|u_2| - 1) \end{bmatrix},$$

it is easy to show $\mathbf{0} \in \partial g(\tilde{\mathbf{u}})$. Next,

$$\begin{bmatrix} \partial_{u_1} g(\mathbf{u}^\delta) \\ \partial_{u_2} g(\mathbf{u}^\delta) \end{bmatrix} = \begin{bmatrix} -2\delta(1 - \delta(1 + \delta)) \\ -2(1 + \delta)(1 - \delta(1 + \delta)), \end{bmatrix},$$

and so for any $\mathbf{s} \in -\partial g(\mathbf{u}^\delta)$ we have that

$$\|\mathbf{s}\|_2 \geq 2(1 + \delta)(1 - \delta(1 + \delta)) \geq 2(1 - 0.1 \cdot 1.1) \geq 1.$$

$\square$

The lemma above establishes that there exist points in any arbitrarily small neighborhood of the saddle point $\tilde{\mathbf{u}}$ where the gradient of $g(\mathbf{u})$ has large norm.

In the next lemma we describe the gradient flow dynamics of $g(\mathbf{u}) = (u_1|u_2| - 1)^2$ near $[1, 0]^T$, and show that gradient flow will escape from any arbitrarily small neighborhood of the saddle point $\tilde{\mathbf{u}}$ in a constant time.

**Lemma I.2.** *Let $\tilde{\mathbf{u}} = [1, 0]^T$, and $\mathbf{u}_\delta(t)$ be a solution of the differential inclusion*

$$\begin{bmatrix} \dot{u}_1 \\ \dot{u}_2 \end{bmatrix} \in - \begin{bmatrix} \partial_{u_1} g(u_1, u_2) \\ \partial_{u_2} g(u_1, u_2) \end{bmatrix}, \begin{bmatrix} u_1(0) \\ u_2(0) \end{bmatrix} = \begin{bmatrix} 1 + \delta \\ \delta \end{bmatrix}. \tag{92}$$

*Then, for any $\delta \in (0, 0.1)$, $\|\mathbf{u}_\delta(0.1) - \tilde{\mathbf{u}}\|_2 \geq 0.09$.*

We show that no matter how close the initialization is to the saddle point $\tilde{\mathbf{u}}$, the gradient flow will escape from it within constant time. Here, $\|\mathbf{u}_\delta(0) - \tilde{\mathbf{u}}\|_2 \leq \sqrt{2}\delta$ where $\delta$ can be arbitrarily small and positive. However, $\|\mathbf{u}_\delta(0.1) - \tilde{\mathbf{u}}\|_2 \geq 0.09$, thus, $\mathbf{u}_\delta(t)$ escapes from the neighborhood of $\tilde{\mathbf{u}}$ within constant time for any arbitrarily small $\delta$.

*Proof.* Choose $\delta \in (0.0.1)$ and let $S := \{(u_1, u_2) : u_1 \in [0.8, 1.2], u_2 \in [\delta/2, 0.35]\}$. Note that for any $\mathbf{u} \in S$, we have

$$\delta/2 \leq \delta(1 - 1.2 \cdot 0.35) \leq -2|u_2|(u_1|u_2| - 1) \leq 2 \cdot 0.35 \cdot (1 - 0.8 \cdot \delta/2) \leq 1, \text{ and} \tag{93}$$
$$0.9 \leq 2 \cdot 0.8(1 - 1.2 \cdot 0.35) \leq -2u_1(u_1|u_2| - 1) \leq 2 \cdot 1.2 \cdot (1 - 0.8 \cdot \delta/2) \leq 2.4. \tag{94}$$

Let $\mathbf{u}_\delta(t)$ be a solution of eq. (92). For the sake of brevity, we use $\mathbf{u}(t)$ instead of $\mathbf{u}_\delta(t)$. Note that $\mathbf{u}(0) \in S$. Let $T$ be the smallest $t \geq 0$ such that $\mathbf{u}(T) \notin S$. For all $t \in [0, T]$, $u_2(t) > 0$, thus, $\mathbf{u}(t)$ satisfies

$$\begin{bmatrix} \dot{u}_1 \\ \dot{u}_2 \end{bmatrix} = \begin{bmatrix} -2|u_2|(u_1|u_2| - 1) \\ -2u_1(u_1|u_2| - 1) \end{bmatrix}, \text{ for all } t \in [0, T]. \tag{95}$$

From eq. (93) and eq. (94), for any $t \in [0, T]$, we have

$$\dot{u}_1 \in [\delta/2, 1], \text{ and } \dot{u}_2 \in [0.9, 2.4]. \tag{96}$$

Using these bounds, we next show that $T > 0.1$. Assume for the sake of contradiction, $T \leq 0.1$. Then, from eq. (96), for any $t \in [0, T]$, we have

$$0.8 < u_1(0) \leq u_1(0) + \delta t/2 \leq u_1(t) \leq u_1(0) + t \leq 1 + \delta + 0.1 < 1.2, \text{ and}$$
$$\delta/2 < u_2(0) \leq u_2(0) + 0.9t \leq u_2(t) \leq u_2(0) + 2.4t \leq \delta + 0.24 \leq 0.34$$

From the above equation, we observe that $\mathbf{u}(T) \in S$, which leads to a contradiction. Thus, $T > 0.1$. Hence, using the lower bound on $\dot{u}_2$ in eq. (96), we have

$$u_2(0.1) \geq u_2(0) + 0.9 \cdot 0.1 \geq 0.09.$$

Thus,

$$\|\mathbf{u}(0.1) - \tilde{\mathbf{u}}\|_2 \geq |u_2(0.1)| \geq 0.09.$$

$\square$

## J   Proof of Lemma C.1

We prove Lemma C.1 in a similar way as in (Filippov, 1988), though that proof considers a more general case. For our problem, the proof can be slightly shortened.

To prove Lemma C.1 we make use fo the following lemma

**Lemma J.1.** *(Filippov, 1988, Lemma 13, Section 5) Let for all $t \in [a, b]$ the vector-valued function $\mathbf{x}_k(t)$ be absolutely continuous, $\mathbf{x}_k(t) \to \mathbf{x}(t)$ as $k \to \infty$, and for each $k = 1, 2, \ldots$ the functions $\dot{\mathbf{x}}_k(t) \in M$ almost everywhere on $t \in (a, b)$, with $M$ being a bounded closed set. Then the vector-valued function $\mathbf{x}(t)$ is absolutely continuous and $\dot{\mathbf{x}}(t) \in conv(M)$ almost everywhere on $t \in (a, b)$.*

*Proof of Lemma C.1.* For the sake of contradiction, we assume that the statement in Lemma C.1 is not true. Thus, for some $\epsilon > 0$ there exists a sequence of solutions $\mathbf{u}_j(t)$ of

$$\frac{d\mathbf{u}}{dt} \in \sum_{i=1}^{n} (z_i + f_i^j(t)) \partial \mathcal{H}(\mathbf{x}_i; \mathbf{u}), \mathbf{u}(0) = \mathbf{u}_0, j = 1, 2, \ldots, \tag{97}$$

where $|f_i^j(t)| \leq \delta_j$, for all $i \in [n]$ and $j \geq 1$, and $\delta_j \to 0$, such that for any solution $\tilde{\mathbf{u}}(t)$ of

$$\frac{d\tilde{\mathbf{u}}}{dt} \in \sum_{i=1}^{n} z_i \partial \mathcal{H}(\mathbf{x}_i; \tilde{\mathbf{u}}), \tilde{\mathbf{u}}(0) = \mathbf{u}_0, \tag{98}$$

we have

$$\max_{t \in [0, T]} \|\mathbf{u}_j(t) - \tilde{\mathbf{u}}(t)\|_2 > \epsilon. \tag{99}$$

We also assume that $\delta_j \leq B/\sqrt{n}$, for some positive constant $B$ and for all $j \geq 1$. We first show that $\{\mathbf{u}_j(t)\}_{j=1}^{\infty}$ has a convergent subsequence. Note that for any $t \in [0, T]$, $\mathbf{u}_j(t)$ is bounded since

$$\frac{d\|\mathbf{u}_j\|_2^2}{dt} = 4 \sum_{i=1}^{n} (z_i + f_i^j(t)) \mathcal{H}(\mathbf{x}_i; \mathbf{u}_j) \leq 4\beta \|\mathbf{u}_j\|_2^2 (\|\mathbf{z}\|_2 + B),$$

which implies

$$\|\mathbf{u}_j(t)\|_2^2 \leq \|\mathbf{u}_0\|_2^2 e^{4t\beta(\|\mathbf{z}\|_2 + B)} \leq \|\mathbf{u}_0\|_2^2 e^{4T\beta(\|\mathbf{z}\|_2 + B)} := B_1^2.$$

We next define

$$\chi := \sup\{\|\partial \mathcal{H}(\mathbf{x}; \mathbf{w})\|_2 : \mathbf{w} \in \mathcal{S}^{k-1}\}.$$

We note that $\{\mathbf{u}_j(t)\}_{i=1}^{\infty}$ is equicontinuous, since for any $t_1, t_2 \in [0, T]$ and $j \geq 1$,

$$\|\mathbf{u}_j(t_1) - \mathbf{u}_j(t_2)\|_2 = \left\| \int_{t_1}^{t_2} \sum_{i=1}^{n} (z_i + f_i^j(s)) \partial \mathcal{H}(\mathbf{x}_i; \mathbf{u}_j) ds \right\|_2 \leq \int_{t_1}^{t_2} \sum_{i=1}^{n} |(z_i + f_i^j(s))| \chi \|\mathbf{u}_j\|_2 ds$$

$$\leq |t_2 - t_1| B_1 \chi \sqrt{n} (\|\mathbf{z}\|_2 + B).$$

Therefore, using the Arzelà–Ascoli Theorem, there exists a subsequence $\{\mathbf{u}_{j_k}(t)\}_{k=1}^{\infty}$ that converges uniformly. We denote the limiting function by $\hat{\mathbf{u}}(t)$. We complete our proof by showing $\hat{\mathbf{u}}(t)$ is a solution of eq. (98) since that will lead to a contradiction.

For any vector $\mathbf{u}$, we define $\mathbf{F}(\mathbf{u}) = \sum_{i=1}^{n} z_i \partial \mathcal{H}(\mathbf{x}_i; \mathbf{u})$. For any $\gamma > 0$, the $\gamma-$neighborhood of $F(\mathbf{u})$, denoted by $F^\gamma(\mathbf{u})$, is defined as

$$F^\gamma(\mathbf{u}) = \{\mathbf{v} : \mathbf{v} = \mathbf{h} + \mathbf{q}, \mathbf{h} \in F(\mathbf{u}), \|\mathbf{q}\|_2 \leq \gamma\}.$$

It is easy to show that for any finite $\gamma$, $F^\gamma(\mathbf{u})$ is a nonempty, convex, and compact set. Also, we define

$$\hat{\mathbf{u}}^\eta =: \{\mathbf{u} : \|\mathbf{u} - \hat{\mathbf{u}}\|_2 \leq \eta\}, \hat{t}^\alpha =: \{t : |t - \hat{t}| \leq \alpha\}$$

Choose any $\hat{t} \in [0, T]$. Since, $F(\mathbf{u})$ is upper semicontinuous, for any $\gamma > 0$, there exist a small enough $\eta > 0$ such that for all $\mathbf{u} \in \hat{\mathbf{u}}^\eta(\hat{t})$, $F(\mathbf{u}) \subseteq F^\gamma(\hat{\mathbf{u}}(\hat{t}))$. Further, since $\hat{\mathbf{u}}(t)$ is continuous, there exists a small enough $\alpha$ such that, for all $t \in \hat{t}^\alpha$, $\hat{\mathbf{u}}(t) \in \hat{\mathbf{u}}^\eta(\hat{t})$. Hence, for any $\gamma > 0$, there exist a small enough $\alpha > 0$, such that for all $t \in \hat{t}^\alpha$, $F(\hat{\mathbf{u}}(t)) \subseteq F^\gamma(\hat{\mathbf{u}}(\hat{t}))$.

Next, since $\{\mathbf{u}_{j_k}(t)\}_{k=1}^\infty$ converges uniformly to $\hat{\mathbf{u}}(t)$, we can choose $k_1$ large enough such that $\|\mathbf{u}_{j_k}(t) - \hat{\mathbf{u}}(t)\|_2 \leq \frac{\eta}{2}$, for all $k > k_1$ and $t \in \hat{t}^\alpha$. Since $\hat{\mathbf{u}}(t)$ is continuous, there exist $\beta > 0$, such that $\|\hat{\mathbf{u}}(t) - \hat{\mathbf{u}}(\hat{t})\|_2 \leq \frac{\eta}{2}$, for all $t \in \hat{t}^\beta$. Thus, for all $t \in \hat{t}^\alpha \cap \hat{t}^\beta$ and $k > k_1$,

$$\|\mathbf{u}_{j_k}(t) - \hat{\mathbf{u}}(\hat{t})\|_2 \leq \|\mathbf{u}_{j_k}(t) - \hat{\mathbf{u}}(t)\|_2 + \|\hat{\mathbf{u}}(t) - \hat{\mathbf{u}}(\hat{t})\|_2 \leq \eta.$$

Hence, $F(\mathbf{u}_{j_k}(t)) \subseteq F^\gamma(\hat{\mathbf{u}}(\hat{t}))$, for all $t \in \hat{t}^\alpha \cap \hat{t}^\beta$ and $k > k_1$.

Also, since $\delta_{j_k} \to 0$, we can choose $k_2$ large enough, such that $\delta_{j_k} \leq \frac{\gamma}{nB_1\chi}$, for all $k > k_2$. Thus, $\|\sum_{i=1}^n f_i^{j_k}(t)\partial\mathcal{H}(\mathbf{x}_i; \mathbf{u}_{j_k}(t))\|_2 \leq nB_1\chi\delta_{j_k} \leq \gamma$, for all $k > k_2$. Therefore, for all $k \geq \max\{k_1, k_2\}$ and $t \in \hat{t}^\alpha \cap \hat{t}^\beta$,

$$\dot{\mathbf{u}}_{j_k}(t) \in F(\mathbf{u}_{j_k}(t)) + \sum_{i=1}^n f_i^{j_k}(t)\partial\mathcal{H}(\mathbf{x}_i; \mathbf{u}_{j_k}(t)) \in F^{2\gamma}(\hat{\mathbf{u}}(\hat{t})), \text{ for a.e. } t \in \hat{t}^\alpha \cap \hat{t}^\beta.$$

From Lemma J.1, $\hat{\mathbf{u}}(t)$ is absolutely continuous in $\hat{t}^\alpha \cap \hat{t}^\beta$ and

$$\dot{\hat{\mathbf{u}}}(t) \in F^{2\gamma}(\hat{\mathbf{u}}(\hat{t})), \text{ for a.e. } t \in \hat{t}^\alpha \cap \hat{t}^\beta.$$

We can cover the interval $[0, T]$ by varying $\hat{t}$. Therefore, $\hat{\mathbf{u}}(t)$ is absolutely continuous in the interval $[0, T]$ and $\dot{\hat{\mathbf{u}}}(t)$ exist almost everywhere. Also, if for any $t \in [0, T]$, $\dot{\hat{\mathbf{u}}}(t)$ exists then $\dot{\hat{\mathbf{u}}}(t) \in F^{2\gamma}(\hat{\mathbf{u}}(t))$, where $\gamma$ can be made arbitrarily small. Hence,

$$\dot{\hat{\mathbf{u}}}(t) \in F(\hat{\mathbf{u}}(t)), \text{ for a.e. } t \in [0, T].$$

$\square$

