# OpenReview forum: "Directional Convergence Near Small Initializations and Saddles in Two-Homogeneous Neural Networks"
_TMLR — Accepted by TMLR_

### Review · Reviewer_zNgm · 2024-03-03

**Summary Of Contributions:**

This paper studies a directional convergence in training 2-homogeneous neural networks with small initialization. It is shown that weight parameters, when initialized sufficiently close to the origin, converge in direction to a KKT point of some neural correlation function. Then the analyses are extended to certain cases where the network weights are initialized around some saddle points.

**Audience:**

Yes

**Claims And Evidence:**

Yes

**Requested Changes:**

Critical changes: Please see "Weakness"

The authors are also recommended to implement the following change:
1. The neural correlation function can be easily motivated by taking the 1-st order Taylor expansion of the loss function around \hat{y}\_i=0 (this is shown in the proof), which leads to a first-order term $\sum_{i=1}^n \nabla_1\ell(0,y_i) \mathcal{H}_i$, exactly the neural correlation function by noticing that  $\nabla_1\ell(0,y_i)=-y_i$. Such simple derivations in the main paper would help with understanding the NC function. Moreover, based on this derivation, I think the $\ell$ can potentially be any loss function with Lipschtz gradient w.r.t. first term around zero, besides the $l_2$ and logistic loss discussed in the paper.

**Strengths And Weaknesses:**

Strength:
1. This paper is mostly well-written. The theoretical results are clear and sound (proofs are checked, albeit not very carefully, and they make sense).
2. The analyses are rigorous. A critical assumption "initialization is non-branching" (the resulting solution to the differential inclusion is unique) is carefully explained, and the authors provide a great example to justify this assumption.
3. The provided results are general. Prior analyses on directional convergence with small initialization mostly studied two-layer ReLU networks and this work studies almost any 2-homogeneous neural networks.

Weakness:
1. Some definitions are not properly defined in the main paper, for example, "non-negative KKT points".
2. The directional convergence discussed here is not exactly the one shown in other works (Maennel et al., 2018; Boursier et al., 2022;  Wang & Ma, 2023; Min et al., 2024;). This paper studies the directional convergence of all parameters, while the others study, for two-layer ReLU nets, the directional convergence of first-layer weights (via balancedness assumption). Such difference needs to be pointed out and the authors are recommended to draw connections between these two types of directional convergence.

---

> ### Author Response · Authors · 2024-04-24
> **Official Comment by Authors**
>
> We thank the reviewer for their detailed feedback. We have made all the major edits resulting from your feedback in red.
>
> * 'Some definitions are not properly defined in the main paper, for example, non-negative KKT points.' - In the beginning of Section 3, we have added the definition of a non-negative KKT point.
>
> * 'The directional convergence discussed here is not exactly the one shown in other works.' - We agree that the statement of Theorem 5.1 is different from the result of Maennel et al., 2018 and others. However, our corollary for separable neural networks in Section 5.1.2 contains the result of Maennel et al., 2018 and others. In Section 5.1.2 of the revised draft, we have now explicitly stated this connection and the details are provided in the Appendix.
>
> * 'Requested Changes' - We have extended our results for general loss functions which have locally Lipschitz gradient and are definable in some o-minimal structure. We have also added more details in Section 5.1.2 to help readers better understand the motivation for the NCF.

---

### Review · Reviewer_wmhE · 2024-03-28

**Summary Of Contributions:**

The paper explores the training dynamics of shallow neural networks with homogeneous activations.
The main result shows that when the networks' weights are initialized with a sufficiently small norm, the weights stay small for some predetermined time and their direction converges. Moreover, the possible directional limits are also characterized in terms of the KKT points of maximizing the networks' output with the sample's labels.
The authors also establish a similar result when the weights are initialized at a certain type of saddle-points, rather than close to the origin.

**Audience:**

Yes

**Claims And Evidence:**

Yes

**Requested Changes:**

As I noted above my biggest concern is about the assumptions the authors impose in the paper. I feel like an in-depth discussion about the interplay of the different assumptions, and the main limitations could greatly help evaluate the contribution. I list below several specific questions:

- Requiring that the network is 2-homogeneous is a very restrictive assumption. Unless I'm missing a specific example, realistically, it only captures 2-layered ReLU networks (the other examples in the paper are a bit artificial). Since 2-layered ReLU networks already form an important class of examples, I do not hold this restriction against the paper too much, but I do feel like something should be discussed in the paper. Why is this assumption important for the analysis? What happens if the degree of homogeneity is changed (for example 1- or 3-homogeneous)? Do you expect a similar result to hold without this assumption?

- The second assumption I find troubling is that of having non-branching solutions. I completely understand why this assumption is necessary for the analysis and I can also see why it is automatically satisfied in some cases. However, I don't see a reason why it will be satisfied in general. In particular, given the limitation imposed by having a 2-homogeneous network, is there any reason to expect non-branching for ReLU networks? The authors honestly raise this limitation in the main body of the text, but I would be happy to hear/read more details. Essentially, at the moment I am not sure whether the result holds for ReLU networks or not.

- I would also be happy to hear the authors' opinion concerning Assumption 2 in the same context as my previous question.


Minor comments:
- In the Single-layer squared ReLU networks example, the authors could also allow for arbitrary signs for each neuron, making this example slightly more general and relevant.

- Page 4, Line -12: "... Each S_n is *a* set..."

**Strengths And Weaknesses:**

Strengths:
- Understanding the training dynamics of neural networks is an important task. Our current understanding is still far from complete even for the simpler cases of two-layered networks. The paper furthers our understanding of these networks and I feel like it can be a valuable addition to the literature.

- While I did not check all mathematical derivations carefully, the presented outline makes sense and the analysis seems to be non-trivial. The idea that the training dynamics can be coupled with the gradient flow of the correlation function is particularly nice.

Weakness:
- The paper makes some non-trivial assumptions both concerning the network and the dynamics. I feel like those assumptions can severely limit the applicability of the results. I expand on this point below.

---

> ### Author Response · Authors · 2024-04-24
> **Official Comment by Authors**
>
> Thank you for the feedback and detailed review. We have made all the major edits resulting from your feedback in blue. We also note that in the revised version, Theorem 5.1 has been generalized for a broader class of loss functions. However, the overall proof structure remains almost the same.
>
> * 'Requiring that the network is 2-homogeneous is a very restrictive assumption.' - We have added Section 5.3 in the revised draft which describes the importance of 2-homogeneity for our analysis. The main issue is the relative scaling between the weights and the gradient at small initialization. For $1$-homogeneous, the gradient could be large at origin and thus gradient flow can quickly escape fom the neighborhood of the origin. For deeper homogeneous models, near the origin, the gradient will be smaller than the weights and thus, in the initial stages of training, neither the direction nor the magnitude of the weights seems to change much.
>
> * 'The second assumption I find troubling is that of having non-branching solutions.' - As we have shown in the paper, the initializations that do not satisfy the non-branching assumption can exist. There are two ways to tackle this problem. First, is to show that directional convergence happens even if the initialization is not non-branching. We are not sure if it can be proved. The alternative is to characterize the set of non-branching initializations. For example, consider the function $f(u_1,u_2) = u_1|u_2|$ discussed in the paper. We can show that except for the set $\\{u_2 = 0, u_1> 0\\}$, all the other points are non-branching, which implies that almost all points are non-branching; for details see Lemma E.2 in the Appendix of the revised draft. Whether a similar behavior happens for a broader class of datasets and neural networks is an interesting future direction, and a positive answer would justify the non-branching assumption in those cases. However, showing that non-branching assumption is almost always satisfied in general seems to be extremely difficult, even for a two-layer (Leaky) ReLU neural network. For two-layer Leaky-ReLU neural networks, we have added Lemma 5.4 which shows that there exists sets, near certain KKT points, within which all points are non-branching. While this lemma is clearly far from proving that almost all points are non-branching, we hope that it could be the first step towards it and can help future invetsigations. We have added the above explanation at the end of Section 5.1.1.
>
> * 'I would also be happy to hear the authors' opinion concerning Assumption 2 in the same context as my previous question.' - We first note that in the revised version Assumption 2 has become Assumption 3. Also, we have slightly modified the assumption to make it more general. Our main aim with Assumption 3 is to provide as general as possible conditions to show that directional convergence can also happen at certain saddle points of the loss function. Now, it depends on the problem if saddle points satisfying Assumption 3 exist or not. We can give one simple example showing existence of such a saddle point.
> Suppose $L(u_1,u_2,a_1,a_2) = \frac{1}{2}\left((u_1|u_2|-1)^2+(a_1|a_2|-1)^2\right)$, and $\bar{u}_1 = \bar{u}_2 = 1$ and $\bar{a}_1 = \bar{a}_2 = 0$. Here, $\mathbf{w}_n = [u_1,u_2]$ and $\mathbf{w}_z = [a_1,a_2]$. Since $L(u_1,u_2,0,0)$ have locally Lipschitz gradient near $[\bar{u}_1, \bar{u}_2]$, it can be easily verified that $[\bar{u}_1, \bar{u}_2, \bar{a}_1, \bar{a}_2  ]$ is a saddle point of $L(u_1,u_2,a_1,a_2) $, and $[\bar{u}_1, \bar{u}_2]$ satisfies Assumption 3 with respect to $L(u_1,u_2,0,0)$.
>
> * We have edited the paper based on the minor comments.

---

### Review · Reviewer_ziAa · 2024-04-10

**Summary Of Contributions:**

This paper studied the directional convergence of gradient flow near small initialization.

**Audience:**

Yes

**Claims And Evidence:**

Yes

**Requested Changes:**

1. The result in the main theorem does not seem intuitive. Previous works can ``calculate'' the convergent directions for some specific dataset, such as $\frac{1}{n}\sum_{i=1}^n y_i x_i$. Could the authors provide some specific examples and calculate the specific expression of the KKT directions of $\max_{||w||\_2=1} z^\top H(X; w)$ in the main theorem? I think it can great help the readers understand this theorem.

2. The KKT directions may be not unique. Is it hopeful to provide a fine-grained characterization about the uniqueness of the convergent direction (under some stricter conditions)?

3. Some other related works. [1] about theoretical analysis of initial condensation; [2][3] about the entire training dynamics.

[1] Chen et al. Phase diagram of initial condensation for two-layer neural networks. (CSIAM Transactions on Applied Mathematics )

[2] Wang and Ma. Early Stage Convergence and Global Convergence of Training Mildly Parameterized Neural Networks. (NeurIPS 2022)

[3] Safran et al. On the Effective Number of Linear Regions in Shallow Univariate ReLU Networks: Convergence Guarantees and Implicit Bias. (NeurIPS 2022)

**Strengths And Weaknesses:**

- Strength. The initial directional convergence phenomenon has been widely observed. However, most previous theoretical works are limited to specific dataset. In this paper, the authors establish a clean theorem to characterize this phenomenon  for general dataset. They prove that the net converges to the KKT directions of some constrained optimization problem.


 - Weaknesses. Please refer to Requested Changes.

---

> ### Author Response · Authors · 2024-04-24
> **Official Comment by Authors**
>
> We thank the reviewer for their detailed feedback.
>
> * 'The result in the main theorem does not seem intuitive.' - We have added some examples in the Appendix G of the revised draft where the KKT points of the NCF can be computed exactly. However, for general case, it may not be possible to get a closed form expression.
>
> * 'Is it hopeful to provide a fine-grained characterization about the uniqueness of the convergent direction (under some stricter conditions)?' - We do not think it is hopeful. In fact, in the simple examples considered in the main part of the paper (Figure 2 and 3), we observe that there are multiple KKT points. Also, in the examples presented in Appendix G, which are quite simple, there are multiple KKT points.
>
> * 'Some other related works.' - Thank you for suggesting these references which have been added into our paper.

---

### Review · Reviewer_GZga · 2024-04-11

**Summary Of Contributions:**

The paper studies the directional convergence of two homogenous networks initialized with small weights. In particular, the authors prove that the weight vectors move towards maximizing directions of the neural correlation function defined on the sphere while their norm remains small. The time to exit this small norm phase is also given. A similar analysis is done to study escape behavior near other saddles far from the origin.

**Audience:**

Yes

**Claims And Evidence:**

Yes

**Requested Changes:**

Question: Can the authors comment on why it is not possible to show the saddle escape behavior for logistic loss although this was possible for the saddle point at the origin?

**Strengths And Weaknesses:**

Strengths
* The problem formalizes and proves the observations of Maennel et al 2018.
* The analysis seems clean and without issues (though I did not check the details carefully).
* Local analysis of gradient flow is relevant as describing the full gradient flow trajectory is very difficult (if not impossible) for generic data even for the simple case of a shallow network with ReLU activation.

Weaknesses
* Writing feels technical but it is clear
* Experiments showing escape behavior near the saddles would be valuable

---

> ### Author Response · Authors · 2024-04-24
> **Official Comment by Authors**
>
> We thank the reviewer for their detailed feedback.
>
> * 'Can the authors comment on why it is not possible to show the saddle escape behavior for logistic loss although this was possible for the saddle point at the origin?' - For logistic loss, the saddle points could exist at infinity, even for simple instances. For example, suppose $\\{\mathbf{x}\_i,y_i\\}_{i=1}^n$ is a linearly separable data such that, for some $\hat{\mathbf{w}}$, $y_i\hat{\mathbf{w}}^\top\mathbf{x}\_i \geq 1$, for all $i$. Now, suppose we have a 2-layer ReLU neural network with two neurons, $\mathcal{H}(\mathbf{x};\\{v_j,\mathbf{u}\_j\\}\_{j=1}^2) = v_1\sigma(\mathbf{u}\_1^\top\mathbf{x})+v_2\sigma(\mathbf{u}\_2^\top\mathbf{x})$. Then, it can be shown that as $t\rightarrow \infty$,   $t[1, \hat{\mathbf{w}}, 0, \mathbf{0}]$ becomes a saddle point of $L(\\{v_j,\mathbf{u}\_j\\}\_{j=1}^2) = \sum\_{i=1}^n\ln(1+\exp(-y_i\mathcal{H}(\mathbf{x}\_i;\\{v_j,\mathbf{u}\_j\\}\_{j=1}^2))$ . However, it is not clear to us how to analyze the gradient flow dynamics near saddle points which are at infinity.
>
>
> * We have added an experiment in Appendix D.3 showing escape behavior near the saddles.

---

### Decision · Action_Editor_Wg6Z · 2024-05-29

**Recommendation:** Accept as is

**Comment:**

This paper investigates the directional convergence in training 2-homogeneous neural networks with small initialization. It demonstrates that when weight parameters are initialized sufficiently close to the origin, they converge directionally to a KKT point of a specific neural correlation function. The analysis is further extended to scenarios where the network weights are initialized near certain saddle points.

This is a very solid theoretical work. While the analysis of directional convergence with small initialization is not new, this work extends previous analysis from 2-layer ReLU networks to almost any 2-homogeneous neural network. All the reviewers acknowledge the paper's contribution and believe the findings will interest the deep learning theory community. I agree with the reviewers and recommend acceptance.

**Audience:**

The result will be of interest to researchers working in the analysis, optimization, and theory of deep learning models.

**Claims And Evidence:**

This paper investigates the directional convergence in training 2-homogeneous neural networks with small initialization. It demonstrates that when weight parameters are initialized sufficiently close to the origin, they converge directionally to a KKT point of a specific neural correlation function. The analysis is further extended to scenarios where the network weights are initialized near certain saddle points.

Prior work has mostly studied for two-layer ReLU networks; this work extends the analysis to almost any 2-homogeneous neural networks. The claims made in this work are supported by rigorous analyses and proofs.